# The co-inhibitory receptor TIGIT promotes tissue-protective functions in T cells

Camilla Panetti[1], Rahel Daetwyler [1], Anja Moncsek [1], Nikolaos Patikas [2], Andreas Agrafiotis[3,4], Adelynn Tang[5,6], Francesco Andreata [7,8], Valeria Fumagalli [7,8], Jean De Lima [9], Lifen Wen[5], Carolyn G. King[9], Ajithkumar Vasanthakumar [5,6,10,11], Matteo Iannacone [7,8], Axel Kallies [5,12], Alexander Yermanos[3,13,14], Martin Hemberg[2] & Nicole Joller [1,15] ✉

The co-inhibitory receptor TIGIT suppresses excessive immune responses in autoimmune conditions while also restraining antitumor immunity. In viral infections, TIGIT alone does not affect viral control but has been shown to limit tissue pathology. However, the underlying mechanisms are incompletely understood. Here we found TIGIT+ T cells to express not only an immunoregulatory gene signature but also a tissue repair gene signature. Specifically, after viral infection, TIGIT directly drives expression of the tissue growth factor amphiregulin (Areg), which is strongly reduced in the absence of TIGIT. We identified regulatory T ($T_{reg}$) cells, but not CD8+ T cells, as the critical T cell subset mediating these tissue-protective effects. In $T_{reg}$ cells, TIGIT engagement after T cell antigen receptor stimulation induces the transcription factor Blimp-1, which then promotes Areg production and tissue repair. Thus, we uncovered a nonclassical function of the co-inhibitory receptor TIGIT, wherein it not only limits immune pathology by suppressing the immune response but also actively fosters tissue regeneration by inducing the tissue growth factor Areg in T cells.

Co-inhibitory receptors are essential for maintaining immune balance, preventing excessive immune activation and tissue damage under normal conditions and during inflammation[1]. T cell immunoglobulin and ITIM domain (TIGIT) is one of these co-inhibitory receptors, and it exerts its inhibitory function by directly inhibiting effector T cell activation as well as tolerizing dendritic cells through engagement of its ligand CD155 (refs. 2,3). Moreover, TIGIT is constitutively expressed on regulatory T ($T_{reg}$) cells and enhances their suppressive capacity[4]. Loss of TIGIT results in increased susceptibility to autoimmunity, while promoting tumor clearance[5]. Similar to other co-inhibitory receptors, TIGIT is expressed after T cell activation, resulting in its upregulation during infections and sustained expression on exhausted T cells in chronic infections[6,7]. Interestingly, while TIGIT modulates the antiviral response following lymphocytic choriomeningitis virus (LCMV) infection, it does not alter viral control but rather limits the tissue pathology resulting from LCMV and influenza infection[7]. However, the underlying mechanism for this is still poorly understood.

[1]Department of Quantitative Biomedicine, University of Zurich, Zurich, Switzerland. [2]The Gene Lay Institute of Immunology and Inflammation of Brigham and Women's Hospital, Massachusetts General Hospital, and Harvard Medical School, Boston, MA, USA. [3]Department of Biosystems Science and Engineering, ETH Zurich, Basel, Switzerland. [4]Institute of Microbiology, ETH Zurich, Zurich, Switzerland. [5]Department of Microbiology and Immunology, The Peter Doherty Institute for Infection and Immunity, The University of Melbourne, Parkville, Victoria, Australia. [6]La Trobe University, Bundoora, Victoria, Australia. [7]Division of Immunology, Transplantation, and Infectious Diseases, IRCCS San Raffaele Scientific Institute, Milan, Italy. [8]Vita-Salute San Raffaele University, Milan, Italy. [9]Department of Biomedicine, University of Basel, Basel, Switzerland. [10]Olivia Newton-John Cancer Research Institute, Heidelberg, Victoria, Australia. [11]Peter MacCallum Cancer Centre, Melbourne, Victoria, Australia. [12]Institute of Molecular Medicine & Experimental Immunology, University Hospital Bonn, Bonn, Germany. [13]Center for Translational Immunology, University Medical Center Utrecht, Utrecht, the Netherlands. [14]Botnar Institute of Immune Engineering, Basel, Switzerland. [15]Center for Human Immunology, University of Zurich, Zurich, Switzerland. ✉e-mail: nicole.joller@uzh.ch

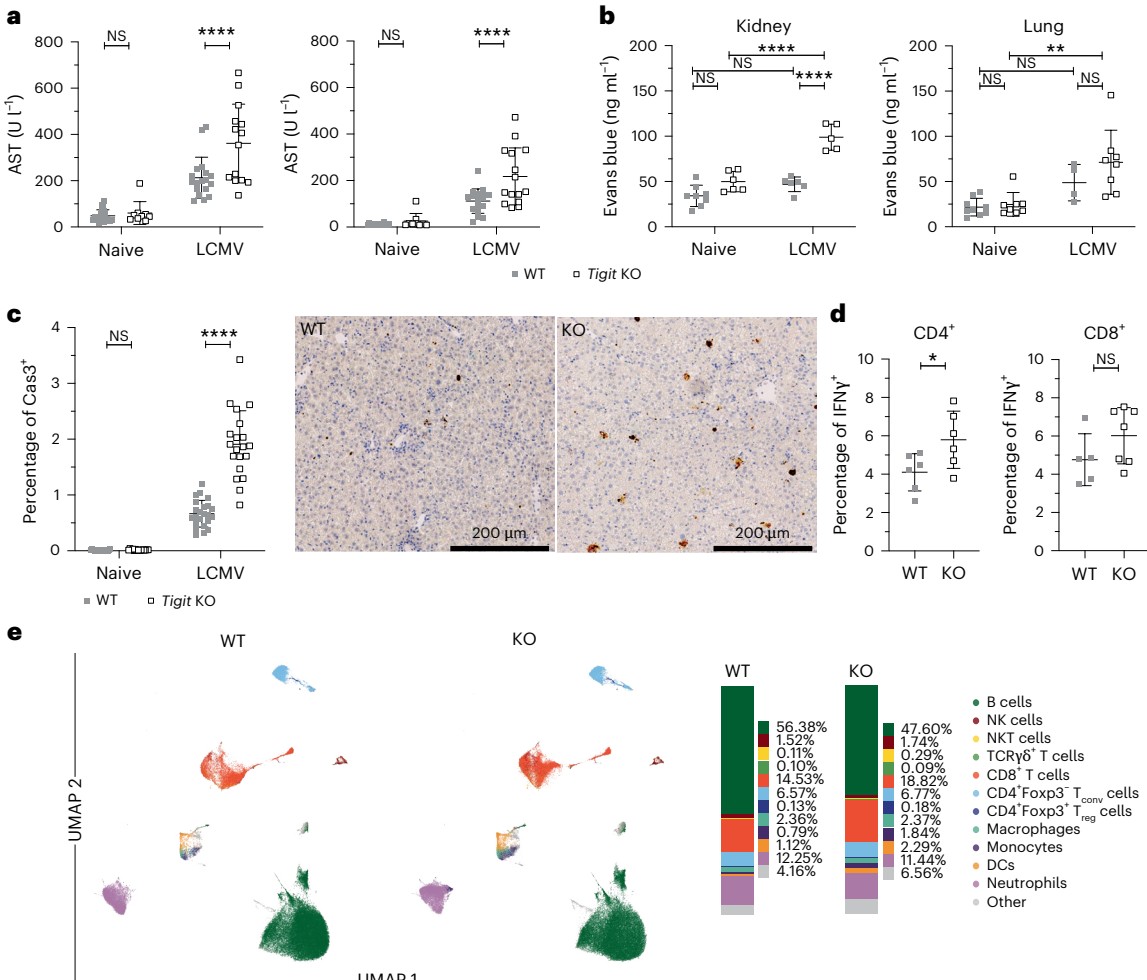

**Fig. 1 | *Tigit*-KO mice show increased tissue pathology following infection.**
WT and *Tigit*-KO mice were left naive or were infected with LCMV Cl13. **a**, AST
and ALT levels in serum (data are shown as mean ∓ s.d.; pool of five independent
experiments); NS, not significant. **b**, Colorimetric quantification of Evans Blue
extravasation from kidney and lung (data are shown as mean ∓ s.d.; pool of two
independent experiments) on day 10 postinfection. **c**, Quantification (left) and
representative images (right) of caspase-3 (Cas3) staining as a marker of cellular
apoptosis in infected liver sections (data are shown as mean ∓ s.d.; representative
ROIs *n* = 19, 15, 23 and 20, biological replicates = 2–3). **d**, Frequencies of

interferon-γ⁺CD4⁺ (IFNγ⁺CD4⁺) and IFNγ⁺CD8⁺ T cells following LCMV
glycoprotein peptide restimulation at the peak of LCMV Cl13 infection in
the spleen (data are shown as mean ∓ s.d.; pool of two to three independent
experiments, *n* = 6 (CD4⁺) and *n* = 5 and 7 (CD8⁺)). Data were analyzed by two-
way analysis of variance (ANOVA) with a Šídák's post hoc test (**a** and **c**), two-way
ANOVA with a Tukey's post hoc test (**b**) or unpaired two-sided *t*-test (**d**).
**e**, Uniform manifold approximation projections (UMAPs) and relative
composition of immune cells in the spleen on day 7 after LCMV Cl13 infection;
NK, natural killer; NKT, natural killer T cells; DCs, dendritic cells.

In recent years, it has become increasingly evident that T cells,
and T$_{reg}$ cells in particular, play a crucial role in maintaining tissue
homeostasis by actively contributing to tissue regeneration and repair[8].
T$_{reg}$ cells accumulate at the site of tissue injury, and their depletion
results in delayed tissue regeneration, exacerbated tissue damage and
fibrosis[9–12]. At sites of tissue damage, T$_{reg}$ cells produce the growth fac-
tor amphiregulin (Areg), which binds to the epidermal growth factor
receptor (EGFR) to promote tissue repair across a range of settings,
including muscle repair, acute lung injury and remyelination of dam-
aged axons[12,13]. In the context of viral infections, T$_{reg}$ cells have been
shown to produce Areg following influenza infection and are thereby
protective against infection-induced lung damage. Areg production
by T$_{reg}$ cells was critical for limiting tissue damage, as mice lacking
Areg specifically in T$_{reg}$ cells exhibited increased lung pathology fol-
lowing influenza infection[11]. T$_{reg}$ cell-derived Areg is sensed by EGFR⁺
mesenchymal cells, which in turn enable epithelial cell regeneration[14].
Similarly, T$_{reg}$ cell-derived Areg mitigates tissue damage following
severe acute respiratory syndrome coronavirus 2 (SARS-CoV-2) infec-
tion[15]. In these settings, alarmins like interleukin-33 (IL-33) and IL-18 can

promote Areg production by T$_{reg}$ cells, while Notch-4 restrains it, thus
limiting tissue repair[11,15]. However, whether additional signals regulate
Areg production through T$_{reg}$ cells during infection remains unknown.

As TIGIT can limit tissue damage during viral infection, we hypoth-
esized that it may act as an additional signal promoting the repair
functions of T cells. To test this, we conducted comprehensive tran-
scriptional profiling of T cells following LCMV infection and found
a transcriptional repair signature to be enriched in *Tigit*-expressing
T cells. Notably, *Areg* was highly coexpressed with *Tigit*, and we demon-
strate that TIGIT is functionally required for Areg production in T cells.
Furthermore, we show that TIGIT expression in T$_{reg}$ cells, but not in
CD8⁺ T cells, is essential for limiting tissue damage following infection.
Areg-producing T$_{reg}$ cells present upon infection are clonally expanded,
and both TIGIT and TCR signaling are necessary for Areg induction by
in vivo-primed T cells. Finally, we found that Blimp-1 acts downstream
of TIGIT to induce Areg production. Thus, the co-inhibitory receptor
TIGIT not only limits immune pathology by suppressing excessive
immune responses but also promotes tissue regeneration by inducing
Areg in T$_{reg}$ cells.

## Results

### Loss of TIGIT exacerbates pathology following viral infection

We have previously shown that TIGIT engagement limits immune pathology following viral infection[7]. Interestingly, TIGIT engagement reduced infection-induced pathology independent of viral clearance, leading us to hypothesize that TIGIT may exert a tissue-protective function that is independent of its immune-suppressive properties. We first confirmed that *Tigit*-knockout (KO) mice indeed show increased pathology, as would be expected[7]. To this end, C57BL/6 (B6) wild-type (WT) and *Tigit*-KO mice (Extended Data Fig. 1a–c) were infected with LCMV clone 13 (Cl13), which induces a high degree of tissue pathology peaking at day 10 after infection (Extended Data Fig. 1d–f). After infection, *Tigit*-KO mice showed a significant increase in pathology, as assessed by measuring the serum aminotransferases aspartate aminotransferase (AST) and alanine aminotransferase (ALT) as indicators of liver injury (Fig. 1a). Similarly, kidneys and lungs of *Tigit*-KO mice also showed exacerbated vascular leakage, as indicated by increased levels of Evans blue dye, compared to WT mice (Fig. 1b). Noticeably, increased tissue damage was observed only following LCMV challenge; hence, lack of TIGIT does not affect tissue integrity at steady state but only following infection. Additionally, we observed increased cellular infiltrates in the livers of *Tigit*-KO mice compared to WT mice, and caspase-3 staining of liver sections revealed increased apoptosis of hepatocytes in *Tigit*-KO mice after LCMV Cl13 infection (Fig. 1c and Extended Data Fig. 1g). Despite these clear differences in pathology, only a relatively small increase in proinflammatory cytokines was observed in T cells from *Tigit*-KO mice (Fig. 1d), the response in natural killer cells, which also express TIGIT, was unaltered (Extended Data Fig. 1h), and broad immunophenotyping analysis of infected spleen and lung revealed no major alteration in the immune cell composition between WT and *Tigit*-KO mice (Fig. 1e and Extended Data Fig. 1i). These results are in line with our previous data[7,16] and confirm that TIGIT is important for limiting tissue pathology following viral infection and that loss of TIGIT exacerbates tissue damage. Nevertheless, TIGIT has limited impact on the proinflammatory response in this context, suggesting that it may have additional functions contributing to tissue protection.

### TIGIT+ T cells display transcriptional repair signature

To investigate how TIGIT limits tissue damage after infection, we performed single-cell RNA sequencing (scRNA-seq) of T cells from the spleen and lung of naive and LCMV Cl13-infected WT and *Tigit*-KO mice, and we enriched for CD4+Foxp3+ $T_{reg}$ cells to obtain a solid representation of all T cell subsets (Extended Data Fig. 2a). We identified 15 T cell clusters (Fig. 2a), which were annotated based on commonly used transcriptional profiles (Extended Data Fig. 2b). As expected, comparison of T cells from naive versus LCMV-infected mice revealed the emergence of proliferating T cell clusters marked by *Ki67* expression with infection, as well as that of exhausted CD8+ T cells and T helper 1 ($T_H$1)-like CD4+ T cells, marked by high expression of *Tox* and *Pdcd1* (Fig. 2a and Extended Data Fig. 2b–d). Also, clusters rich in *Cd44*, indicative of T cell activation, *Tbx21*, highlighting the $T_H$1-dominated immune

response after LCMV infection, and *Lag3*, an early exhaustion marker, were only present after infection. In line with previous reports[6,7], *Tigit* was expressed at higher levels in activated T cells and was induced upon infection (Extended Data Fig. 2b,c). Given the increasing recognition of T cells as mediators of tissue repair[11,12] and our findings of increased pathology after TIGIT loss, we sought to determine whether *Tigit* expression might be linked to tissue repair. Indeed, *Tigit* showed clear coexpression with a curated list of tissue repair genes, including *Areg* and *Batf*, characteristic of tissue repair $T_{reg}$ cells, as well as genes more broadly associated with tissue repair, such as *Klf10* and *Tgfb1* (ref. 12; Fig. 2b and Extended Data Fig. 2e). Although *Tigit* expression correlated with this repair signature in all T cell subsets, the specific co-regulated genes varied slightly depending on the type of T cell (Fig. 2c). For instance, *Ikzf2* (Helios), which marks thymically derived $T_{reg}$ cells, is part of the tissue repair transcriptome of $T_{reg}$ cells and CD4+, but not CD8+, effector T cells. Interestingly, *Areg*, encoding Areg, a key mediator of T cell-mediated tissue protection following viral infection[11], was highly coexpressed with *Tigit* in all T cell subsets, particularly in the effector $T_{reg}$, exhausted-like CD4+ and exhausted CD8+ T cell clusters (Fig. 2d). These data suggest a module consisting of *Areg* and *Tigit* that might be involved in limiting tissue damage upon infection.

To address whether *Areg* induced in T cells is translated into protein to mediate a tissue-protective function during viral infection, we quantified its protein expression. We observed strong Areg induction in spleen and lung $T_{reg}$ cells after LCMV Cl13 infection (Fig. 2e,f and Extended Data Fig. 3a). While CD4+ and CD8+ conventional T ($T_{conv}$) cells produced lower amounts, Areg was also induced with infection (Fig. 2f and Extended Data Fig. 3a). Furthermore, the kinetics of Areg expression over the course of infection clearly mirrored AST and ALT dynamics and thus liver pathology (Fig. 2e and Extended Data Fig. 1d), hinting at a role for $T_{reg}$ cell-derived Areg in the resolution of systemic pathology. Importantly, the coexpression of *Areg* and *Tigit* observed at the mRNA level was also seen at the protein level in both spleen and lung, where TIGIT+ $T_{reg}$ cells were the main producers of Areg and TIGIT was upregulated following infection (Fig. 2g and Extended Data Fig. 3b-d). Indeed, frequencies of Areg+ cells were highest among TIGIT+ $T_{reg}$ cells compared to their TIGIT− counterparts, and the infection-induced increase in Areg was restricted to this population (Fig. 2h). TIGIT also positively correlated with Areg in all T cell populations (Extended Data Fig. 3e). Of note, baseline Areg production observed in naive mice was not restricted to TIGIT+ T cells (Fig. 2h), which fits with our finding that loss of TIGIT does not impair tissue integrity at steady state. Overall, our results reveal that TIGIT expression in T cells correlates with a tissue repair signature that includes *Areg*. Furthermore, infection-induced Areg expression is restricted to the TIGIT+ $T_{reg}$ cell population, suggesting a functional link between TIGIT, Areg and tissue repair in T cells.

### TIGIT regulates Areg expression

To determine whether there is a functional link between TIGIT and Areg, we infected WT mice with LCMV Cl13 and treated them with blocking antibody to TIGIT (1B4) or with an isotype control (IgG1). Indeed, TIGIT

---

**Fig. 2 | TIGIT-expressing T cells are enriched for a transcriptional repair signature. a–d**, scRNA-seq was performed on T cells from the spleen and lung of naive or LCMV Cl13-infected (day 10) WT and *Tigit*-KO mice. **a**, UMAP of the full 198,645 T cell dataset annotated by the identified T cell subsets. T cells are split into naive or LCMV-infected groups, and each panel contains cells from the lung and spleen and WT and *Tigit*-KO conditions. **b**, Coexpression pattern of *Tigit* and repair score signature projected in the UMAP. **c**, Heat maps showing the Pearson *r* correlation coefficient and dendrograms highlighting coexpression patterns between the genes in the repair signature. Each value was calculated using single-cell expression values from the subsequent major cell type. In addition to the repair signature, *Tigit* was included in the gene set. **d**, Coexpression pattern of *Tigit* and *Areg* projected in the UMAP. **e**, Areg+ cell frequencies among $T_{reg}$ cells over time during LCMV Cl13 and LCMV WE infection determined by flow

cytometry (data are shown as mean + s.d.; pool of two independent experiments; data were analyzed by unpaired two-sided *t*-test). **f**, Areg quantification in splenic $T_{reg}$ cells, CD4+ $T_{conv}$ cells and CD8+ T cells of naive, LCMV Cl13- and LCMV WE-infected mice on day 10 (data are shown as mean ∓ s.d.; pool of three to five experiments for LCMV Cl13 and pool of two experiments for LCMV WE; data were analyzed by ordinary one-way ANOVA with a Tukey's post hoc test; $T_{reg}$ cells *n* = 10, 27 and 4; CD4+ $T_{conv}$ cells *n* = 18, 23 and 4; CD8+ cells *n* = 17, 13 and 4). **g**, Representative FACS plots showing TIGIT and Areg expression in splenic T cells from naive and LCMV Cl13-infected mice. **h**, Areg frequencies in TIGIT+ versus TIGIT− splenic $T_{reg}$ cells from naive and LCMV Cl13-infected mice (data are shown as mean ∓ s.d.; pool of six independent experiments; data were analyzed by two-way ANOVA with a Tukey's post hoc test).

blockade led to reduced Areg production compared to the control group (Fig. 3a). This was true for $T_{reg}$ cells and CD4$^+$ and CD8$^+$ $T_{conv}$ cells, even though Areg levels were much lower in the latter two populations. Similarly, comparison of scRNA-seq data from infected WT and *Tigit*-KO mice revealed reduced *Areg* expression in all three KO T cell populations

(Fig. 3b). Similarly, at the protein level, Areg induction was severely impaired in both the spleen and lung of *Tigit*-KO mice after both LCMV and influenza infection, but not in steady-state $T_{reg}$ cells (Fig. 3c and Extended Data Fig. 4a–d), demonstrating a functional link between TIGIT and Areg following infection. Importantly, this reduction was

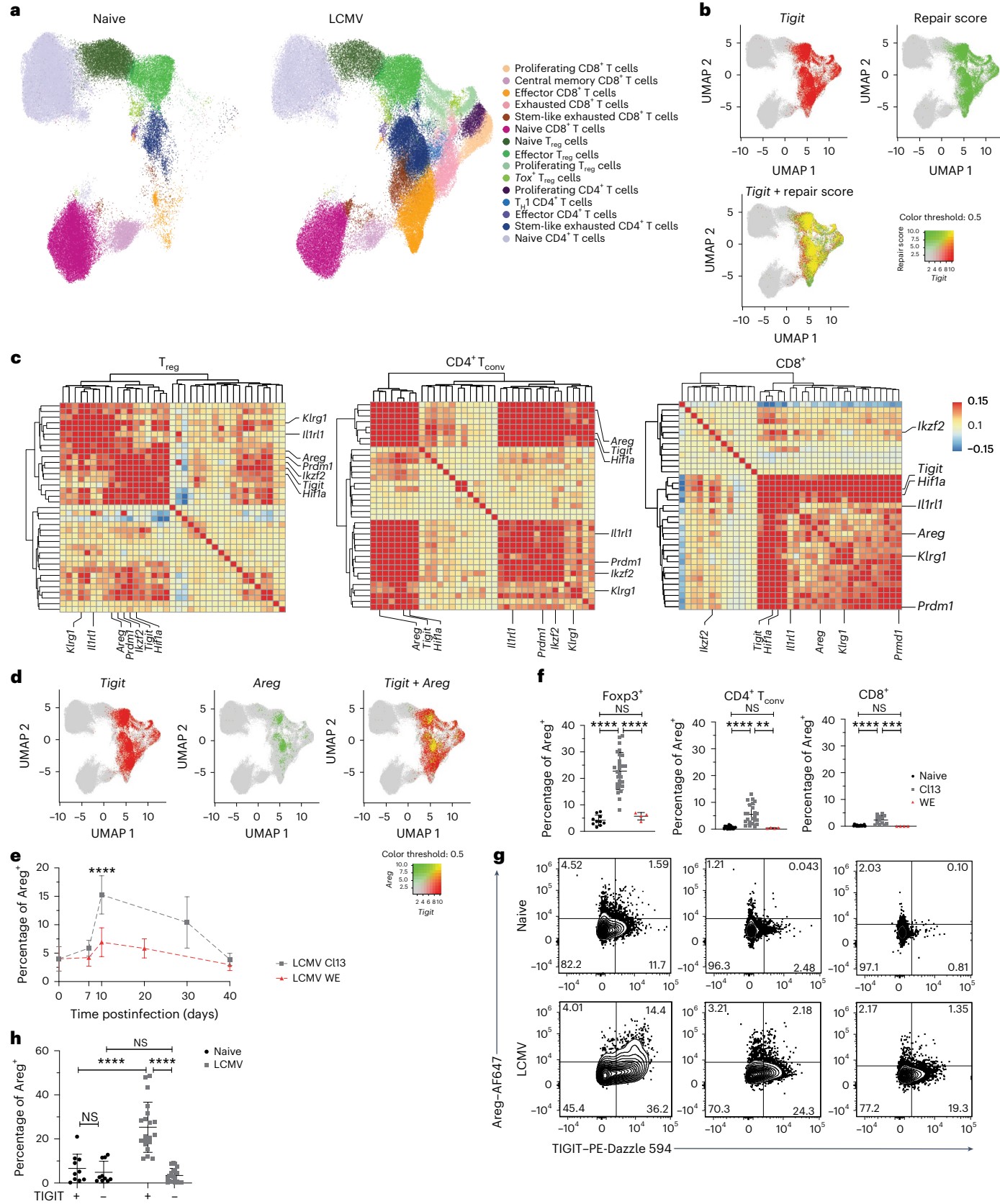

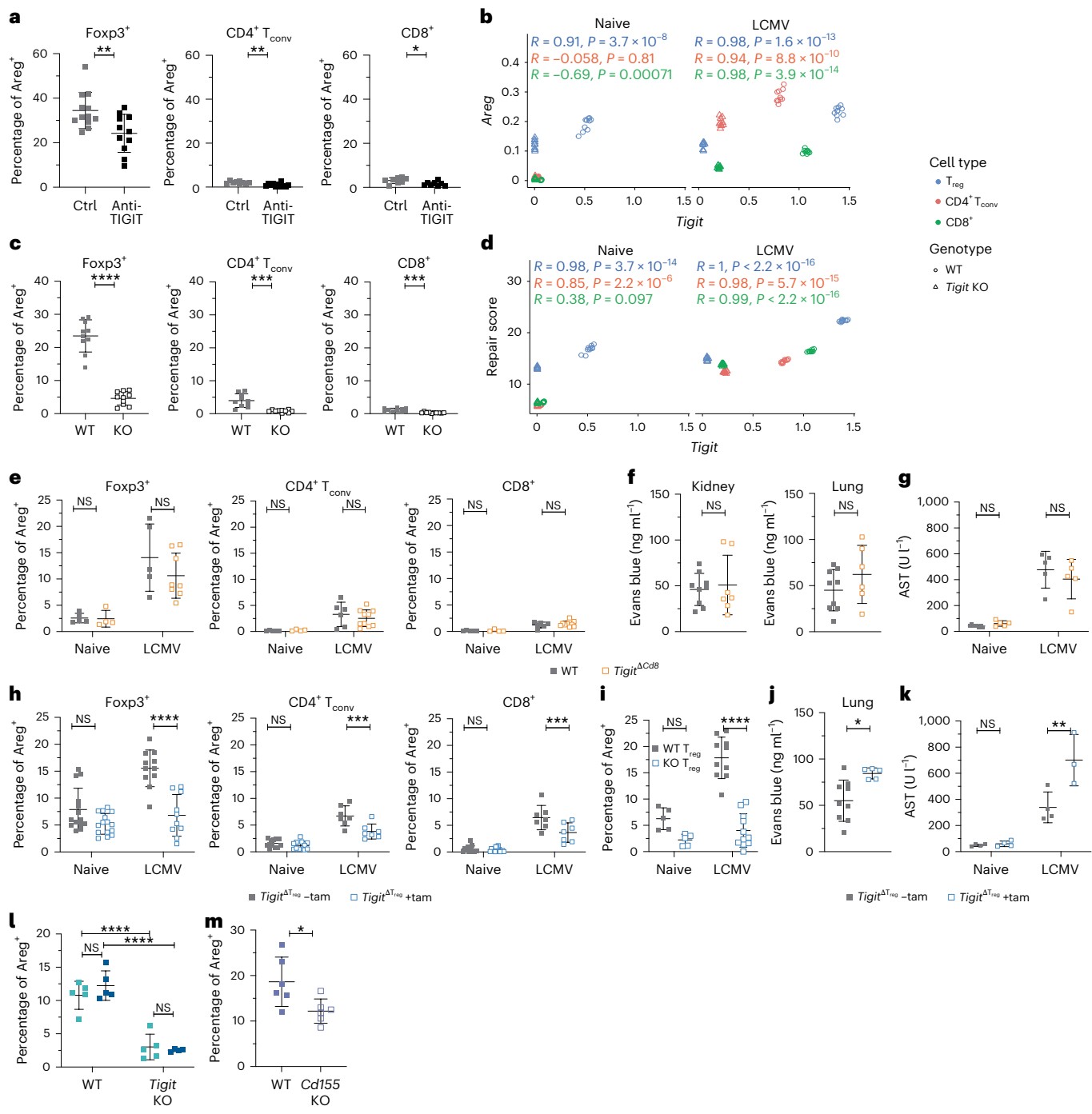

**Fig. 3 | Areg expression is TIGIT dependent. a,c,** Areg in splenic T cells from LCMV-infected mice treated with anti-TIGIT or isotype control (**a**; data are shown as mean ∓ s.d.; pool of three independent experiments; data were analyzed by unpaired two-sided *t*-test) and LCMV-infected *Tigit*-KO and WT mice (**c**; data are shown as mean ∓ s.d.; pool of two independent experiments; data were analyzed by unpaired two-sided *t*-test); Ctrl, control. **b,d,** Scatter plots of scRNA-seq data showing the expression pattern of *Tigit* and *Areg* (**b**) or *Tigit* and repair score (**d**; two-sided *P* value of Pearson correlation). **e–k,** LCMV-infected or naive conditional *Tigit*-KO mice and controls. *Tigit*[ΔCd8] and WT mice were analyzed for Areg expression in splenic T cells (**e**; data are shown as mean ∓ s.d.; pool of three independent experiments; data were analyzed by two-way ANOVA with a Tukey's post hoc test), Evans Blue extravasation (**f**; data are shown as mean ∓ s.d.; pool of two independent experiments; data were analyzed by unpaired two-sided *t*-test) and AST levels (**g**; data are shown as mean ∓ s.d.; pool of two independent experiments; data were analyzed by two-way ANOVA with a Tukey's post hoc test). *Foxp3*[GFPcreERT2] × *Tigit*[ΔTreg] mice with tamoxifen (+tam) or vehicle (−tam)

treatment were analyzed for Areg expression in splenic T cells (**h**; data are shown as mean ∓ s.d.; pool of four independent experiments; data were analyzed by two-way ANOVA with a Tukey's post hoc test), Areg expression in WT and *Tigit*-KO splenic T[reg] cells from tamoxifen-treated heterozygous females (**i**; data are shown as mean ∓ s.d.; pool of two independent experiments; data were analyzed by two-way ANOVA with a Tukey's post hoc test), Evans Blue extravasation (**j**; data are shown as mean ∓ s.d.; pool of two independent experiments; data were analyzed by unpaired *t*-test) and AST levels (**k**; data are shown as mean ∓ s.d.; one representative experiment; data were analyzed by two-way ANOVA with a Tukey's post hoc test; *n* (naive and LCMV −tam) = 4; *n* (LCMV +tam) = 3). **l,m,** Areg expression in splenic T[reg] cells from LCMV-infected mice treated with anti-CD226 or isotype control (**l**; data are shown as mean ∓ s.d.; representative experiment; data were analyzed by two-way ANOVA with a Tukey's post hoc test, *n* = 4 (KO anti-CD226) and 5 (all others)) and in *Cd155*-KO and WT mice (**m**; data are shown as mean ∓ s.d.; pool of three independent experiments; data were analyzed by unpaired two-sided *t*-test).

seen not only for *Areg* but also for the whole repair signature, which showed the highest score in WT CD8⁺ T cells as well as T_reg cells, but was reduced in *Tigit*-KO cells (Fig. 3d). Together, these findings demonstrate that TIGIT can act as a mediator of tissue repair upstream of Areg and that its expression is essential for infection-induced Areg production.

### TIGIT-dependent repair is mediated through Foxp3⁺ T_reg cells

Having established TIGIT's functional role in inducing Areg, we next addressed through which T cell subtype TIGIT exerts its protective function. As we observed the highest repair scores in CD8⁺ and CD4⁺Foxp3⁺ T cells, we focused on these two T cells subsets. To assess whether TIGIT's repair function was mediated through CD8⁺ T cells, we used *Cd8*^Cre × *Tigit*^fl/fl mice, in which the E8I *Cd8* enhancer region drives expression of the Cre recombinase, resulting in *Tigit* deletion specifically in peripheral CD8⁺, but not CD4⁺, T cells (Extended Data Fig. 4e). We observed a similar induction of Areg after LCMV infection in *Cd8*^Cre × *Tigit*^fl/fl mice and littermate control mice (Fig. 3e), demonstrating that TIGIT expression in CD8⁺ T cells is not required for the upregulation of Areg. Furthermore, pathology, determined by vascular permeability in the kidney, lung, liver and spleen, was not altered in *Cd8*^Cre × *Tigit*^fl/fl mice compared to control mice (Fig. 3f and Extended Data Fig. 4f). AST levels, as indicators of liver pathology, were also comparable between the two genotypes (Fig. 3g). To determine whether TIGIT expression on CD8⁺ T cells might indirectly influence tissue repair pathways, we analyzed the expression of key transcription factors and cytokines associated with tissue repair but did not observe any change following *Tigit* deletion in CD8⁺ T cells (Extended Data Fig. 4g). We further examined whether T_reg cells in *Cd8*^Cre × *Tigit*^fl/fl mice displayed phenotypic changes that might compensate for a potential loss of repair function of the *Tigit*-KO CD8⁺ T cell compartment but observed no such alterations (Extended Data Fig. 4h). Thus, we concluded that in the context of systemic viral infection, TIGIT expression on CD8⁺ T cells does not equip them with a specific repair function. Thus, despite the high transcriptional repair score observed in CD8⁺ T cells, TIGIT expression in this subset does not contribute, directly or indirectly, to tissue protection during viral infection.

Next, we tested whether TIGIT expression in T_reg cells was required to limit tissue pathology following LCMV infection. To this end, we generated mice that allow for tamoxifen-induced deletion of *Tigit* specifically in T_reg cells (*Foxp3*^GFPcreERT2 × *Tigit*^fl/fl; Extended Data Fig. 4i,j). Indeed, T_reg cells from *Foxp3*^GFPcreERT2 × *Tigit*^fl/fl mice treated with tamoxifen (but not control mice) showed strong impairment in Areg induction following LCMV infection (Fig. 3h), while baseline *Areg* expression at steady state remained unaffected by *Tigit* deletion (Extended Data Fig. 4k). To control for the differences in the antiviral immune response elicited in WT mice and mice that lack TIGIT on T_reg cells, we used heterozygous *Foxp3*^GFPcreERT2 × *Tigit*^fl/fl females, which harbor both WT and *Tigit*-KO T_reg cells that can be distinguished based on green fluorescent protein (GFP) expression. Although the WT T_reg cell fraction (Foxp3⁻GFP⁻) efficiently produced Areg, TIGIT-deficient T_reg cells (Foxp3⁻GFP⁺) failed to do so (Fig. 3i and Extended Data Fig. 4l). Along with the reduction in T_reg cell-derived Areg in tamoxifen-treated *Foxp3*^GFPcreERT2 × *Tigit*^fl/fl mice, vascular leakage in the lung and liver was increased compared to in vehicle-treated control mice (Fig. 3j and Extended Data Fig. 4m). Similarly, AST levels were significantly higher when *Tigit* was deleted from T_reg cells (Fig. 3k). Thus, in T_reg cells, TIGIT directly induces Areg in a cell-intrinsic manner following infection, and TIGIT expression on T_reg cells is essential for TIGIT-mediated protection from pathology following infection.

### TIGIT-mediated Areg induction is regulated by CD155 but not CD226

The TIGIT ligand CD155 is shared with its co-stimulatory counterpart CD226, raising the possibility that CD226 might influence Areg expression in the absence of TIGIT. To test this, we infected *Tigit*-KO and WT mice with LCMV and treated them with anti-CD226 or isotype control. CD226 blockade did not alter Areg expression in either WT or *Tigit*-KO T_reg cells, indicating that the reduction in Areg seen in *Tigit*-KO T_reg cells is not mediated by compensatory CD226 signaling, but rather reflects a direct requirement for TIGIT (Fig. 3l). We next examined the expression dynamics of TIGIT, CD226 and their shared ligand CD155 to understand their interplay in this regulatory pathway. TIGIT expression in T_reg cells was unaffected by CD226 blockade (Extended Data Fig. 5a). CD155, which is highly expressed on T cells, was slightly upregulated in WT T cells following CD226 blockade, but was mildly reduced in *Tigit*-KO T_reg cells (Extended Data Fig. 5b–d). By contrast, CD155 levels in myeloid cells remained unchanged between WT and *Tigit*-KO mice (Extended Data Fig. 5e–g). CD226 expression was also mostly unaffected in T cells from both full and conditional *Tigit*-KO mice (Extended Data Fig. 5h,i).

To address whether CD155 is required for Areg induction downstream of TIGIT, we analyzed *Cd155*-KO mice (Extended Data Fig. 5j). After LCMV infection, T_reg cells from *Cd155*-KO mice failed to upregulate Areg to the same extent as WT control mice, despite unaltered TIGIT expression (Fig. 3m and Extended Data Fig. 5k), suggesting that in vivo CD155 engagement promotes TIGIT-driven Areg expression in T_reg cells.

### TIGIT induces Areg in T_reg cells in a TCR-dependent manner

Tissue T_reg cells associated with regenerative properties were found to have an oligoclonal TCR repertoire that shows considerable overlap with its precursor population in the spleen[17]. We thus wondered whether the T cell antigen receptor (TCR) repertoire of TIGIT⁺ T_reg cells that produce Areg in LCMV infection overlaps with this tissue T_reg cell repertoire. We performed single-cell TCR sequencing of WT and *Tigit*-KO T cells from the spleens and lungs of both naive and infected mice (Extended Data Fig. 6a–h). The proportion of expanded clones strongly increased after infection in both WT and KO groups, with an overall higher proportion of clonally expanded T cells in the lung than in the spleen (Fig. 4a–c and Extended Data Fig. 6i–p). In WT mice, the majority of the expanded T_reg cell clones after infection were *Tigit*⁺ (Fig. 4d), which is in line with TIGIT expression marking T_reg cells with a highly activated phenotype[4]. Next, we looked at clonal overlap to determine whether our TCR sequences correspond to those previously reported for tissue T_reg cells. For comparison, we also included TCR repertoires of effector T cells obtained after LCMV infection[17–21]. When comparing clonotypes within our dataset, we found a partial overlap between mice within the same organ, but not necessarily between different organs, and, overall, only very few clones were shared between mice (Extended Data Fig. 7a,b). Most importantly, no overlap was seen between the previously reported tissue T_reg cell TCR repertoire and our dataset, while both CD4⁺Foxp3⁻ and CD8⁺ effector T cells partially overlapped with existing datasets (Fig. 4e). This suggests that the *Tigit*-expressing T_reg cells that expand following LCMV infection are not the classical tissue T_reg cells previously associated with tissue repair. To further pursue this notion, we assessed transcriptional and phenotypic differences between classical tissue T_reg cells and TIGIT⁺ repair T_reg cells and found them to have distinct transcriptional profiles and different cluster composition (Fig. 4f,g and Extended Data Fig. 7c–f). Infection-induced TIGIT⁺ repair T_reg cells also did not express the tissue T_reg cell markers ST2, KLRG1 or GATA3 at the transcriptional or protein level (Fig. 4f,g and Extended Data Fig. 7d–f). Additionally, we observed that LCMV infection broadly shifted the T_reg cell compartment toward a higher frequency of peripherally induced T_reg cells, including both TIGIT⁺ and TIGIT⁻ populations, consistent with previous findings[22] (Extended Data Fig. 7f). These results confirm that TIGIT⁺ repair T_reg cells represent an infection-induced, phenotypically and transcriptionally distinct population versus classical tissue T_reg cells.

When we further analyzed our dataset for clonal expansion based on the transcriptional expression of *Tigit* and *Areg*, we noticed that a higher number of *Tigit*⁺*Areg*⁺ T cells was clonally expanded than

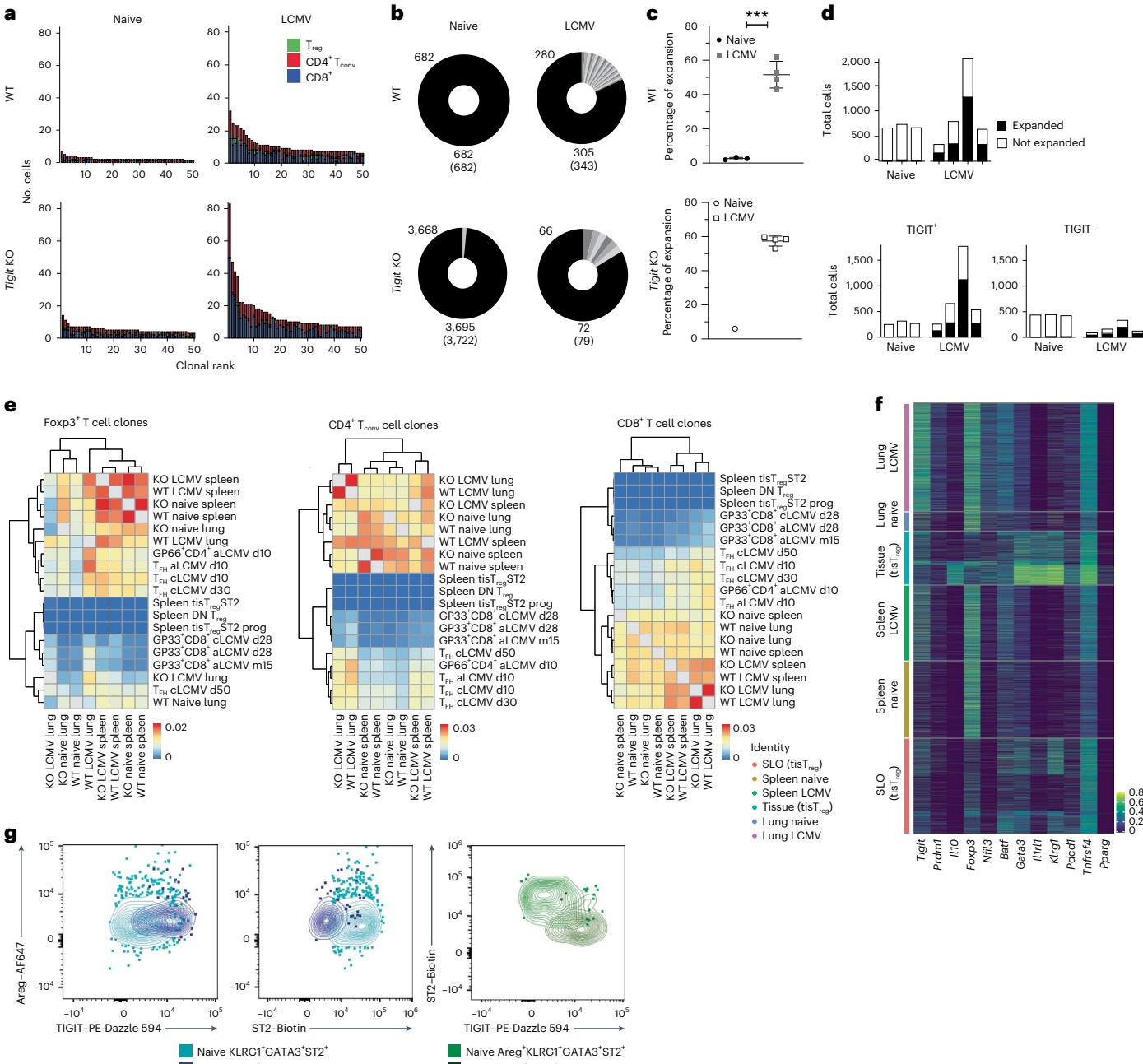

**Fig. 4 | TIGIT⁺ repair T_reg cells are clonally expanded and distinct from classical tissue T_reg cells. a–f,** scRNA-seq/TCR-seq data from lung T cells of naive and LCMV Cl13-infected WT and *Tigit*-KO mice. **a,** Representative expansion plots for one individual mouse per condition in the lung for the 50 most expanded clones per sample, colored based on T cell subset. **b,** Clonal expansion donut plots for Foxp3⁺ lung T cells for one representative mouse per condition. Shades of gray correspond to individual expanded clones, and black represents the proportion of nonexpanded clones. **c.** Frequency of clonal expansion in WT and *Tigit*-KO mice in LCMV-infected lung (data are shown as mean ∓ s.d.; WT *n* = 3 and 4; KO *n* = 1 and 4; data were analyzed by unpaired two-sided *t*-test). **d,** Total, *Tigit*⁺ and

*Tigit*⁻ T_reg cells with expansion information per mouse. **e,** Jaccard indexes of clonal overlap of the VDJ CD3 sequences across different datasets split by T cell subset. Datasets used originated from referenced publications[17–21]; DN, double negative; aLCMV, acute LCMV infection; cLCMV, chronic LCMV infection; tisT_reg, tissue T_reg cells; prog, progenitor; d, day; T_FH, follicular helper T cells; m, month. **f,** Heat map showing tissue repair gene expression in our scRNA-seq data and published ST2⁺ tissue T_reg cell dataset[17]. **g,** Representative FACS plots showing the overlay for the indicated markers between naive tissue T_reg cells (KLRG1⁺GATA3⁺ST2⁺) and TIGIT⁺ T_reg cells induced following LCMV infection.

*Tigit⁻Areg⁺* T cells, which was particularly pronounced after infection (Fig. 5a). We thus wondered whether TCR signaling itself might contribute to Areg induction in TIGIT⁺ T_reg cells. To test whether direct TCR stimulation would lead to increased Areg production, we stimulated splenocytes from naive and LCMV-infected WT mice with titrated amounts of anti-CD3 in vitro. Indeed, Areg levels increased proportionally with increasing TCR stimulation, particularly in T_reg

cells (Fig. 5b and Extended Data Fig. 8a). Importantly, the fraction of T_reg cells that produced Areg following in vitro stimulation with anti-CD3 was characterized by TIGIT expression, while Areg induction was not seen in TIGIT⁻ T_reg cells (Fig. 5c). Most TIGIT⁺Areg⁺ T_reg cells did not stain with LCMV tetramers (Extended data Fig. 8b,c), suggesting that TIGIT⁺Areg⁺ T_reg cells arising during LCMV infection are mostly not virus specific but likely activated by self-antigens derived from

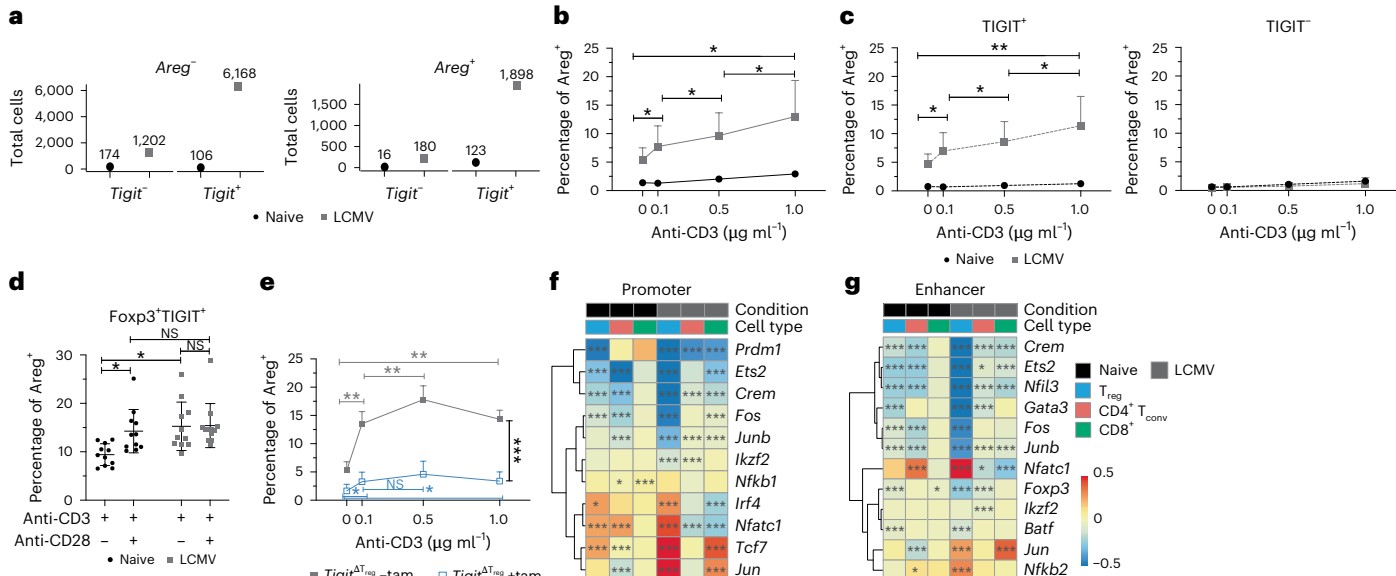

**Fig. 5 | Areg is induced through TIGIT in a TCR-dependent manner. a**, Total number of clonally expanded cells with a specific phenotype in relation to *Tigit* and *Areg* expression split by condition. **b**–**e**, Areg frequencies in splenic T$_{reg}$ cells stimulated with anti-CD3 and anti-CD28 in vitro. Comparison between T$_{reg}$ cells from naive and LCMV Cl13-infected mice among total T$_{reg}$ cells (**b**; data are shown as mean ∓ s.d.; two pooled experiments; data were analyzed by paired two-sided *t*-test) and TIGIT$^+$ or TIGIT$^-$ T$_{reg}$ cells (**c**; data are shown as mean ∓ s.d.; two pooled experiments; data were analyzed by paired two-sided *t*-test). **d**, Areg frequencies in TIGIT$^+$ naive (black) or LCMV (gray) T$_{reg}$ cells stimulated with different combinations of anti-CD3 and anti-CD28 (data are shown as mean ∓ s.d.; three

pooled experiments; data were analyzed by two-way ANOVA with a Tukey's post hoc test). **e**, Total T$_{reg}$ cells from LCMV-infected *Foxp3*$^{GFPcreERT2}$ × *Tigit*$^{fl/fl}$ (*Tigit*$^{△Treg}$) mice treated with tamoxifen (+tam) or with vehicle as a control (−tam; data are shown as mean ∓ s.d.; one representative experiment; data were analyzed by paired two-sided and unpaired two-sided *t*-test for 1 μg ml$^{-1}$ +tam versus −tam comparisons). **f**,**g**, SCENIC analysis of scRNA-seq data from T cells from naive or LCMV Cl13-infected WT and *Tigit*-KO mice. Heat maps show differential regulon activity between WT and *Tigit*-KO T cells for selected transcription factors in promoter (**f**) and enhancer (**g**) regions.

damaged tissue. Interestingly, TCR-dependent induction of Areg was only observed in in vivo-primed T cells, whereas T cells isolated from naive mice did not produce significant levels of Areg following TCR stimulation (Fig. 5b,c), which is in line with previous reports[11]. Thus, we next asked which infection-induced factors might license naive T$_{reg}$ cells to express Areg. Co-stimulation with anti-CD28 in addition to anti-CD3 enabled naive T$_{reg}$ cells to upregulate Areg expression to levels comparable to those observed in infected T$_{reg}$ cells but did not further enhance Areg expression in in vivo-primed T$_{reg}$ cells (Fig. 5d and Extended Data Fig. 8d,e). Additional stimulation with cytokines previously reported to induce Areg expression, such as IL-18, TGFβ or Areg itself[11,23], did not further enhance Areg expression beyond anti-CD3/anti-CD28 stimulation (Extended Data Fig. 8e). These results show that TCR signaling promotes Areg production by TIGIT$^+$ T$_{reg}$ cells in vitro and that co-stimulation is required to license naive T$_{reg}$ cells for Areg expression following TCR stimulation. Importantly, when cells from infected *Foxp3*$^{GFPcreERT2}$ × *Tigit*$^{fl/fl}$ mice were stimulated in the same manner, we found that TCR stimulation was unable to induce Areg expression in *Tigit*-KO T$_{reg}$ cells, highlighting the requirement of both the TCR as well as TIGIT for Areg induction (Fig. 5e). By contrast, CD8$^+$ T cells from infected *Cd8*$^{Cre}$ × *Tigit*$^{fl/fl}$ mice showed comparable Areg production as that observed in cells from WT mice following in vitro stimulation (Extended Data Fig. 8f), confirming that TIGIT on CD8$^+$ T cells is dispensable for Areg production.

To further explore downstream signaling in TIGIT$^+$ T$_{reg}$ cells, we performed single-cell regulatory network inference and clustering (SCENIC)[24] analysis of our scRNA-seq data. Comparison of WT and *Tigit*-KO cells revealed distinct regulons in the two genotypes (Fig. 5f,g and Extended Data Fig. 8g), suggesting they may contribute to TIGIT-dependent functions. In line with TIGIT$^+$ T$_{reg}$ cells being highly suppressive[4], we observed increased scores for the IL-2 inhibitors *Crem* and *Ets2* (refs. [25,26]), as well as *Helios*, *Fos* and *Junb*, which are

important for T$_{reg}$ cell development and stability[27,28], in the presence of TIGIT (Fig. 5f,g). TIGIT$^+$ T$_{reg}$ cells also expressed higher levels of JunB protein (Extended Data Fig. 8h,i). Conversely, and in line with TIGIT's co-inhibitory activity, transcription factors downstream of TCR signaling, such as *Nfatc1*, *Tcf7* and *Jun*, were upregulated in *Tigit*-KO mice (Fig. 5f,g). Focusing on T$_{reg}$ cells, in vitro engagement of TIGIT showed only a minor impact on intermediates of TCR signaling, such as pAKT or pS6 (Extended Data Fig. 8j,k), confirming that TCR signaling in T$_{reg}$ cells remains intact following TIGIT engagement. Together, we show that TIGIT$^+$ T$_{reg}$ cells, distinct from the classical tissue T$_{reg}$ cell population but still expressing *Areg*, are clonally expanded. Areg production by these in vivo-primed, clonally expanded T$_{reg}$ cells was strictly dependent on both TIGIT expression and TCR stimulation.

## TIGIT engagement induces Areg through Blimp-1

To identify candidate transcription factors that could mediate signaling downstream of TIGIT to induce Areg transcription, we looked for transcription factors that were part of our repair signature and co-regulated with *Tigit* and *Areg* in our scRNA-seq dataset. *Hif1a* and *Prdm1* were enriched in the effector T$_{reg}$ cell cluster, in the proliferating and T$_H$1 CD4$^+$ T cell clusters and in the exhausted, effector and proliferating CD8$^+$ T cell clusters, which all showed high *Tigit* expression (Fig. 6a and Extended Data Fig. 2e). HIF1α is upregulated under stressful conditions that cause hypoxia and through STAT3 and TCR signaling[29], whereas *Prdm1*, encoding Blimp-1, is induced by TCR stimulation in conjunction with inflammatory cytokines[30,31]. We observed a significant correlation between *Areg* and both *Hif1a* and *Prdm1* expression after infection, especially in T$_{reg}$ cells, and thus tested their impact on Areg induction following LCMV infection (Fig. 6b and Extended Data Fig. 9a). To determine whether Hif1α can promote Areg production in T cells, we infected *Cd4*$^{Cre}$ × *Hif1a*$^{fl/fl}$ mice or *Hif1a*$^{fl/fl}$ control mice with LCMV. *Cd4*$^{Cre}$ × *Hif1a*$^{fl/fl}$ mice failed to induce Areg expression to WT levels

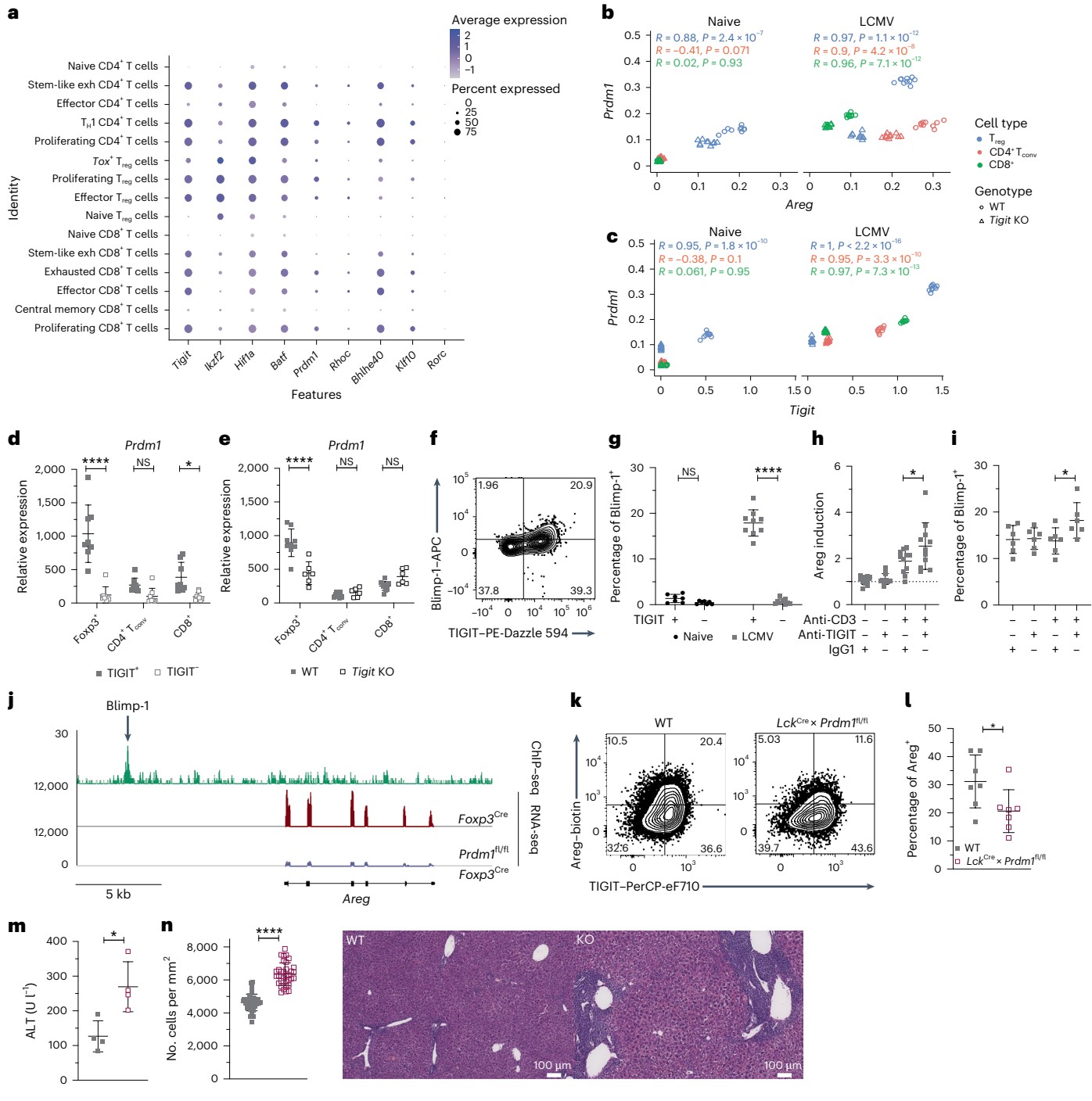

**Fig. 6 | TIGIT induces Areg through Blimp-1. a–c**, scRNA-seq data.
**a**, Transcription factor expression within repair signature genes and *Tigit*; exh, exhausted. **b,c**, Scatter plots of *Prdm1* and *Areg* (**b**) or *Tigit* (**c**) expression (two-sided, *P* value derived from a Pearson correlation). **d,e**, *Prdm1* mRNA expression in sorted $T_{reg}$ cells, CD4$^+$ $T_{conv}$ cells and CD8$^+$ T cells from LCMV-infected mice (data are shown as mean ∓ s.d.; pool of two independent experiments analyzed by two-way ANOVA with a Šídák's post hoc test (**d**) or pool of three independent experiments analyzed by two-way ANOVA with a Šídák's post hoc test (**e**)). **f,g**, Representative plot of splenic $T_{reg}$ cells after LCMV infection (**f**) and summary plots of TIGIT$^+$ and TIGIT$^-$ splenic $T_{reg}$ cells from naive and LCMV-infected mice (**g**; data are shown as mean ∓ s.d.; pool of two independent experiments; data were analyzed by two-way ANOVA with a Tukey's post hoc test). **h,i**, Splenic $T_{reg}$ cells were stimulated in vitro with anti-CD3, agonistic anti-TIGIT or IgG1, and Areg induction (**h**; normalized to unstimulated) and Blimp-1 frequencies (**i**) were determined (data in **h** are shown as mean ∓ s.d., pool of three independent

experiments, data were analyzed by paired two-sided *t*-test; data in **i** are shown as mean ∓ s.d., one representative experiment, data were analyzed by paired two-sided *t*-test). **j**, Blimp-1 ChIP–seq track and RNA-seq track on the *Areg* locus showing the Blimp-1 binding site and *Areg* transcription in WT and *Prdm1*-KO $T_{reg}$ cells. **k,l**, Representative plots (**k**) and summary (**l**) of Areg and TIGIT expression in splenic $T_{reg}$ cells from LCMV-infected WT and $Lck^{Cre} \times Prdm1^{fl/fl}$ mice (day 14; data are shown as mean ∓ s.d.; pool of two independent experiments; data were analyzed by unpaired two-sided *t*-test). **m**, ALT levels of LCMV-infected (day 14) WT and $Lck^{Cre} \times Prdm1^{fl/fl}$ mice (data are shown as mean ∓ s.d.; pooled data from one to two experiments; data were analyzed by unpaired two-sided *t*-test, *n* = 4). **n**, Cellular infiltrate in liver from LCMV-infected (day 14) WT and $Lck^{Cre} \times Prdm1^{fl/fl}$ mice (data are shown as mean ∓ s.d; one representative experiment; selected ROI, *n* = 38 and 34; biological replicates, *n* = 3–4; data were analyzed by unpaired two-sided *t*-test) and representative hematoxylin and eosin (H&E) staining.

in any of the T cell populations addressed (Extended Data Fig. 9b,c), demonstrating that HIF1α is essential for Areg production by T cells. However, the absence of TIGIT did not affect HIF1α expression as WT and *Tigit*-KO T cells showed comparable *Hif1a* transcript expression both at the single-cell level and by quantitative real-time PCR (qPCR) of sorted T cell populations (Extended Data Fig. 9a,d,e). Hence, the impact of HIF1α on Areg expression is independent of TIGIT. We next focused on Blimp-1 (encoded by *Prdm1*), which also showed reduced regulon activity in *Tigit*-KO cells (Fig. 5f), as a potential downstream mediator of TIGIT-dependent induction of Areg. We observed a substantial correlation between *Tigit* and *Prdm1* expression in the scRNA-seq data (Fig. 6c), with a strong *Prdm1* downregulation in *Tigit*-KO T cells, particularly in $T_{reg}$ cells and CD8⁺ T cells from infected mice. We thus measured *Prdm1* transcript expression in sorted TIGIT⁺ versus TIGIT⁻ T cells and detected higher *Prdm1* mRNA expression in the TIGIT⁺ T cell subpopulations (Fig. 6d and Extended Data Fig. 9f). We further validated this dependence by performing qPCR on sorted T cells from WT and *Tigit*-KO mice, confirming reduced *Prdm1* transcription in *Tigit*-KO $T_{reg}$ cells (Fig. 6e and Extended Data Fig. 9g). TIGIT and Blimp-1 expression also correlated at the protein level, where Blimp-1 was exclusively expressed in TIGIT⁺ $T_{reg}$ cells and induced only after infection with LCMV or influenza (Fig. 6f,g and Extended Data Fig. 9h,i). Moreover, Blimp-1 was downregulated in vivo in the absence of TIGIT engagement in *Cd155*-KO mice (Extended Data Fig. 9j). Furthermore, Blimp-1 downregulation in *Tigit*-KO mice was a direct effect of *Tigit* deletion, as its expression was not further altered by CD226 blockade (Extended Data Fig. 9k). Finally, to provide a functional link between TIGIT, Blimp-1 and Areg, we stimulated splenocytes with TIGIT and TCR agonists in vitro and found that the combination of TIGIT plus TCR engagement not only resulted in Areg induction by $T_{reg}$ cells but also increased Blimp-1 expression (Fig. 6h,i). Hence, TIGIT directly induces Blimp-1 in activated $T_{reg}$ cells.

To determine whether Blimp-1 can regulate *Areg* transcription, we made use of our Blimp-1 chromatin immunoprecipitation with sequencing (ChIP–seq) data[32] and found that Blimp-1 directly binds upstream of the *Areg* promoter and also regulates its expression in tissue $T_{reg}$ cells[31] (Fig. 6j). Furthermore, *Areg* transcription was downregulated in *Prdm1*-KO $T_{reg}$ cells isolated from *Lck*^Cre × *Prdm1*^fl/fl mice (Fig. 6j). Most importantly, the induction of Areg protein following LCMV infection was impaired in *Lck*^Cre × *Prdm1*^fl/fl mice, confirming their functional dependence on the protein level (Fig. 6k,l). Moreover, ALT levels and cellular infiltrates, both characteristic for liver pathology, were increased in *Lck*^Cre × *Prdm1*^fl/fl mice compared to in WT control mice, confirming an involvement of Blimp-1 in the repair program we identified (Fig. 6m,n). Thus, our data show that the Areg-dependent tissue repair function of $T_{reg}$ cells is regulated by both HIF1α and Blimp-1. We further uncover a signaling axis in which the engagement of TIGIT and TCR induces the transcription factor Blimp-1, which then directly drives *Areg* transcription to facilitate tissue repair following infection.

## Discussion

Co-inhibitory receptors are essential for ensuring a controlled immune response during infection, preventing immune-mediated tissue damage. Without this regulation, collateral damage can lead to immune pathology as seen in infections like SARS-CoV-2 and in individuals with cancer receiving checkpoint therapy, where immune-related tissue damage can occur following blockade. In this study, we demonstrate that the co-inhibitory receptor TIGIT not only restrains effector immune responses but also actively promotes tissue repair. Specifically, we show that TIGIT induces Areg in a Blimp-1-dependent manner, activating a tissue repair program in $T_{reg}$ cells that is essential for limiting pathology during infection.

We have previously shown that TIGIT plays a critical role in limiting immune pathology during LCMV infection by shifting the cytokine balance and decreasing tissue damage[7]. Consistent with these findings, we observed increased systemic pathology in *Tigit*-KO mice, with no major

changes in T cell activation and immune cell composition. Instead, we identify a transcriptional tissue repair signature in TIGIT-expressing T cells, which is strongly impaired following TIGIT loss, revealing an additional protective function of TIGIT in promoting tissue repair and independent from its immune-suppressive capacity.

Over the past decade, growing evidence has highlighted a key role for T cells, particularly $T_{reg}$ cells, in tissue repair[10,11,33]. At steady state, tissue-resident $T_{reg}$ cells promote homeostasis and regeneration, maintaining tissue integrity and facilitating repair after sterile injuries, such as muscle damage and stroke[8,12]. They also promote metabolic homeostasis, where $T_{reg}$ cells in visceral adipose tissue influence adipogenesis and insulin sensitivity[34,35]. These tissue $T_{reg}$ cells are characterized by IL-33 receptor (ST2) expression and Areg production in response to soluble mediators like IL-33, adipokines and metabolites, enabling tissue–immune cross-talk. However, the mechanisms driving Areg induction in $T_{reg}$ cells remain poorly understood. Cytokines like IL-18 and IL-33 can promote Areg expression in $T_{reg}$ cells independently of TCR signaling and unrelated to their suppressive function[11]. In line with these findings, we did not observe Areg induction in steady-state $T_{reg}$ cells without additional co-stimulatory signals. However, we found that during an active immune response, such as the one induced following an infection, Areg production by in vivo-primed $T_{reg}$ cells is driven by combined signals from the TCR and TIGIT. Additionally, we observed clonal expansion of Areg-expressing TIGIT⁺ $T_{reg}$ cells during infection, emphasizing the antigen-driven expansion of this tissue repair $T_{reg}$ cell subset. As the Areg-secreting TIGIT⁺ $T_{reg}$ cells are not LCMV-specific, this is likely driven by self-antigens released from damaged tissues following infection. Notably, the TCR repertoire of Areg-producing TIGIT⁺ $T_{reg}$ cells that promote tissue repair in this context differed from that of ST2⁺ tissue $T_{reg}$ cells, as did their transcriptional and proteomic profiles, clearly distinguishing them from this previously described subset[17,36]. Furthermore, TIGIT blockade or deletion severely impaired Areg expression and reduced the entire transcriptional repair signature in T cells, highlighting the importance of TIGIT in tissue repair following infection. Differences in the magnitude of Areg induction between the LCMV and influenza infection models are likely due to different pathology and extent of tissue damage. Although LCMV is a noncytolytic virus, the early phase of Cl13 infection is dominated by extensive immune-mediated pathology, whereas the influenza infection model used here caused limited pathology due to the low infectious dose used. Consistent with this, we also observed that the distinct pathological profiles of LCMV WE and Cl13 infections translated into differences in the magnitude of Areg induction, further supporting the notion that the extent of tissue damage is a key determinant of the Areg⁺ $T_{reg}$ cell response. Interestingly, TCR-dependent Areg expression has also been reported in human T cells, which are unresponsive to IL-18 or IL-33 stimulation, as well as in mouse models of ischemic stroke[33]. Similarly, mucosal-associated invariant T cells upregulate Areg expression in a TCR-dependent manner during liver fibrosis. A recent study identified human PD-1⁺TIGIT⁺CD8⁺ T cells that express Areg, suggesting a role in tissue remodeling[37]. In our model, TIGIT expression in CD8⁺ T cells was not required for Areg production nor to limit pathology, suggesting distinct regulatory mechanisms for tissue repair by CD8⁺ T cells and $T_{reg}$ cells. Together, these findings indicate that Areg induction in $T_{reg}$ cells is highly context dependent, with combined TCR and TIGIT signaling being critical during infection.

Despite the progress in understanding the signals that induce Areg, the transcription factors regulating its expression remain largely unknown. Here, we identified HIF1α and Blimp-1 as key regulators of Areg in $T_{reg}$ cells during infection, with HIF1α acting independently of TIGIT. Interestingly, TIGIT and HIF1α may act synergistically to inhibit tumor growth in preclinical cancer models[38], but modulation of Areg expression was not assessed in this study.

The role of Blimp-1 in immune regulation and T cell differentiation is well established[39,40]. Blimp-1, known to be induced by TCR signaling

and IL-2 (ref. 41), is crucial for $T_{reg}$ cell stability in inflamed tissues[42] and marks highly suppressive effector $T_{reg}$ cells, playing an essential role in IL-10 production[43,44]. Blimp-1 is closely associated with effector $T_{reg}$ cell differentiation and, together with c-Maf, co-regulates IL-10 expression in effector T cells[43,45]. Here, we show that Blimp-1 drives the expression of Areg in $T_{reg}$ cells, with its induction being dependent on combined signals from the TCR and TIGIT. Notably, TIGIT is also expressed on highly suppressive $T_{reg}$ cells and directly promotes IL-10 production[4]. This raises the possibility that TIGIT may enhance the suppressive function of $T_{reg}$ cells and their IL-10 production through Blimp-1 induction, a hypothesis that requires further investigation.

The primary function of most co-inhibitory receptors is to limit immune activation, as seen in alterations in and KO of *Ctla4*, which leads to severe autoimmunity[46,47], and *Pdcd1*, which is crucial for controlling the activation of autoreactive T cells in peripheral tissues[48,49]. Although TIGIT also dampens T cell activation, it is less potent in this regard than other co-inhibitory receptors[5]. However, TIGIT stands out for its ability to shift the cytokine environment toward a $T_H2$-like response[4,50], a profile traditionally linked to tissue repair. In this study, we further demonstrate that TIGIT induces the expression of Areg, a key mediator of tissue repair. These findings could potentially serve as a basis for new therapeutic approaches for fibrotic disorders or chronic wounds where targeting TIGIT may promote and improve controlled tissue repair. Thus, beyond merely suppressing immune responses, TIGIT plays a dominant role in actively promoting tissue regeneration. Our findings expand the understanding of co-inhibitory receptors, revealing their capacity to not only limit immune activation but also actively contribute to tissue repair processes.

## Online content

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

## Methods

### Mice

Experiments were conducted according to the institutional policies and national regulations and have been reviewed and approved by the Cantonal Veterinary Office and the University of Melbourne Animal Ethics Committee. B6 mice were purchased from Janvier Labs. *Foxp3*–GFP.KI and *Tigit*fl/fl mice have been described previously[51,52] (from V. K. Kuchroo, Gene Lay Institute of Immunology and Inflammation). B6.129(Cg)-*Foxp3*tm4(YFP/icre)Ayr/J (*Foxp3-cre*), *Foxp3*tm9(EGFP/cre/ERT2)Ayr/J[53] (*Foxp3*creERT2) and B6-Tg(*Cd8a-cre*)1Itan/J (*Cd8*Cre) strains were obtained from The Jackson Laboratory and crossed with the *Tigit*fl/fl strain. B6.Cg-Tg(*Cd4-cre*)1Cwi/BfluJ (*Cd4*Cre) and B6.129-*Hif1a*tm3Rsjo/J (*Hif1a*fl/fl) mice were provided by the laboratory of N. Aceto (Swiss Federal Institute of Technology Zurich). B6N.129S2-*Pvr*tm1Gbn/J (*Cd155*-KO) mice[54] were kindly provided by the laboratory of T. Korn (Technical University of Munich). *Prdm1*fl/fl mice were previously described[40]. All mice were on a B6 background. Adult 7- to 45-week-old male and female mice were used, age and sex were matched between experimental groups, and animals with the same genotype were randomized to the different treatment groups. *Lck*Cre × *Prdm1*fl/fl mice were bred and housed at the Peter Doherty Institute, Melbourne, Australia, and all other strains were located at the University of Zurich or the University of Basel, Switzerland, under specific pathogen-free conditions. Animals were housed in individually ventilated cages containing autoclaved bedding and nesting material, with standard diet and water ad libitum on a 12-h light/12-h dark cycle (18–23 °C, 40–60% humidity).

### In vivo treatments and infections

To induce Cre-dependent KO, *Foxp3*tm9(EGFP/cre/ERT2)Ayr/J[53] (*Foxp3*creERT2) × *Tigit*fl/fl mice received intraperitoneal (i.p.) injections of tamoxifen (2.5 mg in oil) or vehicle as a control three times per week throughout the course of infection starting 1 week before infection. For TIGIT blockade, B6 mice were injected i.p. with 100 µg of anti-TIGIT (1B4)[16] or IgG1 isotype control (BioXCell) on days −2, 0, 2, 5 and 8 of infection. For CD226 blockade, B6 WT or *Tigit*-KO mice were injected i.p. with 100 µg of anti-CD226 (10E5) or IgG2b isotype control.

For LCMV infections, virus was produced as previously described[7], and mice were infected intravenously (i.v.) with either $2 × 10^6$ focus-forming units of LCMV Cl13 or 200 focus-forming units of LCMV WE. To determine vascular integrity, mice were injected i.v. with 200 µl of 0.5% Evans blue dye (Sigma) in sterile PBS 20 min before euthanasia. Animals were perfused with PBS, organ weights were measured, and organs were placed in 1 ml of formamide (Sigma) and incubated at 56 °C overnight. The amount of extracted Evans blue dye was measured by photometry at 620 nm and quantified against a standard curve using the following formula: (Evans blue concentration (ng ml$^{-1}$) × volume of formamide (ml)) / total organ weight (g). Serum AST and ALT were measured at the Clinical Chemistry Department of the University Hospital Zurich by determining the catalytic concentration of both enzymes at 37 °C with pyridoxal phosphate activation on a Roche Cobas 8000 (c502) according to International Federation of Clinical Chemistry and Laboratory Medicine.

### Immune cell isolation

Single-cell suspensions of spleen tissue were generated by mechanical disruption in RPMI 1640 (Gibco) supplemented with 10% fetal calf serum (Corning or Capri), penicillin and streptomycin (100 U ml$^{-1}$; Gibco) and 2 mmol l$^{-1}$ L-glutamine (Gibco). Red blood cells were lysed using ACK lysis buffer (155 mM NH$_4$Cl, 10 mM KHCO$_3$ and 0.1 mM Na$_2$EDTA, pH 7.4) for 3–5 min. Single-cell suspensions of lung tissue were generated by enzymatic tissue digestion at 37 °C with 0.2 mg ml$^{-1}$ deoxyribonuclease I (Sigma) and 2.4 mg ml$^{-1}$ collagenase type I (Gibco) and mechanical disruption using a GentleMACS. Lung immune cells were separated from epithelial and parenchymal cells by 30% Percoll gradient centrifugation (GE Healthcare).

### Flow cytometry

Single-cell suspensions were obtained from spleen and lung tissue processed in RPMI 1640 (Gibco) supplemented with Brefeldin A (BioLegend) and marimastat (Sigma) and directly stained ex vivo or incubated at 37 °C for 2 h and 45 min with Brefeldin A (BioLegend), monensin (BioLegend) and marimastat (Sigma). Samples were incubated with surface antibodies diluted in PBS for 20 min at 22 °C. For intracellular cytokine staining, cells were permeabilized using a Cytofix/Cytoperm kit (BD Bioscience) for 5–10 min at 22 °C, followed by a 30-min incubation at 22 °C with a mix of intracellular antibodies. For transcription factor staining, cells were permeabilized with a Foxp3/Transcription factor staining buffer set (eBioscience) for 40 min at 22 °C. The Zombie NIR and LD Blue fixable dyes were used to exclude dead cells. Data were acquired on a BD LSR Fortessa, BD FACS Symphony A5 (BD Bioscience) or Cytek Aurora (Cytek Biosciences) and analyzed using SpectroFlo (Cytek Biosciences, v3.0.3) and FlowJo (BD Bioscience, v10.8.2 or 10.10.0) software. For R analysis, unmixed and pregated cells were imported into RStudio using R (v4.4.1) and the flowCore (v2.16.0) and CATALYST (v1.28.0) packages[55,56]. Data were transformed using a hyperbolic arcsine transformation. This was followed by dimensionality reduction applying UMAP and FlowSOM algorithms. Clusters were annotated by overlaying the FlowJo workspace using the flowWorkspace (v4.16.0) and CytoML (v2.16.0) packages[57,58]. Heat maps and stacked population plots were generated based on the frequency of respective marker expression and visualized using the ggplot2 (v3.5.1) package in R.

The following anti-mouse monoclonal antibodies were used: Areg, Alexa Fluor 647, 1:100, G-4, Santa Cruz; Areg, Biotin, 1:200, R&D; B220, BUV563, 1:200, RA3-6B2, BD; Blimp-1, APC, 1:100, 5E7, BioLegend; Blimp-1, BV421, 1:100, 5E7, BD; CD112, BV711, 1:100, 829038, BD; CD11b, PerCP, 1:400, M1/70, BioLegend; CD11c, BUV496, 1:100, HL3, BD; CD11c, BV605, 1:100, N418, BioLegend; CD134, BV421, 1:100, OX-86, BioLegend; CD155, BUV737, 1:100, TX56, BD; CD155, PE, 1:300, TX56, BioLegend; CD226, APC/Fire750, 1:100, 10E5, BioLegend; CD226, PE-Cy7, 1:100, 10E5, BD; CD3, BUV805, 1:100, 17A2, eBioscience; CD3, BV785, 1:200, 17A2, BioLegend; CD4, BUV737, 1:300, RM4-5, BD; CD4, APC, 1:200, RM4-5, BioLegend; CD4, BUV496, 1:500, RM4-5, BD; CD4, APC-Cy7, 1:300, GK1.5, BD; CD44, APC/Fire750, 1:300, IM7, BioLegend; CD45, BUV805, 1:300, 30-F11, BD; CD62L, BV650, 1:400, MEL-14, BioLegend; CD8, BUV395, 1:500, 53-6.7, BD; CXCR3, Biotin, 1:100, SA051D1, BioLegend; F4/80, APC-Fire810, 1:100, BM8, BioLegend; Foxp3, FITC, 1:200, FJK-16s, eBioscience; Foxp3, PE, 1:200, FJK-16s, Thermo; Foxp3, eF450, 1:200, FJK-16s, eBioscience; GATA3, PE-Cy5, 1:200, TWAJ, eBioscience; GM-CSF, PE-Cy7, 1:100, MP1-22E9, BioLegend; granzyme B, APC, 1:100, GB12, Thermo; GFP tag, Alexa Fluor 488, 1:400, FM264G, BioLegend; Helios, BUV563, 1:100, 22F6, eBioscience; IFNγ, PE, 1:300, XMG1.2, BioLegend; JunB, CoraLitePlus488, 1:125, ProteinTech; Ki-67, BUV737, 1:300, SolA15, Thermo; KLRG1, BUV661, 1:200, 2F1, BD; LD, Zombie NIR, 1:500, BioLegend; LD, Blue, 1:500, Thermo; Ly6G, BV750, 1:200, 1A8, BD; NK1.1, BUV615, 1:200, PK136, BD; NRP-1, BV711, 1:500, V46-1954, BD; pAKT, Alexa Fluor 647, 1:10, M89-61, BD; PD-1, BV785, 1:500, 29F.1A12, BioLegend; pS6, PerCP-eF710, 1:10, cupk43k, eBioscience; ST2, Biotin, 1:100, DIH9, BioLegend; streptavidin, BUV563, 1:800, eBioscience; streptavidin, BV711, 1:300, BioLegend; streptavidin, BV480, 1:500, BD; TCF1/TCF7, R718, 1:100, S33-966, BD; TCRβ, PE-Dazzle594, 1:300, H57-597, BioLegend; TCRγδ, BB700, 1:200, GL3, BD; TIGIT, PE-Dazzle594, 1:50, 1G9, BioLegend; TIGIT, BV421, 1:50, TX99, BD; TIGIT, PerCP-eF710, 1:200, GIGD7, Thermo; TNF, BV421, 1:100, MP6-XT22, BioLegend. PE-labelled tetramers for LCMV GP$_{66-77}$, LCMV GP$_{126-140}$, LCMV NP$_{309-328}$ were used at 1:100 and provided by the NIH Tetramer Core Facility (NIH Contract 75N93020D00005 and RRID:SCR_026557).

### Immunohistochemistry

Livers were perfused at the moment the mice were killed, and lobes were collected and fixed overnight at 4 °C in 4% formalin.

The tissues were subsequently processed, embedded in paraffin blocks and sectioned for H&E staining and caspase-3 immunohistochemistry. Primary anti-cleaved caspase-3 (polyclonal rabbit, Cell Signaling Technology, clone Asp175, 9661) was applied to the tissue sections at a 1:500 dilution and incubated for 2 h at 22 °C to ensure optimal binding. After primary antibody incubation, sections were treated with horseradish peroxidase-conjugated anti-rabbit secondary antibodies for 30 min at 22 °C. The antigen–antibody complex was visualized using 3,3',5,5'-tetramethylbenzidine as a substrate. Images were captured using the Axio Scan Z.1 (Zeiss) and analyzed using ImageScope software (Leica Biosystem), following the manufacturer's instructions.

For image quantification, QuPath software was used to perform automated analysis of tissue samples, ensuring consistent and reliable cell counting across all ROIs. For caspase-3 analysis, multiple ROIs were examined, with cell boundaries delineated using hematoxylin counterstaining. Caspase-3$^+$ hepatocytes were automatically identified based on a color threshold. The data were expressed as the percentage of caspase-3$^+$ hepatocytes per liver sample. Total cellularity for each sample was calculated using the same ROI-based approach, with QuPath's segmentation algorithm facilitating the automated cell counts. Analysis was performed blind to the conditions of the experiments.

## In vitro stimulation
Isolated splenocytes were stimulated at 37 °C in 10% $CO_2$ for 2 h and 45 min with titrated amounts of anti-CD3 (clone 145-2C11, BioXCell), anti-CD28 (clone PV-1, BioXCell) and 50 ng ml$^{-1}$ agonistic anti-TIGIT (4D4) or corresponding IgG1 antibody in the presence of Brefeldin A (BioLegend), monensin (BioLegend) and marimastat (Sigma). Where indicated, IL-18 (10 ng ml$^{-1}$; BioLegend), TGFβ (10 ng ml$^{-1}$; BioLegend) or Areg (10 ng ml$^{-1}$; BioLegend) were added to the culture.

## scRNA-seq and TCR-seq
One day before LCMV infection or killing of naive mice, animals were adoptively transferred with MACS-purified LCMV-specific or nonspecific CD4$^+$ T cells (Smarta-1 or OTII, $2.5 \times 10^3$ and $10^5$ cells i.v., respectively) and CD8$^+$ T cells (P14 or OTI, $5 \times 10^3$ and $2 \times 10^5$ cells i.v., respectively). Spleen and lung single-cell suspensions were obtained from naive and LCMV-infected (day 10) WT and *Tigit*-KO mice. CD3$^+$ T cells were purified by MACS using a CD3 Cell Isolation kit (negative selection, BioLegend) and then sorted by FACS using a BD FACSAria III 5L (BD Bioscience). Sorted CD3$^+$CD4$^+$Foxp3$^+$ and CD3$^+$Foxp3$^-$ cells were mixed at a ratio of approximately 1:3 and labeled before pooling with anti-mouse Hashtag reagents (TotalSeq-C301, 302, 303, 304, 305 anti-mouse Hashtag 1, 2, 3, 4, 5, respectively, BioLegend). Library preparation was performed by the Functional Genomic Center Zurich using 10x Chromium Next GEM Single Cell 5' Reagent kits, according to manufacturer's instructions (10x Genomics). scRNA-seq/TCR-seq libraries were sequenced on an Illumina Novaseq 6000 S4 (Illumina).

## scRNA-seq analysis
**Processing.** In total, three to five 10x runs were generated per condition. For each sample, we generated the digital gene expression matrix from the raw reads using Cell Ranger 7.1.0 (ref. 59). The unfiltered dataset contained a total of 261,094 cells. Using the digital gene expression matrix from each sample separately, DoubletFinder was applied to filter doublet cells[60], and subsequent filters were used to remove cells with high mitochondria counts and low RNA content. After quality control, the datasets were merged to a dataset consisting of 198,645 cells with a median of 8,116 reads per cell using Seurat v5.0.1 (ref. 61). Count normalization was performed using Seurat's NormalizeData routine to normalize each cell to 10,000 counts and log scale. Highly variable features within normalized counts were selected using the mvp method, which identified 596 highly variable features. Using the highly variable features, scaling of the genes to unit variance was

performed using Seurat's ScaleData routine; dimensionality reduction was performed using principal component analysis. Subsequent batch correction was performed using Harmony v1.2 (ref. 62) and the 10x run identifier. Louvain clustering as implemented through Seurat's FindClusters() outline was performed with a resolution of 0.5, yielding 15 clusters. Clusters that contained a mixture of *Cd4*$^+$ and *Cd8a*$^+$ cells were subsequently subclustered. Repair scores[12,63,64] were assigned by first normalizing the genes of the repair signature to unit variance without zero centering using Seurat's ScaleData. Subsequently, a repair score was assigned to all single cells by summing the individual scores of each gene.

**Data visualization.** Seurat's built-in functionality was used for UMAPs and dot plots. Gene expression UMAP heat maps were generated setting the order parameter to true. Blend UMAPs were generated using min.cutoff = 'q01' and max.cutoff = 'q99'. For the scatter plots, each genotype and condition (naive and LCMV) were randomly split into ten technical replicates (cell pools), and the average normalized expression value (log$_{10}$ cpk (process capability index)) was calculated. Each technical replicate was split according to T cell subset. Correlation coefficients were calculated per cell type and condition. Visualizations were performed with ggplot2, and Pearson *r* correlations were obtained from ggpubr. For repair signature gene correlation heat maps, each value was calculated using single-cell expression values from the subsequent major cell type (T$_{reg}$ and CD4$^+$ and CD8$^+$ T$_{conv}$ cells).

**Integration with the Delacher et al. T$_{reg}$ cell dataset.** T$_{reg}$ cells from our filtered dataset (19,100 cells) were integrated with the Delacher et al.[36] dataset (10,233 cells) using the Harmony method[62] (v1.2; covariate and sample ID information was used to correct for batch effects). Seurat's Louvain clustering routine using a resolution of 0.5 was used to calculate clusters of the integrated dataset. To visualize Delacher T$_{reg}$ cells were grouped into tissue (fat, liver, lung and skin) and secondary and lymphoid organs (blood, bone marrow, lymph node and spleen).

**Identification of statistically significant regulons.** To identify regulons that are associated with TIGIT, randomized subsets of 5,000 T$_{reg}$ and CD8$^+$ T cells across infection and genotype conditions were generated. Regulons were generated using the SCENIC method[24] with default parameters for promoter and enhancer regions. T$_{reg}$, T$_{conv}$ and CD8$^+$ cells were projected onto the identified regulons using SCENIC's AUCell method. Regulons with more than 70% zeros in all conditions according to their AUCell were removed, and permutation tests ($n = 10,000$) were performed for each of the cell types and conditions to identify statistically enriched regulons. After calculating the double-sided *P* values, multiple hypothesis testing corrections were performed using the Holm–Bonferroni method. Effect size is shown as the log (fold change) between WT and KO.

## Single-cell TCR-seq analysis
Raw sequencing files arising from Illumina sequencing lanes were supplied as input to the command line program Cell Ranger (v7.0.0) on a high-performance cluster. Raw reads were aligned to the mouse reference genome (mm10) using 10x Genomics Cell Ranger and subsequently supplied into the VDJ_GEX_matrix function of the R package Platypus (v3.1)[65], which relies on the R package Seurat (v4.0.3)[66] and generates an integrated transcriptome and repertoire data object. Annotations from GEX were transferred to VDJ and vice versa by matching cellular 10x barcodes. Cells with more than 7% mitochondrial reads were filtered out, and gene expression was log normalized with a scaling factor of 10,000. UMAP was calculated, and feature plots were produced using the FeaturePlot function from Seurat. Cells were assembled into clones based on the default enclone clonotyping strategy within Cell Ranger. Smarta-1 and P14 TCR sequences were excluded from the repertoire analysis. Clonal frequency was determined by

counting the number of distinct cell barcodes for each unique clonotype. Those cells in clones supported by only one cell were considered unexpanded clones, whereas those clones supported by two or more cells were considered expanded. Clonal expansion plots were created using the functions VDJ_clonal_expansion and VDJ_clonal_donut from Platypus[67]. Expansion profiles of different T cell subsets were categorized based on the identification of the corresponding Seurat clusters using transcriptional information ($Cd4^+Foxp3^+$, $Cd4^+Foxp3^-$ and $Cd8a^+$). All three T cell subsets were further subdivided into $Tigit^{+/-}$ and $Areg^{+/-}$ groups based on expression of the respective genes. Overlap matrices were calculated by quantifying the exact CDRb3 amino acid matches across repertoires. No filtering regarding the number of TRB and TRA chains was performed for the repertoire data, with the exception of the repertoire overlap comparison with the public datasets where those clones not containing exactly one TRA and one TRB chain were removed from the analysis. Jaccard indices were calculated by quantifying the intersection between two groups divided by the length of the union of the same groups. Heat maps displaying clonal overlap were produced using the pheatmap function in the pheatmap package (v1.0.12). Figures were imported from R (v4.3.2) and further processed using Adobe illustrator.

## qPCR
RNA was extracted using an RNeasy kit (Qiagen) and reverse transcribed to cDNA using a High-Capacity cDNA RT kit (Life Technologies) according to the manufacturer's instructions. qPCR was performed with a QuantStudio 5 Real-Time PCR system (384-well; Thermo). The following primer–probe mixtures were purchased from Applied Biosystems: $Hif1a$ (Mm00468869_m1), $Prdm1$ (Mm00476128_m1), $Areg$ (Mm01354339_m1), $Il10$ (Mm01288386_m1), $Tgfb1$ (Mm01178820_m1), $Batf$ (Mm00479410_m1) and $Nfil3$ (Mm00600292_s1). For $Tigit$, the following primers and probe were used: 5′-CTGATACAGGCTGCCTTCCT-3′ (forward primer), 5′-TGGGTCACTTCAGCTGTGTC-3′ (reverse primer) and 5′-AGGAGCCACAGCAGGCACGA-3′ (probe; FAM, TAMRA). $Actb$ (Thermo) was used as a housekeeping gene (VIC).

## Blimp-1 ChIP–seq and RNA-seq
Blimp-1 ChIP–seq and RNA-Seq data were previously reported and analyzed for the $Areg$ locus as previously described[32]. Briefly, Blimp-1 ChIP–seq reads were mapped to the mouse reference genome, and Blimp-1 binding peaks were called and assigned to their nearest gene. RNA-seq data were generated from $Prdm1^{fl/fl}Foxp3^{Cre}$ and control $T_{reg}$ cells, and reads were aligned to the mouse reference genome and assigned to mouse genes.

## Statistical analysis
Statistical analyses were performed using GraphPad Prism (GraphPad Software, v10.0.3). Appropriate statistical tests were selected as indicated in the figure legends with significant and relevant nonsignificant differences marked in the figures. Data collection and analysis were not blinded due to technical limitations, with the exception of the histological analysis. No data points were excluded, except when technical issues relating to the execution of the experiment or data acquisition were detected. Sample sizes were chosen based on power calculations using estimated effect size ranges that were based on previous publications[7,16,50]. Data distribution was assumed to be normal, but this was not formally tested. To compare more than two groups, ordinary one-way or two-way ANOVA with a Tukey's or Šídák's multiple comparisons test was used. Significance was defined as *$P \leq 0.05$, **$P \leq 0.01$, ***$P \leq 0.001$, ****$P \leq 0.0001$ or NS (not significant; $P > 0.05$).

## Reporting summary
Further information on research design is available in the Nature Portfolio Reporting Summary linked to this article.

## Data availability
All scRNA-seq and single-cell TCR-seq data from WT and $Tigit$-KO T cells have been deposited on the ArrayExpress database at EMBL-EBI (www.ebi.ac.uk/arrayexpress) and is available via accession number E-MTAB-8861. The Seurat object is available at Zenodo at https://doi.org/10.5281/zenodo.14041419 (ref. 68). Blimp-1 ChIP–seq and RNA-seq data were previously reported and are available in the Gene Expression Omnibus database under accession numbers GSE79339 and GSE121838. Source data are provided with this paper.

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

## Acknowledgements

We would like to thank the members of the laboratories of N.J. and A. Oxenius for helpful discussions and members of the laboratory of N.J. for technical help. We thank the Functional Genomic Center Zürich and the Clinical Chemistry Department of the University Hospital Zürich for technical support and expertise and T. Korn for the $Cd155$-KO mice. We thank the NIH Tetramer Core Facility (NIH Contracts 75N93020D00005 and RRID:SCR_026557) for providing LCMV tetramers. We thank Chun-Hsi Su for his help with the infection

experiments. This work was supported by the Swiss National Science Foundation (PP00P3_181037, 310030_197590 and 320030-236138 to N.J. and 197683 to C.G.K.), the European Research Council (677200 Immune Regulation to N.J. and 101141363 to M.I.), the EMBO Young Investigator Program (N.J.), the Italian Association for Cancer Research (19891 and 22737 to M.I.), the National Health and Medical Research Council (NHMRC, to A.V. and A.K.) and a fellowship from the University of Melbourne (to A.V.).

## Author contributions

Conceptualization: C.P. and N.J. Experimentation: C.P., R.D., A.M., A.T., V.F., J.D.L. and L.W. Analysis: C.P., R.D., A.T. and F.A. Formal analysis: N.P., A.A. and A.V. Writing, original draft: C.P. and N.J. Review and editing: all authors. Funding acquisition: N.J. Supervision: C.G.K., M.I., A.K., A.Y., M.H. and N.J.

## Competing interests

The authors declare no competing interests.

## Additional information

**Extended data** is available for this paper at https://doi.org/10.1038/s41590-025-02300-w.

**Correspondence and requests for materials** should be addressed to Nicole Joller.

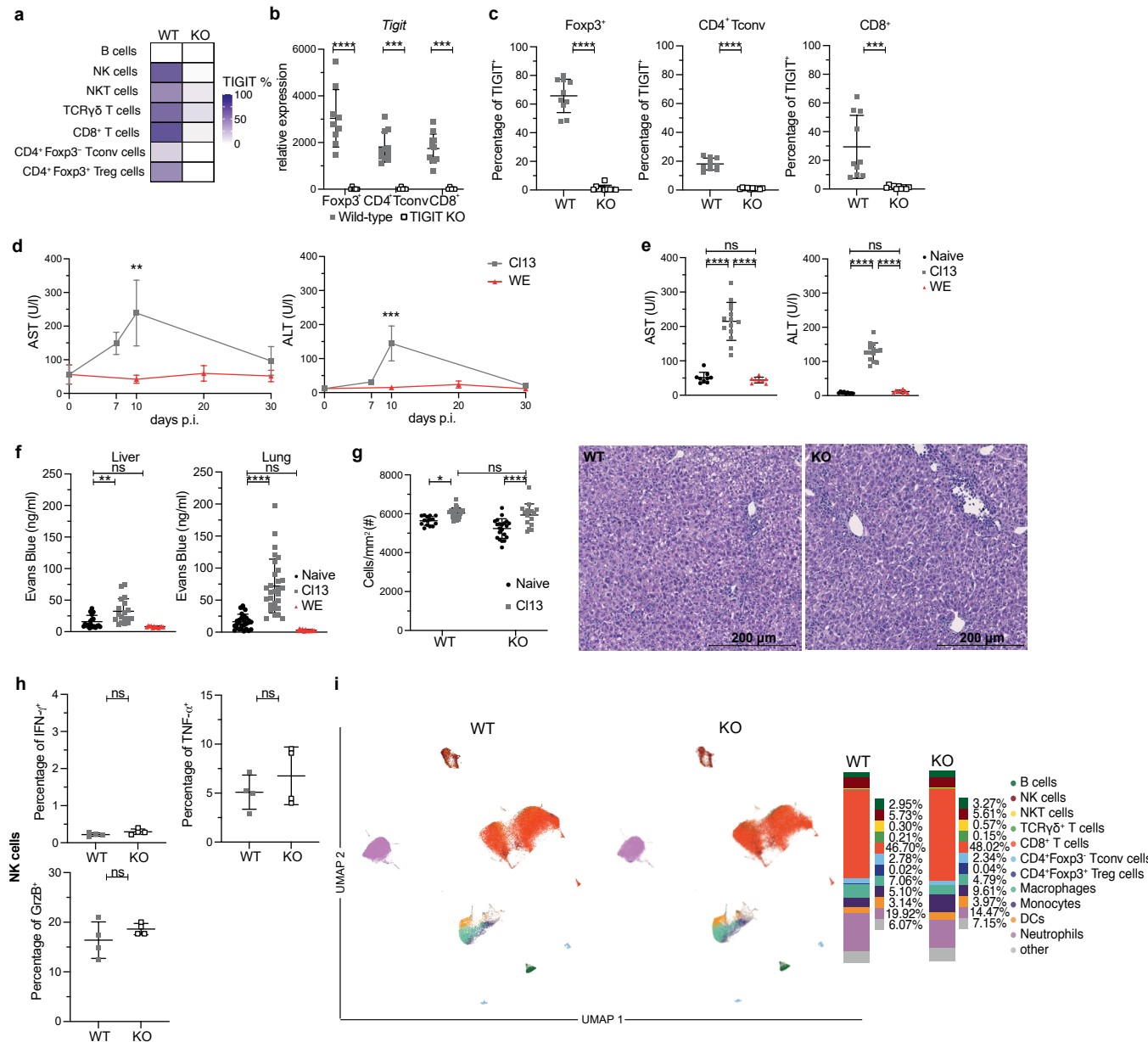

**Extended Data Fig. 1 | Tissue pathology is increased in LCMV Cl13 versus LCMV WE infection. a.** Heatmap of TIGIT expression in splenocytes from LCMV Cl13 infected (d10) WT and *Tigit*- KO mice. **b., c.** Relative *Tigit* mRNA (**b.**) and protein (**c.**) expression in splenic T cells sorted or gated by subset from LCMV Cl13 infected WT and *Tigit*- KO mice (mean ∓ SD; **b.** pool of 3 independent experiments, 2way ANOVA, Tukey's test; **c.** pool of 2 independent experiments, unpaired two-sided t test). **d.** Serum AST and ALT levels in LCMV Cl13 and WE infection over time (mean ∓ SD; pool of 2 independent experiments, unpaired two-sided t test) and **e.** on day 10 p.i. and in naïve controls (mean ∓ SD; pool of 2 independent experiments, ordinary one-way ANOVA, Tukey's test). **f.** Quantification of Evans Blue extravasation in liver and lung of naïve and LCMV

Cl13 and WE infected mice (mean ∓ SD; pool of 3-8 independent experiments, ordinary one-way ANOVA, Tukey's test, liver n = 20, 18, 12; lung n = 28, 29, 12). **g.** Cellular infiltrate in liver from naïve and LCMV Cl13 infected WT and *Tigit*-KO mice (mean ∓ SD; 1 representative experiment, selected ROI, WT n = 15, 23; KO n = 21, 16, biological replicates n = 2-3, 2way ANOVA, Tukey's test) and representative hematoxylin and eosin (H&E) staining of infected liver sections. **h.** IFN-γ, TNF-α and Granzyme B frequencies in NK cells from LCMV Cl13 infected WT and *Tigit*-KO spleens (mean ∓ SD; 1 representative experiment, unpaired two-sided t test, n = 4). **i.** UMAPs and relative composition of immune cells in the lung of WT and *Tigit*-KO mice after LCMV Cl13 infection.

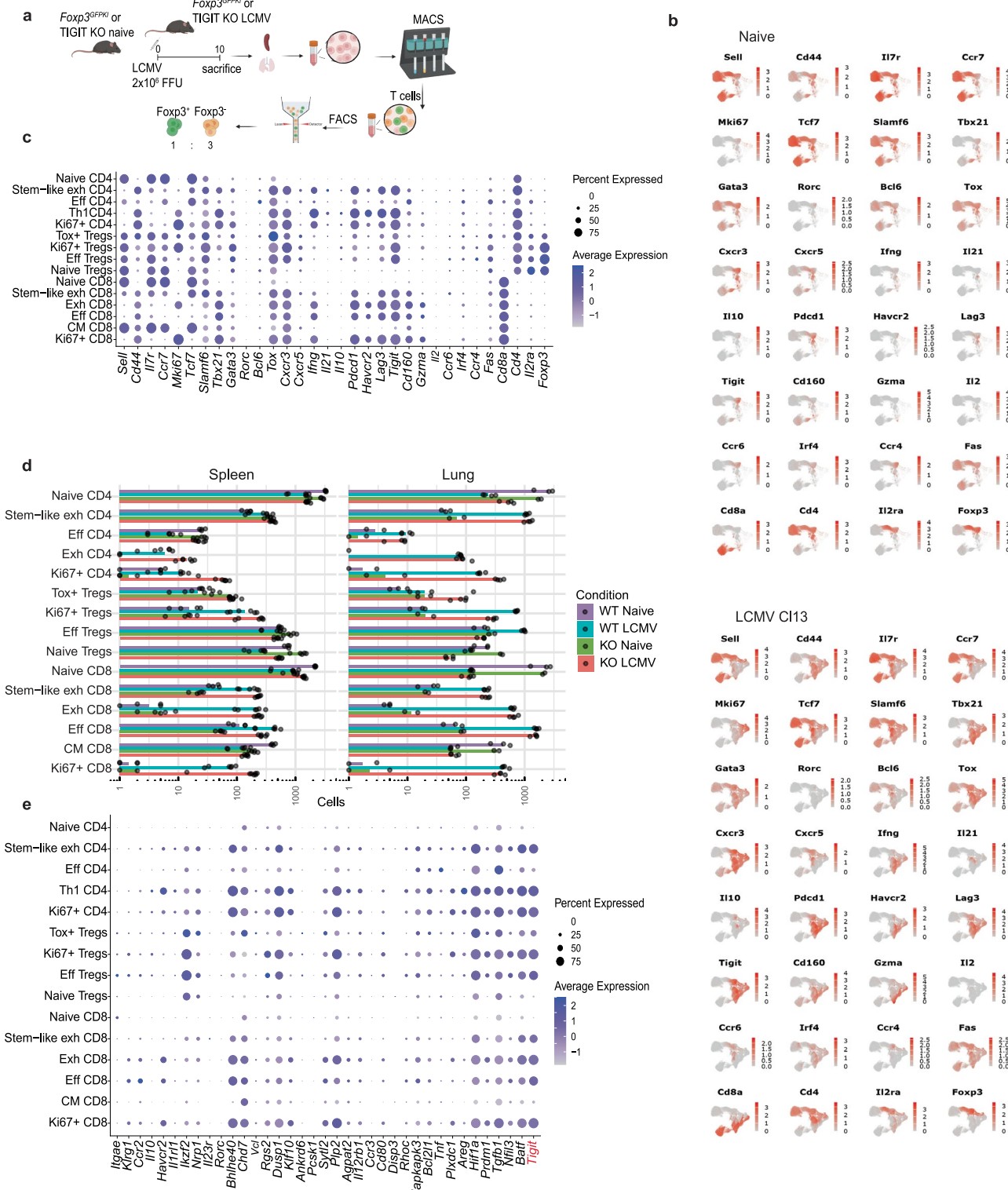

**Extended Data Fig. 2 | Transcriptional profiling of WT and *Tigit*-KO T cells.**
**a**. Experimental scheme for sample preparation for scRNA- and scTCR-seq (Created in BioRender. Joller, N. (2025) https://BioRender.com/pprb5wp).
**b**. UMAP plots showing log-normalized transcript levels of marker genes used for determining the T cell subset in the dataset. UMAPs are split by condition (naïve and LCMV Cl13). **c**. DotPlot showing standardized gene expression values of genes used for cell type and cluster annotation. **d**. Barplots showing cell numbers captured in each sample tabulated by cell type in each tissue of the

scRNA-seq dataset. Each point shows the cell number of each 10x run, and the barplot shows the median of the cell type in the denoted condition. Within cell types, samples are grouped by genotype (WT and *Tigit*-KO) and naïve or LCMV Cl13 (technical replicates: WT spleen naïve n = 4; WT spleen LCMV n = 6; WT lung naïve n = 3; WT lung LCMV n = 4; KO spleen naïve n = 5; KO spleen LCMV n = 6; KO lung naïve n = 2; KO lung LCMV n = 4). **e**. DotPlot showing standardized gene expression values of the repair signature genes and *Tigit*.

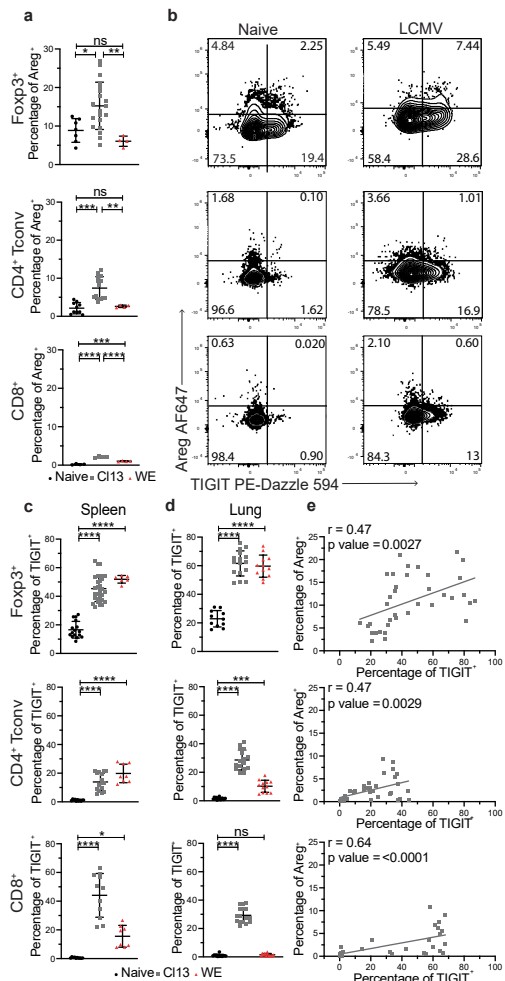

**Extended Data Fig. 3 | TIGIT and Areg are coexpressed. a**. Areg frequencies in lung Treg cells, conventional CD4[+] T cells, CD8[+] T cells after LCMV Cl13, LCMV WE infection and in naïve controls (mean ∓ SD; pool of 2-6 independent experiments, ordinary one-way ANOVA, Tukey's test, Treg cells n = 7, 20 4; CD4[+] Tconv n = 10, 14, 4; CD8[+] n = 4, 4, 4). **b**. Representative FACS plots for TIGIT and Areg expression in lung T cell subtypes from naïve and LCMV Cl13 infected mice (d10). **c.-d**. TIGIT frequencies in splenic (**c**.) and lung (**d**.) Treg cells, conventional CD4[+] T cells, CD8[+] T cells after LCMV Cl13, LCMV WE infection and in naïve controls (mean ∓ SD; pool

of 5-10 independent experiments, ordinary one-way ANOVA, Tukey's test; **c**. Treg cells n = 16, 32, 6; CD4[+] Tconv n = 11, 16, 8; CD8[+] n = 10, 11, 8; **d**. Treg cells n = 11, 16, 12; CD4[+] Tconv n = 15, 16, 12; CD8[+] n = 13, 16, 12). **e**. Correlation plots for Areg and TIGIT in splenic Treg cells, conventional CD4[+] T cells and CD8[+] T cells after LCMV Cl13 infection (pool of 7 independent experiments, simple linear regression and Pearson r correlation; Treg cells p value = 0.0027, CD4[+] Tconv p value = 0.0029, CD8[+] p value = <0.0001).

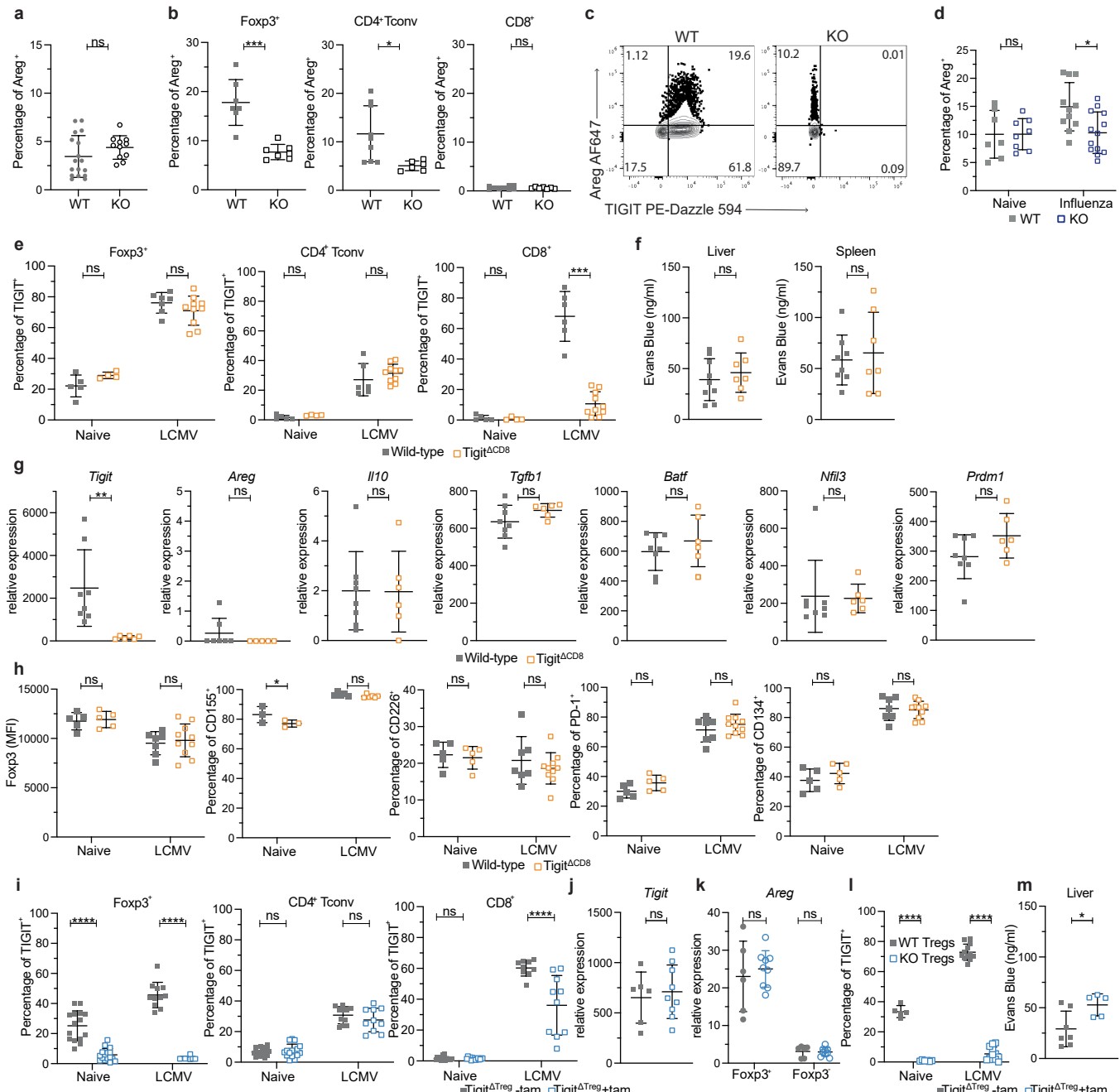

**Extended Data Fig. 4 | Areg expression is TIGIT-dependent. a.-b.** Areg in (**a.**) naïve spleen Treg cells and (**b.**) LCMV lung T cells (WT and *Tigit* KO, mean ∓ SD; pool of (**a.**) 5 and (**b.**) 2 independent experiments, unpaired two-sided t test). **c.-d.** Representative plots (**c.**) and summary (**d.**) of Areg in lung Treg cells upon influenza infection (WT and *Tigit*-KO, mean ∓ SD; pool of 3 independent experiments, 2way ANOVA, Šídák's test). **e.** TIGIT in splenic T cells (WT and *Tigit*^ΔCD8, mean ∓ SD; pool of 3 independent experiments, 2way ANOVA, Tukey's test). **f.** Evans Blue in WT and *Tigit*^ΔCD8 mice after LCMV (mean ∓ SD; pool of 2 independent experiments, unpaired two-sided t test). **g.** *Tigit* and repair signature gene expression in splenic CD8⁺ T cells after LCMV (mean ∓ SD; pool of 2 independent experiments, unpaired two-sided t test). **h.** Foxp3 MFI and frequencies of indicated markers on splenic Treg cells after LCMV (mean ∓ SD;

pool of 2 independent experiments, 2way ANOVA, Tukey's test). **i.** TIGIT in splenic T cells from *Foxp3*^GFPcreERT2*xTigit*^fl/fl (*Tigit*^ΔTreg) mice (+tamoxifen (tam) or vehicle (-tam)) (mean ∓ SD; pool of 4 independent experiments, 2way ANOVA, Tukey's test). **j.** *Tigit* in naïve splenic Foxp3⁻ T cells of *Tigit*^ΔTreg mice (+tam or -tam) (mean ∓ SD; 1 representative experiment, unpaired two-sided t test, n = 6, 9). **k.** *Areg* expression in naïve splenic T cells from *Tigit*^ΔTreg mice (+tam or -tam) (mean ∓ SD; pool of 2 independent experiments, 2way ANOVA, Tukey's test). **l.** TIGIT in splenic WT and *Tigit*-KO Treg cells from heterozygous *Tigit*^ΔTreg females (+tam) (mean ∓ SD; pool of 2 independent experiments, 2way ANOVA, Tukey's test). **m.** Evans Blue in LCMV infected *Tigit*^ΔTreg mice (+tam or -tam) (mean ∓ SD; pool of 2 independent experiments, unpaired two-sided t test).

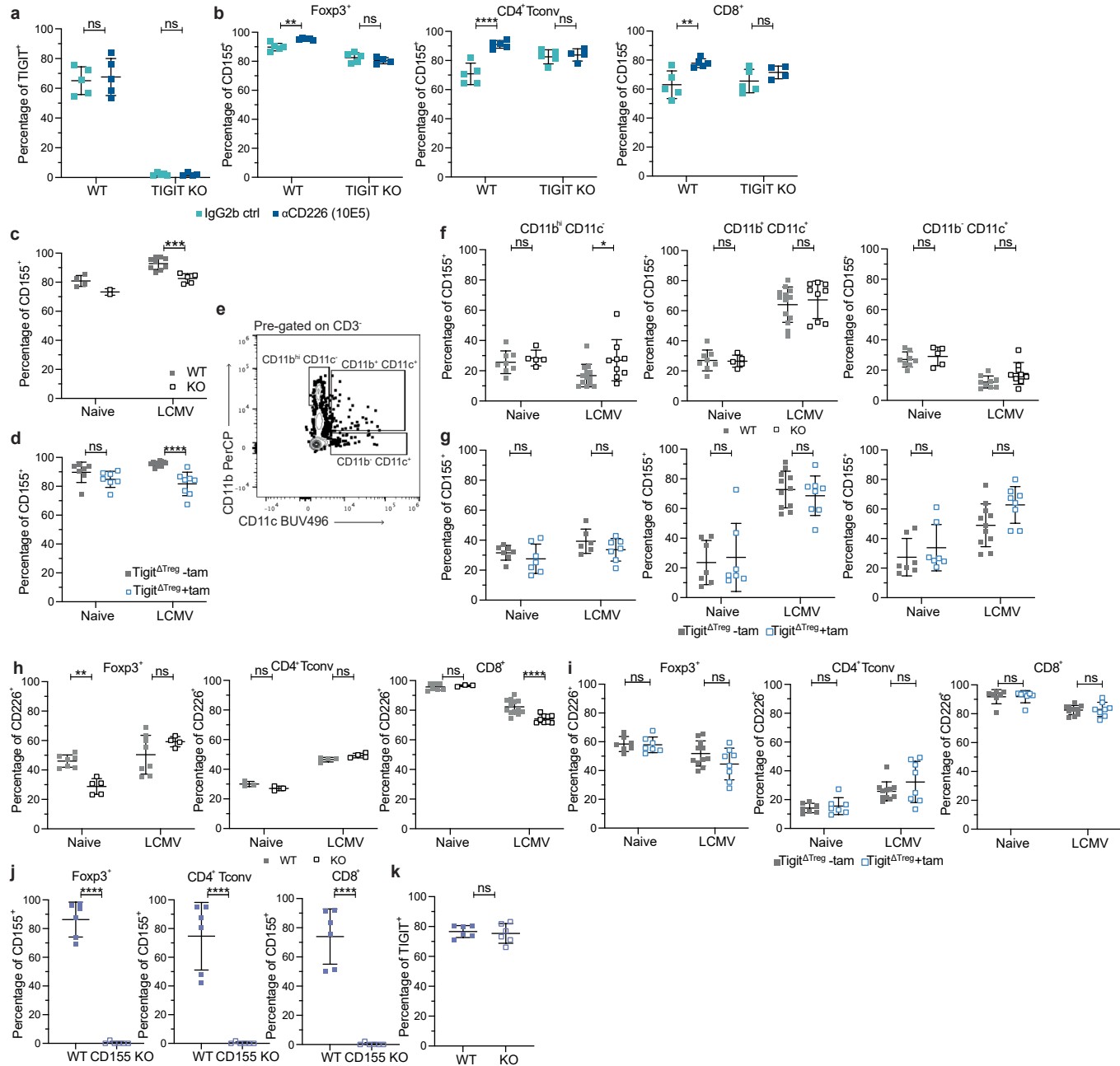

**Extended Data Fig. 5 | CD155 and CD226 expression dynamics. a.-b.** WT and *Tigit*-KO mice infected with LCMV Cl13 were treated with anti-CD226 blocking Ab (or isotype control) and analyzed for expression of TIGIT in Foxp3⁺ Treg cells (**a**.) and CD155 (**b**.) in T cells (mean ∓ SD; 1 representative experiment, 2way ANOVA, Tukey's test, n = 4 for KO aCD226, 5 for all others). **c.-d.** CD155 frequencies in splenic Foxp3⁺ T cells from naïve and LCMV infected WT and Tigit-KO mice (**c**.) or *Foxp3^{GFPcreERT2}xTigit^{fl/fl}* (*Tigit^{ΔTreg}*) mice treated with tamoxifen or with vehicle as control (**d**.) (mean + SD; **c**. 1 representative experiment, naïve n = 4, 2; LCMV n = 9, 5 and **d**. pool of 2 independent experiments, 2way ANOVA, Tukey's test). **e.** Representative FACS plot of myeloid cell gating, pre-gated on CD3⁻ live cells. **f.-g.** CD155 frequencies in CD11b^{hi}CD11c⁻, CD11b⁺CD11c⁺ and CD11b⁻CD11c⁺ myeloid cells isolated from spleens of naïve and LCMV infected WT and *Tigit*-KO mice (**f**.) or *Foxp3^{GFPcreERT2}xTigit^{fl/fl}* (*Tigit^{ΔTreg}*) mice treated with tamoxifen or with vehicle as control (**g**.) (mean ∓ SD; pool of 2 independent experiments, 2way ANOVA, Tukey's test). **h.-i.** CD226 frequencies in T cells isolated from spleens of naïve and LCMV infected WT and *Tigit*-KO mice (**h**.) or *Foxp3^{GFPcreERT2}xTigit^{fl/fl}* (*Tigit^{ΔTreg}*) mice treated with tamoxifen or with vehicle as control (**i**.) (mean ∓ SD; pool of 2 independent experiments, 2way ANOVA, Tukey's test). CD155 (**j**.) and TIGIT (**k**.) expression in splenic T cells isolated from LCMV infected WT and *Cd155*-KO mice (mean ∓ SD; pool of 3 independent experiments, unpaired two-sided t test).

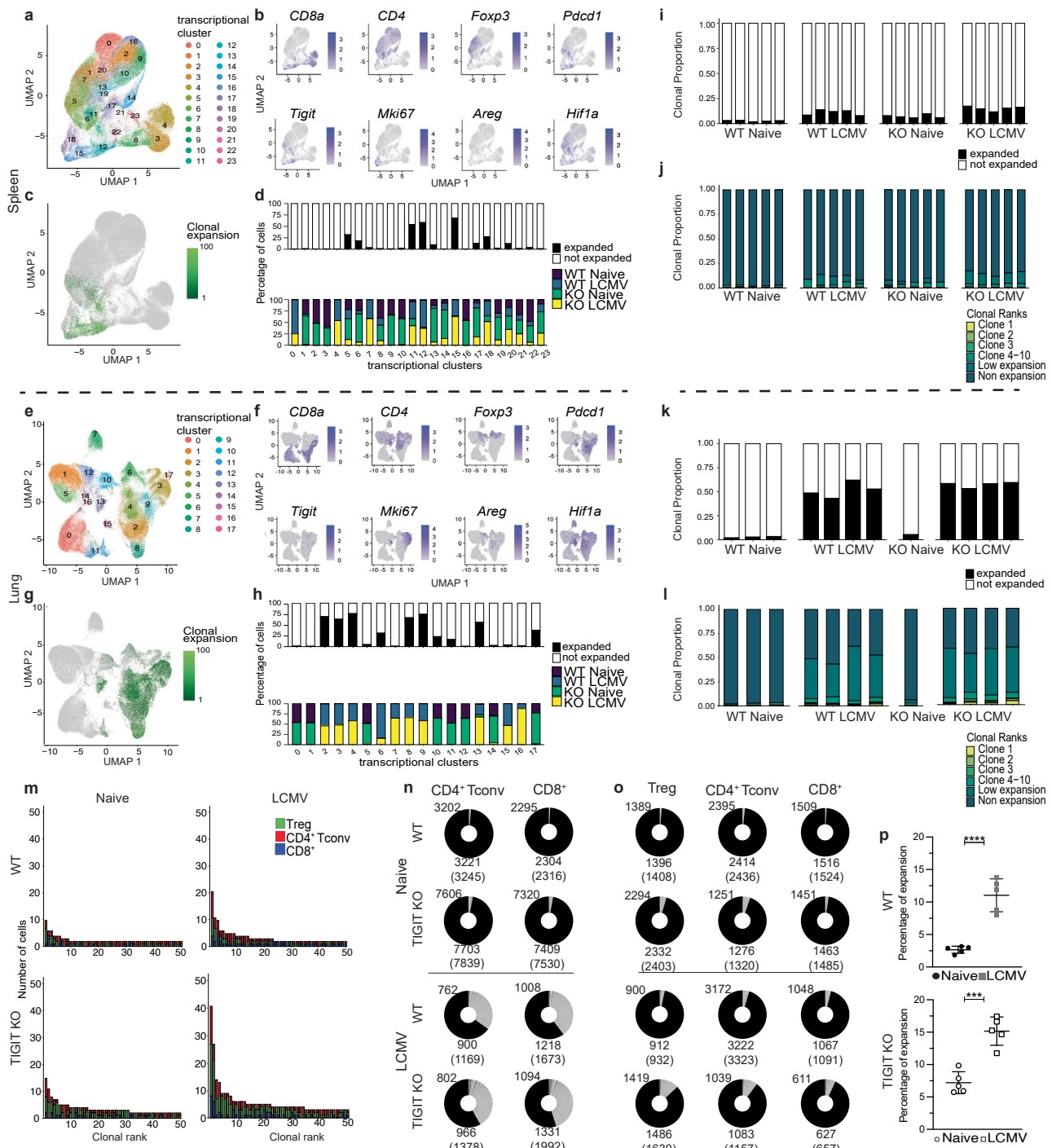

**Extended Data Fig. 6 | TCR repertoire analysis.** scRNA/TCR-seq data from lung T cells of naïve and LCMV Cl13 infected WT and *Tigit*-KO mice. Cells from spleen (**a.**-**d.**, **i.**-**j.**, **m.**, **o.**) and lung (**e.**-**h.**, **k.**-**l.**, **n.**) were analyzed separately. **a.**-**c.**, **e.**-**g.** UMAP based on total gene expression annotated by cluster (**a.**, **e.**), transcript levels of marker genes (**b.**, **f.**) and indicating clonal expansion (**c.**, **g.**). **d.**, **h.** Proportion of expanded cells and condition distribution by transcriptional clusters. **i.**, **k.** Clonal proportion of expanded vs non-expanded cells based on VDJ information across individual mice and (**j.**, **l.**) corresponding expansion clonal rank distributions for repertoires. **m.** Representative expansion plots for one individual mouse per condition in spleen for the 50 most expanded clones per mouse colored based on T cell subset. **n.**, **o.** Clonal expansion donut plots for Foxp3+ Treg cells, conventional CD4+ T cells and CD8+ T cells for one representative mouse per group. **p.** Frequencies of clonal expansion in WT and *Tigit*-KO mice in LCMV infected spleen (mean ∓ SD; n = 5, unpaired two-sided t test).

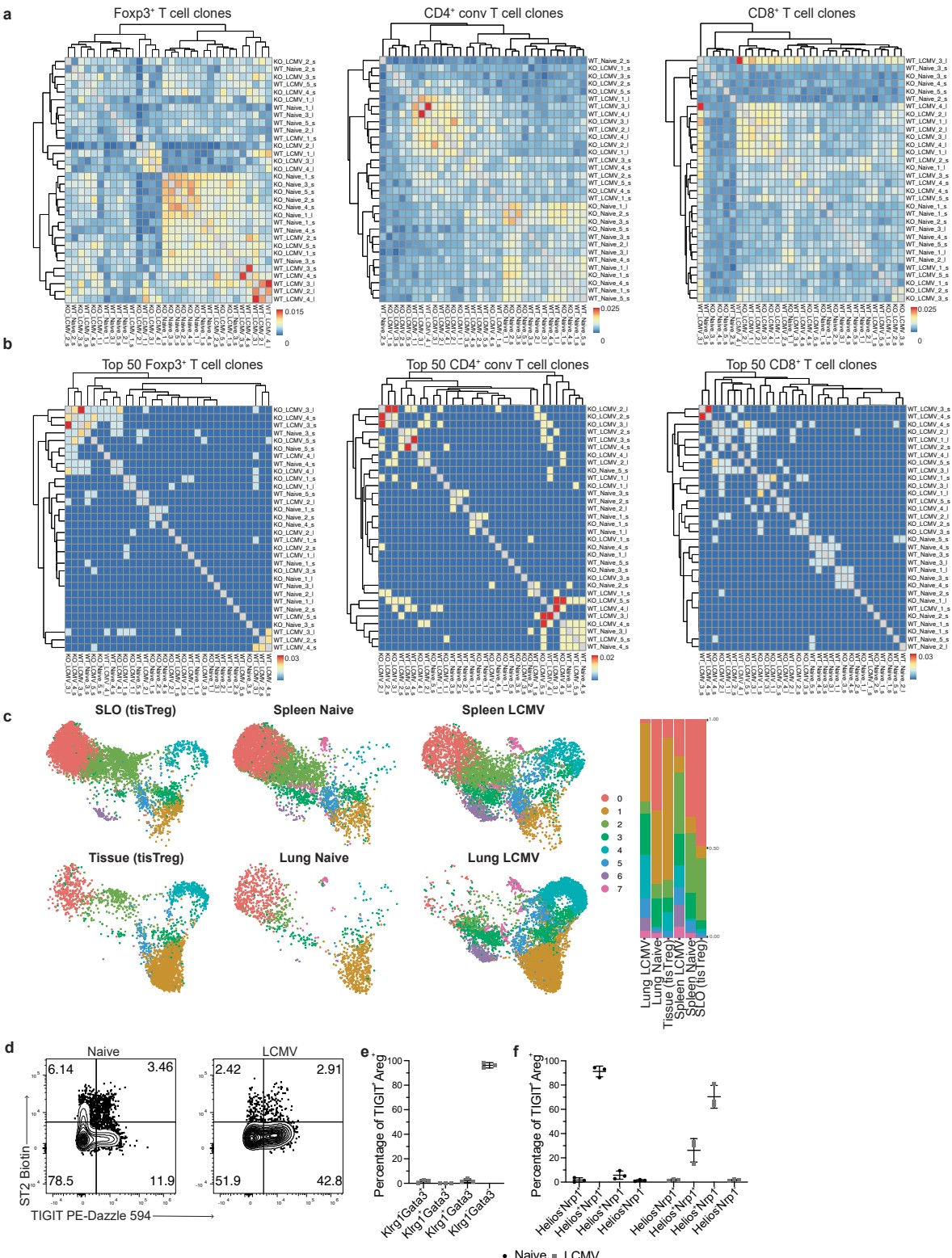

**Extended Data Fig. 7 | T cell clonality and comparison to classical tissue Treg cells. a.**, **b.** scRNA/TCR-seq data from lung T cells of naïve and LCMV Cl13 infected WT and *Tigit*-KO mice. Clonal overlap of the VDJ sequences across all clones (**a.**) and for the 50 most expanded clones (**b.**) per mouse sorted by T cell subset. **c.** Integrated RNA-seq data from LCMV infection and published naïve ST2[+] tissue Treg dataset[17]. UMAPs of indicated organs and datasets and cluster composition

are shown. **d.** Representative FACS plots of ST2 and TIGIT expression in splenic Treg cells from naïve and LCMV Cl13 infected mice. **e.**-**f.** Frequencies of TIGIT[+]Areg[+] Treg cells among Klrg1 and Gata3 SP, DP and DN Foxp3[+] T cells (**e.**) or among Helios and Nrp1 SP, DP and DN Foxp3[+] T cells (**f.**) (mean ∓ SD; one representative experiment, n = 3).

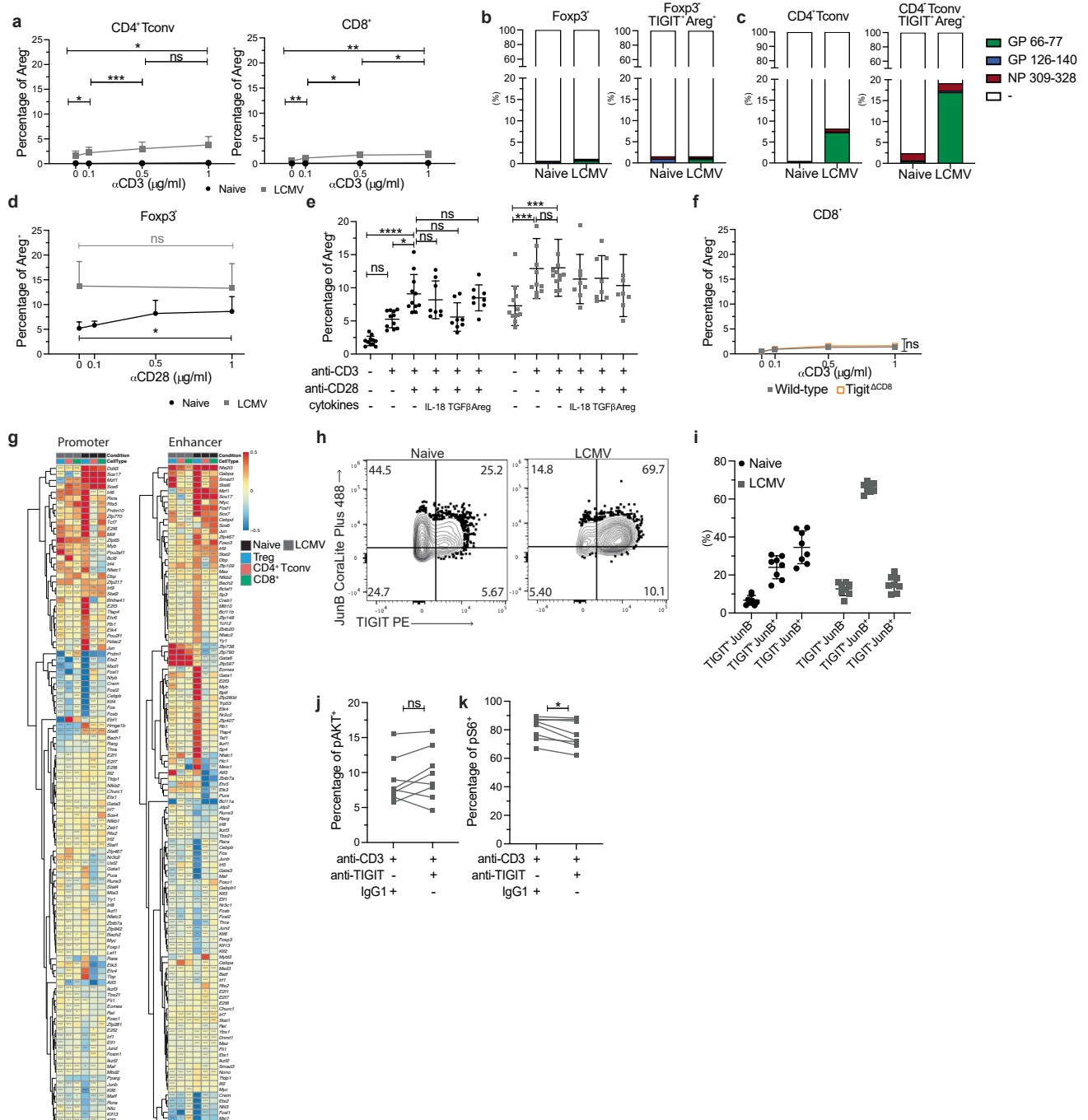

**Extended Data Fig. 8 | TCR-dependent stimulation of TIGIT⁺ Treg cells.**
**a.**, **d.**, **f.** Areg frequencies in splenic conventional CD4⁺ T cells and CD8⁺ T cells
(**a.**), Treg cells (**d.**) and CD8⁺ T cells (**f.**) stimulated with titrated amounts of αCD3
(**a.**, **f.**) or αCD28 (**d.**) *in vitro*. Comparison between cells from naïve and LCMV
Cl13 infected mice (**a.**, **d.**) (mean ∓ SD; pool of 2 independent experiments,
paired two-sided t test) and (**f.**) LCMV infected WT and *CD8ᶜʳᵉxTigitᶠˡ/ᶠˡ* (*Tigitᐟᴰ⁸*)
mice (mean ∓ SD; 1 representative experiment, paired t test). **b.**-**c.** Frequencies
of LCMV tetramer-positive total (left) or TIGIT⁺Areg⁺ (right) Foxp3⁺ T cells (**b.**)
and CD4⁺ conventional T cells (**c.**) in naïve and LCMV Cl13 infected mice (pool of
2 independent experiments). **e.** Areg frequencies in total naïve (black) or LCMV
infected (grey) splenic Treg cells stimulated with αCD3, αCD28 and cytokines

*in vitro* (mean ∓ SD; 3 pooled experiments, 2way ANOVA, Tukey's test).
**g.** SCENIC analysis of scRNA-seq of T cells from naïve or LCMV Cl13 infected
WT and *Tigit*-KO mice. Heatmaps show differential regulon activity between
WT and *Tigit*-KO T cells in promoter (left) and enhancer (right) regions.
**h.**-**i.** Representative FACS (**h.**) and summary plots (**i.**) of TIGIT and JunB
expression on Treg cells from naïve and LCMV infected mice (mean ∓ SD; pool
of 2 independent experiments, 2way ANOVA, Tukey's test). **j.**-**k.** pAKT (**j.**) and
pS6 (**k.**) frequencies in Treg cells from LCMV infected mice treated *in vitro* with
αCD3 in combination with TIGIT agonistic or isotype control Ab (pool of 2
independent experiments, paired t test).

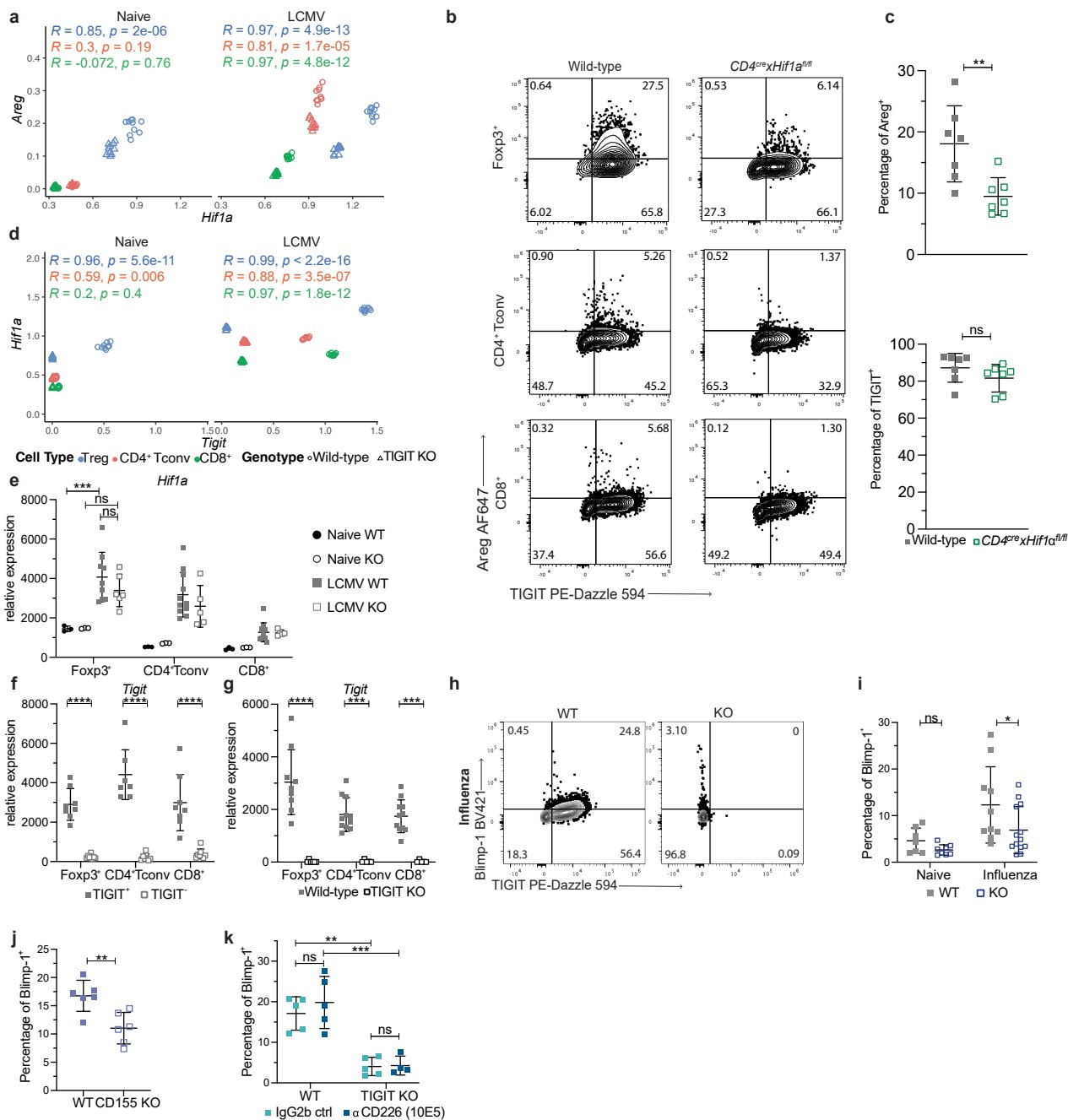

**Extended Data Fig. 9 | Hif1α regulates Areg expression in a TIGIT-independent manner. a.** Scatterplots showing the expression pattern of *Hif1a* and *Areg* (two-sided p-value for the R ≠ 0 null hypothesis – p-value of pearson correlation, no multiple testing correction was applied). **b., c.** Representative FACS plots for TIGIT and Areg expression in splenic Treg cells, CD4⁺ Tconv cells and CD8⁺ T cells from LCMV infected *Cd4^cre^xHif1a^fl/fl^* mice and controls and **c.** quantification for expression in Treg cells (mean + SD; pool of 2 independent experiments, unpaired two-sided t test). **d.** Scatterplots showing the expression pattern of *Hif1a* and *Tigit*. **e.** Relative *Hif1a* mRNA expression in FACS sorted splenic Treg cells, CD4⁺ Tconv cells and CD8⁺ T cells from naïve and LCMV infected WT and *Tigit*-KO mice (mean ∓ SD; pool of 3 independent experiments, 2way ANOVA,

Tukey's test). **f., g.** Relative *Tigit* mRNA expression in FACS sorted splenic TIGIT⁺ and TIGIT⁻ (**f.**) and WT vs *Tigit*-KO (**g.**) Treg cells, CD4⁺ Tconv cells and CD8⁺ T cells from LCMV infected mice (mean ∓ SD; pool of 2 and 3, respectively, independent experiments, 2way ANOVA, Tukey's test). **h., i.** Representative FACS (**h.**) and summary plots (**i.**) of TIGIT and Blimp-1 expression on Treg cells from control or influenza infected WT and *Tigit*-KO mice (mean ∓ SD; pool of 2 independent experiments, 2way ANOVA, Tukey's test). **j.-k.** Blimp-1 frequencies in splenic Treg cells from LCMV infected (**j.**) WT or *Cd155*-KO mice and (**k.**) WT or *Tigit*-KO mice treated with anti-CD226 blocking Ab or isotype (mean ∓ SD; **j.** pool of 3 independent experiments, unpaired two-sided t test; **k.** one representative experiment, 2way ANOVA, Tukey's test).

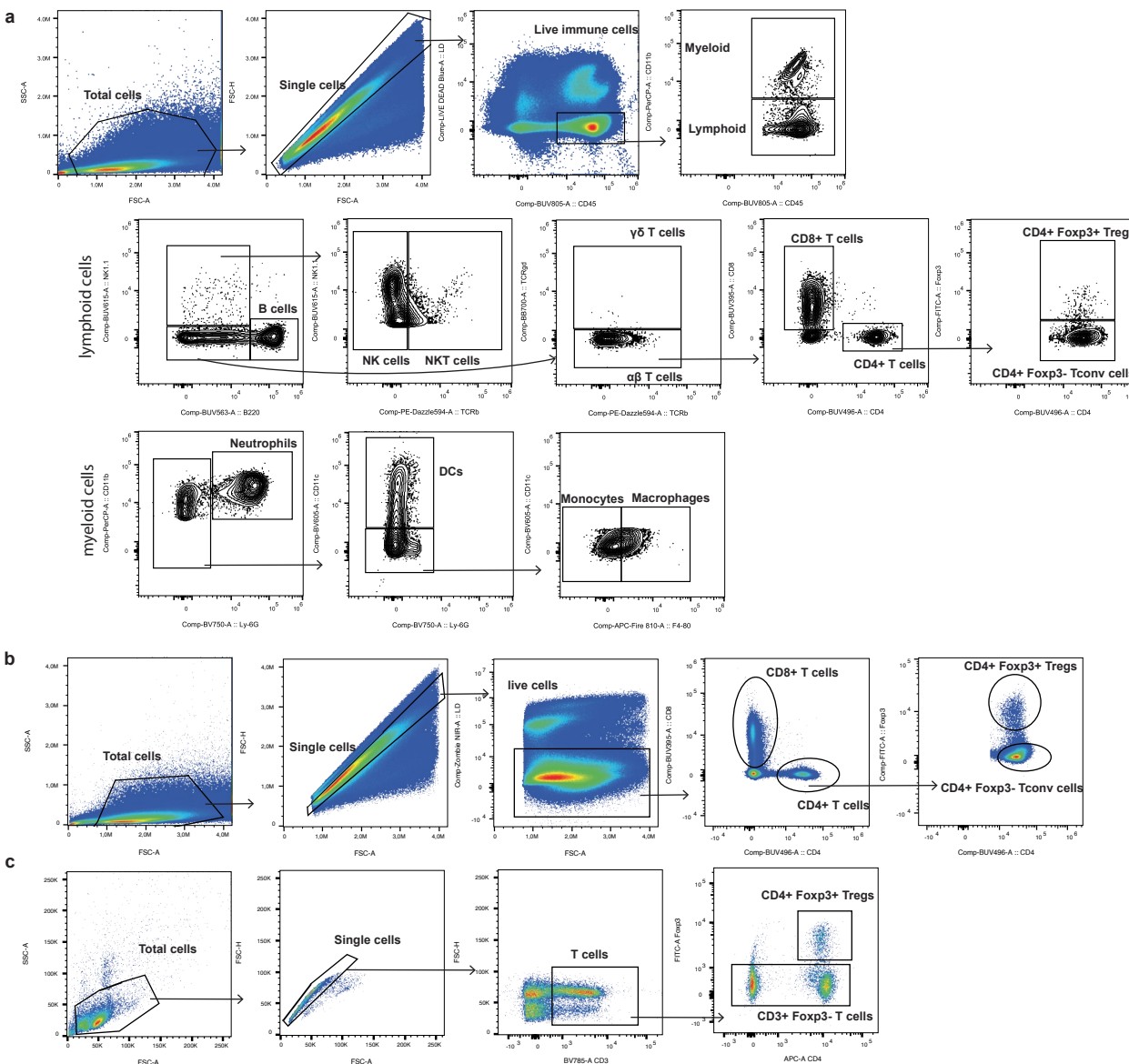

**Extended Data Fig. 10 | Gating strategies. a**. Gating strategy of lymphoid and myeloid cells used for immunophenotyping. **b**. Gating strategy of T lymphocytes used throughout all the experiments. **c**. Gating strategy of T lymphocytes used for T cell sorting for the scRNA/scTCR-seq experiment.

# Reporting Summary

## Statistics

For all statistical analyses, confirm that the following items are present in the figure legend, table legend, main text, or Methods section.

| n/a | Confirmed | |
|---|---|---|
| ☐ | ☒ | The exact sample size (*n*) for each experimental group/condition, given as a discrete number and unit of measurement |
| ☐ | ☒ | A statement on whether measurements were taken from distinct samples or whether the same sample was measured repeatedly |
| ☐ | ☒ | The statistical test(s) used AND whether they are one- or two-sided<br>*Only common tests should be described solely by name; describe more complex techniques in the Methods section.* |
| ☒ | ☐ | A description of all covariates tested |
| ☐ | ☒ | A description of any assumptions or corrections, such as tests of normality and adjustment for multiple comparisons |
| ☐ | ☒ | A full description of the statistical parameters including central tendency (e.g. means) or other basic estimates (e.g. regression coefficient) AND variation (e.g. standard deviation) or associated estimates of uncertainty (e.g. confidence intervals) |
| ☐ | ☒ | For null hypothesis testing, the test statistic (e.g. *F*, *t*, *r*) with confidence intervals, effect sizes, degrees of freedom and *P* value noted<br>*Give P values as exact values whenever suitable.* |
| ☒ | ☐ | For Bayesian analysis, information on the choice of priors and Markov chain Monte Carlo settings |
| ☒ | ☐ | For hierarchical and complex designs, identification of the appropriate level for tests and full reporting of outcomes |
| ☐ | ☒ | Estimates of effect sizes (e.g. Cohen's *d*, Pearson's *r*), indicating how they were calculated |

*Our web collection on statistics for biologists contains articles on many of the points above.*

## Software and code

Policy information about availability of computer code

| Data collection | The Flow cytometric assays: BD LSR Fortessa (BD Bioscience), BD FACS Symphony A5 (BD Bioscience), Cytek Aurora (Cytek Biosciences)<br>The single-cell RNA-Seq analysis: NovaSeq6000 (Illumina) |
|---|---|
| Data analysis | Digital gene expression matrix of single cell sequencing data was obtained from the raw reads using Cell Ranger (v7.0.0).<br>Seurat (v4.0.3 and v5.0.1) was used to further analyze the data.<br>The R package DoubletFinder (v2.0) was applied to filter doublet cells. Batch correction was performed using Harmony (v1.2). SCENIC (v1.1.2) was used to identify regulons. Visualizations were performed with ggplot2 (v3.5.1). Pearson r correlation was obtained from ggpubr (v0.6.0).<br>The code is available at https://github.com/hemberg-lab/tigit-paper-figures<br>The integrated transcriptome and TCR repertoire data object was generated using the R package Platypus (v3.1). Heatmaps displaying clonal overlap were produced using the pheatmap package (v1.0.12).<br>Statistical analysis: GraphPad Prism (v10.0.3)<br>Flow cytometric analysis: FlowJo (v10.8.2 and 10.10.0), SpectroFlo (Cytek Biosciences, v3.0.3) , |

For manuscripts utilizing custom algorithms or software that are central to the research but not yet described in published literature, software must be made available to editors and reviewers. We strongly encourage code deposition in a community repository (e.g. GitHub). See the Nature Portfolio guidelines for submitting code & software for further information.

## Data

Policy information about availability of data

All manuscripts must include a data availability statement. This statement should provide the following information, where applicable:
- Accession codes, unique identifiers, or web links for publicly available datasets
- A description of any restrictions on data availability
- For clinical datasets or third party data, please ensure that the statement adheres to our policy

All scRNA-seq and scTCR-seq data from WT and TIGIT KO T cells have been deposited on the ArrayExpress database at EMBL-EBI (www.ebi.ac.uk/arrayexpress) and is available via accession number E-MTAB-8861, the Seurat Object was deposited at Zenodo and is available via doi: 10.5281/zenodo.14041419. Blimp-1 ChIP-seq and RNA-seq data was previously reported and is available in the Gene Expression Omnibus database under accession numbers GSE79339 and GSE121838.

## Research involving human participants, their data, or biological material

Policy information about studies with human participants or human data. See also policy information about sex, gender (identity/presentation), and sexual orientation and race, ethnicity and racism.

| | |
|---|---|
| Reporting on sex and gender | not applicable |
| Reporting on race, ethnicity, or other socially relevant groupings | not applicable |
| Population characteristics | not applicable |
| Recruitment | not applicable |
| Ethics oversight | not applicable |

Note that full information on the approval of the study protocol must also be provided in the manuscript.

# Field-specific reporting

Please select the one below that is the best fit for your research. If you are not sure, read the appropriate sections before making your selection.

☒ Life sciences  ☐ Behavioural & social sciences  ☐ Ecological, evolutionary & environmental sciences

For a reference copy of the document with all sections, see nature.com/documents/nr-reporting-summary-flat.pdf

# Life sciences study design

All studies must disclose on these points even when the disclosure is negative.

| | |
|---|---|
| Sample size | Sample sizes were chosen based on power calculations using estimated effect size ranges that were based on our previous experience. When additional datapoints were collected under identical conditions in later experiments with, these were included in the pooled data. Sample size was sufficient to demonstrate statistically significant differences in comparisons between experimental groups by the respective statistical tests as indicated in the figure legends. Adequate sample size was confirmed based on reproducibility between independent experiments. |
| Data exclusions | Wherever possible, experiments included a positive and negative control. If results obtained for these controls were not as expected, data points were excluded. |
| Replication | All experiments were repeated at least twice to confirm reproducibility, with specific sample sizes and details indicated in figure captions. |
| Randomization | For animal experiments, animals with the same genotype were randomized to the the different treatment groups. |
| Blinding | For histological analysis, samples were blinded. Due to technical reasons (indication of genotypes and treatment on cage cards of animals) animal experiments were not carried out in a blinded manner. |

# Reporting for specific materials, systems and methods

We require information from authors about some types of materials, experimental systems and methods used in many studies. Here, indicate whether each material, system or method listed is relevant to your study. If you are not sure if a list item applies to your research, read the appropriate section before selecting a response.

## Materials & experimental systems

| n/a | Involved in the study |
|-----|----------------------|
| ☐ | ☒ Antibodies |
| ☒ | ☐ Eukaryotic cell lines |
| ☒ | ☐ Palaeontology and archaeology |
| ☐ | ☒ Animals and other organisms |
| ☒ | ☐ Clinical data |
| ☒ | ☐ Dual use research of concern |
| ☒ | ☐ Plants |

## Methods

| n/a | Involved in the study |
|-----|----------------------|
| ☒ | ☐ ChIP-seq |
| ☐ | ☒ Flow cytometry |
| ☒ | ☐ MRI-based neuroimaging |

# Antibodies

| Antibodies used | Antibody, Fluorochrome, Dilution factor, Clone, Supplier:<br>Areg Alexa-Fluor647 1:100 G-4 Santa Cruz<br>Areg  Biotin 1:200 - R&D<br>B220 BUV563 1:200 RA3-6B2 BD<br>Blimp-1 APC 1:100 5E7 BioLegend<br>Blimp-1 BV421 1:100 5E7 BD<br>CD112  BV711 1:100 829038 BD<br>CD11b  PerCP 1:400 M1/70 BioLegend<br>CD11c  BV605 1:100 N418 BioLegend<br>CD155  BUV737 1:100 TX56 BD<br>CD155 PE 1:300 TX56 BioLegend<br>CD226 APC/Fire750 1:100 10E5 BioLegend<br>CD226  PE-Cy7 1:100 10E5 BD<br>CD3 BUV805 1:100 17A2 eBioscience<br>CD3  BV785 1:200 17A2 BioLegend<br>CD4 BUV737 1:300 RM4-5 BD<br>CD4 APC 1:200 RM4-5 BioLegend<br>CD4 BUV496 1:500 RM4-5 BD<br>CD4 APC-Cy7 1:300 GK1.5 BD<br>CD44 APC/Fire750 1:300 IM7 BioLegend<br>CD45  BUV805 1:300 30-F11 BD<br>CD62L  BV650 1:400 MEL-14 BioLegend<br>CD8 BUV395 1:500 53-6.7 BD<br>CXCR3  Biotin 1:100 SA051D1 BioLegend<br>F4/80  APC-Fire810 1:100 BM8 BioLegend<br>Foxp3 FITC 1:200 FJK-16s eBioscience<br>Foxp3 PE 1:200 FJK-16s Thermo<br>Foxp3 eF450 1:200 FJK-16s eBioscience<br>Gata3 PE-Cy5 1:200 TWAJ eBioscience<br>GM-CSF  PE-Cy7 1:100 MP1-22E9 BioLegend<br>Granzyme B  APC 1:100 GB12 Thermo<br>GFP Tag AF488 1:400 FM264G BioLegend<br>Helios BUV563 1:100 22F6 eBioscience<br>IFN-g  PE 1:300 XMG1.2 BioLegend<br>JunB CoraLitePlus488 1:125 - ProteinTech<br>Ki-67  BUV737 1:300 SolA15 Thermo<br>Klrg1 BUV661 1:200 2F1 BD<br>LD  Zombie NIR 1:500 - BioLegend<br>LD  Blue 1:500 - Thermo<br>Ly6G  BV750 1:200 1A8 BD<br>NK1.1  BUV615 1:200 PK136 BD<br>Nrp1 BV7711 1:500 V46-1954 BD<br>pAKT AF647 1:10 M89-61 BD<br>PD-1 BV785 1:500 29F.1A12 BioLegend<br>pS6 PerCP-eF710 1:10 cupk43k eBioscience<br>ST2 Biotin 1:100 DIH9 BioLegend<br>Streptavidin BUV563 1:800 - eBioscience<br>Streptavidin BV711 1:300 - BioLegend<br>Streptavidin  BV480 1:500 - BD<br>TCF1/7  R718 1:100 S33-966 BD<br>TCRb  PE-Dazzle594 1:300 H57-597 BioLegend<br>TCRgd BB700 1:200 GL3 BD<br>TIGIT PE-Dazzle594 1:50 1G9 BioLegend<br>TIGIT  BV421 1:50 TX99 BD<br>TIGIT PerCP-eF710 1:200 GIGD7 Thermo<br>TNF-a  BV421 1:100 MP6-XT22 BioLegend |
|---|---|

| Validation | All antibodies used in this study were validated for species and application by the vendors. All antibodies were further validated and titrated using isotype (and where possible biological) control samples in each experiment. |
|---|---|

# Animals and other research organisms

Policy information about studies involving animals; ARRIVE guidelines recommended for reporting animal research, and Sex and Gender in Research

| Laboratory animals | Adult 7-45 weeks old male and female mice were used and age and sex were matched between experimental groups. The following strains were used: C57BL/6 (from Janvier) Foxp3-GFP.KI (from VK Kuchroo) Tigitfl/fl x B6.129(Cg)-Foxp3tm4(YFP/icre)Ayr/J (from VK Kuchroo) Tigitfl/fl x Foxp3tm9(EGFP/cre/ERT2)Ayr/J (crossed in Zurich, parental strains from VK Kuchroo and JAX) Tigitfl/fl x C57BL/6-Tg(Cd8a-cre)1Itan/J (crossed in Zurich, parental strains from VK Kuchroo and JAX) B6.Cg-Tg(Cd4-cre)1Cwi/BfluJ x B6.129-Hif1atm3Rsjo/J (from N Aceto) Lckcre x Prdm1fl/fl (from JAX) B6N.129S2-Pvr<tm1Gbn>/J (from T Korn) LckcrexPrdm1fl/fl mice were bred and housed at the Peter Doherty Institute, Melbourne, Australia, all other strains at the University of Zurich or the University of Basel, Switzerland, under specific pathogen-free conditions. Animals were housed in individually ventilated cages containing autoclaved bedding and nesting material, with standard diet and water ad libitum and a 12h light/dark cycle (18–23°C, 40-60% humidity). |
|---|---|
| Wild animals | not applicable |
| Reporting on sex | Male and female mice were used and sex was matched between experimental groups. For experiments using heterozygous Tigitfl/fl x Foxp3tm9(EGFP/cre/ERT2)Ayr/J mice, females were used as the Foxp3 gene is located on the x chromosome. |
| Field-collected samples | not applicable |
| Ethics oversight | Cantonal veterinary office of Zurich Cantonal veterinary office of Basel University of Melbourne animal ethics committee |

Note that full information on the approval of the study protocol must also be provided in the manuscript.

# Plants

| Seed stocks | not applicable |
|---|---|
| Novel plant genotypes | not applicable |
| Authentication | not applicable |

# Flow Cytometry

## Plots

Confirm that:

☒ The axis labels state the marker and fluorochrome used (e.g. CD4-FITC).

☒ The axis scales are clearly visible. Include numbers along axes only for bottom left plot of group (a 'group' is an analysis of identical markers).

☒ All plots are contour plots with outliers or pseudocolor plots.

☒ A numerical value for number of cells or percentage (with statistics) is provided.

## Methodology

| Sample preparation | Single-cell suspensions of spleen were generated by mechanical disruption in RPMI 1640 (Gibco) supplemented with 10% FCS (Corning or Capri), penicillin and streptomycin (100 U/ml, Gibco), 2mmol/l L-glutamine (Gibco), Brefeldin A (BioLegend) and Marimastat (Sigma). Red blood cells were lysed using ACK lysis buffer (155 mM $NH_4Cl$, 10 mM $KHCO_3$, 0.1 mM Na2EDTA, pH: 7,4) for 3-5 minutes. Single-cell suspensions of the lung were generated by enzymatic tissue digestion at 37 °C with 0.2 mg/ |
|---|---|

ml Deoxyribonuclease I (Sigma) and 2.4 mg/ml Collagenase type I (Gibco) and mechanical disruption using the GentleMACS. Media was supplemented with Brefeldin A (BioLegend) and Marimastat (Sigma). Lung immune cells were separated from epithelial and parenchymal cells by 30% Percoll gradient centrifugation (GE Healthcare).

Single-cell suspensions were directly stained ex vivo or incubated at 37°C for 2h 45min with Brefeldin A (BioLegend), Monensin (BioLegend) and Marimastat (Sigma). Samples were incubated with surface antibodies diluted in PBS for 20 minutes at RT. For intracellular cytokines staining, cells were permeabilized using the Cytofix/Cytoperm kit (BD Bioscience) for 5-10 minutes at RT, followed by 30 minutes incubation at RT with the intracellular antibodies mix. For transcription factors staining, cells were permeabilized with the Foxp3/Transcription factor staining buffer set (eBioscience) for 40 minutes at RT. The Zombie NIR or LD Blue fixable dyes were used to exclude dead cells.

| | |
|---|---|
| Instrument | BD LSR Fortessa (BD Bioscience), BD FACS Symphony A5 (BD Bioscience), Cytek Aurora (Cytek Biosciences) |
| Software | SpectroFlo (Cytek Biosciences, v3.0.3) and FlowJo (BD Bioscience, v10.8.2 or 10.10.0) software were used for data analysis.. For the R analysis: unmixed and pre-gated cells were imported into RStudio using R (v4.4.1) and the flowCore (v2.16.0) and CATALYST (v1.28.0) packages. Data was analyzed in the FlowJo workspace using the flowWorkspace (v4.16.0) and CytoML (v2.16.0) packages and the UMAP and FlowSOM plugins. |
| Cell population abundance | not applicable |
| Gating strategy | Cells cells were first gated by FSC/SSC to exclude debris, followed by gating by FSC-A/FSC-H to exclude doublets and by LiveDead dye/CD45 to exclude dead cells. Target cell populations for further analysis were gated by cell lineage markers (CD4, CD8, Foxp3). Gates for all subsequent markers were set based on isotype controls. The gating strategy is illustrated in Extended Data Figure 10. |

☒ Tick this box to confirm that a figure exemplifying the gating strategy is provided in the Supplementary Information.

