## [Peer Review File · Nature Immunology]

The co-inhibitory receptor TIGIT promotes tissue protective functions in T cells

Corresponding Author: Professor Nicole Joller

Version 0:

Reviewer comments:

Reviewer #1

(Remarks to the Author)

The biological functions of TIGIT remain incompletely understood and are of significant clinical importance. This study elucidates a role for Treg-expressed TIGIT in tissue repair in the context of viral infection. Expression of TIGIT in T cells is shown to correlate with a repair signature, including Areg and Batf, with convincing co-expression of Areg and TIGIT being demonstrated at the level of both mRNA and protein.

Experiments involving selective loss of TIGIT from CD8 T cells or Treg are used to reveal that Treg-expression of TIGIT is critical for the regulation of tissue pathology following LCMV infection. The inhibition of Areg in Treg following anti-TIGIT antibody treatment appears to be partial (Fig. 3a), so it is reassuring to see such clear data in the genetic loss experiments (Fig. 3h). Loss of Treg TIGIT expression, and thus Areg production, is shown to be associated with tissue pathology, including vascular leakage in lung and liver. TCR analysis indicates that the TIGIT-expressing Treg responding to LCMV infection are distinct from the classic "tissue repair" Treg characterised by ST2 expression.

Lastly, Blimp-1 is identified as a downstream mediator of TIGIT-dependent Areg induction, with the authors showing that Blimp-1 directly binds upstream of the Areg promoter. Hif1a is also found to promote Areg induction, in a manner independent of TIGIT.

Overall, this is an exciting and well performed study that makes a substantial contribution to knowledge in the field. The discovery of a role for TIGIT in tissue regeneration has implications for fundamental biological research as well as clinical translation.

Minor comments:

TCR stimulation is shown to trigger Areg production in TIGIT-expressing Treg (Fig 4i): it would be interesting to explore how this is influenced by costimulation or cytokines.

Loss of TIGIT from Treg also reduced Areg production from Tconv and CD8 (Fig 3h) – did the authors check whether TIGIT expression had been altered on these populations?

Line 123: Typo, Fig. 2 g (not 2 h)

Reviewer #2

(Remarks to the Author)

Strengths

The study explores a novel function of TIGIT, transitioning its role from immune regulation to active tissue repair, thereby extending the understanding of co-inhibitory receptors in immunology.

It establishes TIGIT-induced Areg production as a critical mechanism for limiting tissue pathology during viral infections, emphasizing its translational relevance.

The study utilizes a combination of genetic models (e.g., TIGIT KO, Foxp3GFPcreERT2xTigitfl/fl) and cutting-edge

techniques such as single-cell RNA sequencing (scRNA-seq) and TCR repertoire analysis. The integration of transcriptomic, proteomic, and functional assays strengthens the conclusions. Identification of the TIGIT-Blimp1-Areg axis as a driver of tissue repair. Highlights the dependency of Tregs on both TCR and TIGIT signaling for Areg production. The use of LCMV Clone 13 infection as a model highlights the physiological relevance of the findings to infection-induced tissue pathology.

Weaknesses

1. Limited Scope of Experimental Validation

Single Viral Model: The conclusions are primarily based on the LCMV Clone 13 model. Broader validation using additional infection models (e.g., Influenza, MCMV) or sterile injury models (e.g., mechanical injury, ischemia-reperfusion) would enhance the generalizability of the findings.

2 Despite the claim that CD8+ T cells are dispensable for TIGIT-mediated tissue repair, the study does not explore alternative roles these cells might play. Recommendation: Investigate whether TIGIT+ CD8+ T cells contribute to other repair-related functions or indirectly modulate Tregs.

3. The TIGIT-DNAM1-PVR axis, which is central to TIGIT signaling, is not thoroughly explored in the context of Areg production. The roles of CD155 and DNAM1 remain unclear. The authors should characterize the dynamics of DNAM1 and CD155 expression during LCMV infection and investigate their influence on TIGIT-mediated Areg induction.

4. The manuscript states that TIGIT drives Blimp-1 expression, which subsequently induces Areg. However, the upstream signaling pathways leading to Blimp-1 induction are not fully elucidated. Please explore whether TIGIT engagement activates specific signaling cascades, such as STAT or PI3K pathways, contributing to Blimp-1 regulation.

5. The study identifies TIGIT+ Tregs as distinct from classical tissue-resident Tregs but does not explore the origin or specific phenotypic differences of this subset. Please include a detailed characterization of TIGIT+ Tregs, comparing their transcriptomic and proteomic profiles with classical tissue Tregs.

6. The reported Areg expression levels in some figures (e.g., Fig. 5I) are substantially higher than those in other experiments, raising questions about consistency.

7 Baseline Areg expression in naïve mice (Fig. 2H) is inconsistent with earlier claims that TIGIT is required for infection-induced Areg production. The authors should Clarify whether baseline Areg is truly TIGIT-independent or represents a different regulatory mechanism.

8. Statements such as "TIGIT acts as a mediator of tissue repair upstream of Areg" and "TIGIT directly induces the expression of Areg" are overly definitive. The indirect contribution of TIGIT (e.g., via Tregs or accessory signals) remains a possibility.

9. Fig. 1 lacks information about NK cell dynamics, which are known contributors to LCMV pathology and could be influenced by TIGIT deficiency.

Minor Comments

1. Figure-Specific Feedback:

1. Figure 1A: The ALT levels in TIGIT KO mice without infection show a trend toward higher baseline pathology. A larger sample size might clarify significance.

2. Figure 3H: Adjust sample sizes in Foxp3GFPcreERT2xTigitfl/fl Tregs to improve statistical

4. Figure 4I: Repeat experiments with purified T cells to confirm that observed effects are T

5. Ensure consistent terminology when referring to "repair Tregs," "tissue Tregs," and "TIGIT+ Tregs" to avoid confusion.

6. The discussion does not address potential translational implications, such as targeting TIGIT for therapeutic modulation of tissue repair. Including this perspective would enhance the manuscript's clinical relevance.

Reviewer #3

(Remarks to the Author)

In this manuscript, Panetti et al showed that upon viral infection, TIGIT directly drives expression of the tissue growth factor amphiregulin (Areg) in Treg cells and contributes to tissue repair. They also showed that TIGIT engagement upon TCR stimulation induces Blimp-1, promoting Areg production and tissue repair. The studies were well performed, and the data looks convincing. However, it has already been reported that Tregs produce Areg and contribute to tissue repair. It was also known that TIGIT is expressed on Tregs and that TIGIT limits tissue pathology during viral infection. For these reasons, this paper does not represent a significant conceptual advance.

The specific comments are as follows:

1. The authors showed that the treatment with TIGIT-specific antibodies suppressed the Areg production from Tregs. This suggests that the binding of TIGIT to its ligands is involved in Areg production. However, it is unclear which cells express the ligand and interact with Tregs. Clarification of this issue would help our understanding of the pathogenesis.

2. TIGIT and DNAM-1 compete for ligand binding each other. Since TIGIT antibody treatment or TIGIT deficiency promotes the binding of DNAM-1 to CD155, the DNAM-1 downstream signaling might be involved in the Areg production. To clarify this issue, it would be helpful to use an anti-DNAM-1 antibody with anti-TIGIT antibodies or single or double knockout mice for TIGIT and DNAM-1.

3. The authors showed that TCR stimulation is important for TIGIT-induced production of Areg in an in vitro assay in Fig. 4,

using anti-CD3 antibodies. What stimulates the TCR of Tregs in vivo? Are the Areg⁺ Tregs LCMV-specific Tregs? The authors should discuss the necessity of TCR stimulation in vivo Areg⁺ Tregs during LCMV infection, taking into account the immunological mechanisms.

4. In Fig. 2, there is a big difference in the production of Areg by TIGIT between LCMV CL13-infected and LCMV WE-infected mice. The authors should explain the mechanisms of this difference. Also, in Fig. 2h, it is unclear how TIGIT promotes the production of Areg only in LCMV-infected mice but not in naive mice.

Minor

1. Label in Fig 3f, g, l, j, and k, Ext Fig 3c and d, Ext Fig 4c and e-g

Decision Letter:

23rd Dec 2024

Dear Nicole,

Thank you for providing a point-by-point response to the referees' comments on your manuscript entitled, "The co-inhibitory receptor TIGIT promotes tissue protective functions in T cells". As noted previously, while they find your work of considerable potential interest, they have raised quite substantial concerns that must be addressed. In light of these comments, we cannot accept the current manuscript for publication, but would be very interested in considering a revised version that addresses these concerns along the lines proposed in your point-by-point rebuttal.

We invite you to submit a substantially revised manuscript, however please bear in mind that we will be reluctant to approach the referees again in the absence of major revisions.

Specifically, the revision should include new experiments to address:

- (1) address how co-stimulation or cytokines influence Areg production by TIGIT expressing cells
- (2) assess Areg production in WT and TIGIT KO mice in additional infection models
- (3) further investigate the repair capacity of CD8⁺ T cells
- (4) determine the expression of CD226 and CD155 in naive and LCMV CL13 infected WT and TIGIT cKO mice
- (5) determine the functional impact of CD226 and CD155 on Areg expression and/or the expression of TIGIT during LCMV infection in vivo
- (6) explore additional signals and signaling pathways that might be involved in TIGIT-dependent induction of tissue repair

Please include the additional textual clarifications as indicated in your response letter.

When you revise your manuscript, please take into account all reviewer and editor comments, please highlight all changes in the manuscript text file in Microsoft Word format.

* If you have not done so already please begin to revise your manuscript so that it conforms to our Article format instructions at <http://www.nature.com/nl/authors/index.html>. Refer also to any guidelines provided in this letter.

The Reporting Summary can be found here:

When submitting the revised version of your manuscript, please pay close attention to our <https://www.nature.com/nature-portfolio/editorial-policies/image-integrity> Digital Image Integrity Guidelines and to the following points below:

Extended Data figures and tables are online-only (appearing in the online PDF and full-text HTML version of the paper), peer-reviewed display items that provide essential background to the Article but are not included in the printed version of the paper due to space constraints or being of interest only to a few specialists. A maximum of ten Extended Data display items (figures and tables) is typically permitted. When re-submitting your manuscript, please ensure that any supplementary figures and tables that are more critical to the manuscript's conclusions are converted to Extended data to increase these data's visibility.

Link Redacted

If you wish to submit a suitably revised manuscript we would hope to receive it within 6 months. If you cannot send it within this time, please let us know. We will be happy to consider your revision so long as nothing similar has been accepted for publication at Nature Immunology or published elsewhere.

Nature Immunology is committed to improving transparency in authorship. As part of our efforts in this direction, we are now requesting that all authors identified as 'corresponding author' on published papers create and link their Open Researcher and Contributor Identifier (ORCID) with their account on the Manuscript Tracking System (MTS), prior to acceptance. ORCID helps the scientific community achieve unambiguous attribution of all scholarly contributions. You can create and link your ORCID from the home page of the MTS by clicking on 'Modify my Springer Nature account'. For more information please visit please visit www.springernature.com/orcid.

Thank you for the opportunity to review your work.

Happy Holidays,

Laurie

Laurie A. Dempsey, Ph.D.
Senior Editor
Nature Immunology
l.dempsey@us.nature.com
ORCID: 0000-0002-3304-796X

Reviewers' Comments:

Reviewer #1 (Remarks to the Author):

The biological functions of TIGIT remain incompletely understood and are of significant clinical importance. This study elucidates a role for Treg-expressed TIGIT in tissue repair in the context of viral infection. Expression of TIGIT in T cells is shown to correlate with a repair signature, including Areg and Batf, with convincing co-expression of Areg and TIGIT being demonstrated at the level of both mRNA and protein.

Experiments involving selective loss of TIGIT from CD8 T cells or Treg are used to reveal that Treg-expression of TIGIT is critical for the regulation of tissue pathology following LCMV infection. The inhibition of Areg in Treg following anti-TIGIT antibody treatment appears to be partial (Fig. 3a), so it is reassuring to see such clear data in the genetic loss experiments (Fig. 3h). Loss of Treg TIGIT expression, and thus Areg production, is shown to be associated with tissue pathology, including vascular leakage in lung and liver. TCR analysis indicates that the TIGIT-expressing Treg responding to LCMV infection are distinct from the classic "tissue repair" Treg characterised by ST2 expression.

Lastly, Blimp-1 is identified as a downstream mediator of TIGIT-dependent Areg induction, with the authors showing that

Blimp-1 directly binds upstream of the Areg promoter. Hif1a is also found to promote Areg induction, in a manner independent of TIGIT.

Overall, this is an exciting and well performed study that makes a substantial contribution to knowledge in the field. The discovery of a role for TIGIT in tissue regeneration has implications for fundamental biological research as well as clinical translation.

Minor comments:

TCR stimulation is shown to trigger Areg production in TIGIT-expressing Treg (Fig 4i): it would be interesting to explore how this is influenced by costimulation or cytokines.

Loss of TIGIT from Treg also reduced Areg production from Tconv and CD8 (Fig 3h) – did the authors check whether TIGIT expression had been altered on these populations?

Line 123: Typo, Fig. 2 g (not 2 h)

Reviewer #2 (Remarks to the Author):

Strengths

The study explores a novel function of TIGIT, transitioning its role from immune regulation to active tissue repair, thereby extending the understanding of co-inhibitory receptors in immunology.

It establishes TIGIT-induced Areg production as a critical mechanism for limiting tissue pathology during viral infections, emphasizing its translational relevance.

The study utilizes a combination of genetic models (e.g., TIGIT KO, Foxp3GFPcreERT2xTigitfl/fl) and cutting-edge techniques such as single-cell RNA sequencing (scRNA-seq) and TCR repertoire analysis.

The integration of transcriptomic, proteomic, and functional assays strengthens the conclusions.

Identification of the TIGIT-Blimp1-Areg axis as a driver of tissue repair. Highlights the dependency of Tregs on both TCR and TIGIT signaling for Areg production. The use of LCMV Clone 13 infection as a model highlights the physiological relevance of the findings to infection-induced tissue pathology.

Weaknesses

1. Limited Scope of Experimental Validation

Single Viral Model: The conclusions are primarily based on the LCMV Clone 13 model. Broader validation using additional infection models (e.g., Influenza, MCMV) or sterile injury models (e.g., mechanical injury, ischemia-reperfusion) would enhance the generalizability of the findings.

2 Despite the claim that CD8+ T cells are dispensable for TIGIT-mediated tissue repair, the study does not explore alternative roles these cells might play. Recommendation: Investigate whether TIGIT+ CD8+ T cells contribute to other repair-related functions or indirectly modulate Tregs.

3. The TIGIT-DNAM1-PVR axis, which is central to TIGIT signaling, is not thoroughly explored in the context of Areg production. The roles of CD155 and DNAM1 remain unclear. The authors should characterize the dynamics of DNAM1 and CD155 expression during LCMV infection and investigate their influence on TIGIT-mediated Areg induction.

4. The manuscript states that TIGIT drives Blimp-1 expression, which subsequently induces Areg. However, the upstream signaling pathways leading to Blimp-1 induction are not fully elucidated. Please explore whether TIGIT engagement activates specific signaling cascades, such as STAT or PI3K pathways, contributing to Blimp-1 regulation.

5. The study identifies TIGIT+ Tregs as distinct from classical tissue-resident Tregs but does not explore the origin or specific phenotypic differences of this subset. Please include a detailed characterization of TIGIT+ Tregs, comparing their transcriptomic and proteomic profiles with classical tissue Tregs.

6. The reported Areg expression levels in some figures (e.g., Fig. 5l) are substantially higher than those in other experiments, raising questions about consistency.

7 Baseline Areg expression in naïve mice (Fig. 2H) is inconsistent with earlier claims that TIGIT is required for infection-induced Areg production. The authors should Clarify whether baseline Areg is truly TIGIT-independent or represents a different regulatory mechanism.

8. Statements such as "TIGIT acts as a mediator of tissue repair upstream of Areg" and "TIGIT directly induces the expression of Areg" are overly definitive. The indirect contribution of TIGIT (e.g., via Tregs or accessory signals) remains a possibility.

9. Fig. 1 lacks information about NK cell dynamics, which are known contributors to LCMV pathology and could be influenced by TIGIT deficiency.

Minor Comments

1. Figure-Specific Feedback:

1. Figure 1A: The ALT levels in TIGIT KO mice without infection show a trend toward higher baseline pathology. A larger sample size might clarify significance.

2. Figure 3H: Adjust sample sizes in Foxp3GFPcreERT2xTigitfl/fl Tregs to improve statistical

4. Figure 4l: Repeat experiments with purified T cells to confirm that observed effects are T

5. Ensure consistent terminology when referring to "repair Tregs," "tissue Tregs," and "TIGIT+ Tregs" to avoid confusion.

6. The discussion does not address potential translational implications, such as targeting TIGIT for therapeutic modulation of

tissue repair. Including this perspective would enhance the manuscript's clinical relevance.

Reviewer #3 (Remarks to the Author):

In this manuscript, Panetti et al showed that upon viral infection, TIGIT directly drives expression of the tissue growth factor amphiregulin (Areg) in Treg cells and contributes to tissue repair. They also showed that TIGIT engagement upon TCR stimulation induces Blimp-1, promoting Areg production and tissue repair. The studies were well performed, and the data looks convincing. However, it has already been reported that Tregs produce Areg and contribute to tissue repair. It was also known that TIGIT is expressed on Tregs and that TIGIT limits tissue pathology during viral infection. For these reasons, this paper does not represent a significant conceptual advance.

The specific comments are as follows:

1. The authors showed that the treatment with TIGIT-specific antibodies suppressed the Areg production from Tregs. This suggests that the binding of TIGIT to its ligands is involved in Areg production. However, it is unclear which cells express the ligand and interact with Tregs. Clarification of this issue would help our understanding of the pathogenesis.
2. TIGIT and DNAM-1 compete for ligand binding each other. Since TIGIT antibody treatment or TIGIT deficiency promotes the binding of DNAM-1 to CD155, the DNAM-1 downstream signaling might be involved in the Areg production. To clarify this issue, it would be helpful to use an anti-DNAM-1 antibody with anti-TIGIT antibodies or single or double knockout mice for TIGIT and DNAM-1.
3. The authors showed that TCR stimulation is important for TIGIT-induced production of Areg in an in vitro assay in Fig. 4, using anti-CD3 antibodies. What stimulates the TCR of Tregs in vivo? Are the Areg⁺ Tregs LCMV-specific Tregs? The authors should discuss the necessity of TCR stimulation in in vivo Areg⁺ Tregs during LCMV infection, taking into account the immunological mechanisms.
4. In Fig. 2, there is a big difference in the production of Areg by TIGIT between LCMV CL13-infected and LCMV WE-infected mice. The authors should explain the mechanisms of this difference. Also, in Fig. 2h, it is unclear how TIGIT promotes the production of Areg only in LCMV-infected mice but not in naive mice.

Minor

1. Label in Fig 3f, g, i, j, and k, Ext Fig 3c and d, Ext Fig 4c and e-g

Version 1:

Reviewer comments:

Reviewer #1

(Remarks to the Author)

The authors have provided insightful responses and additional data to address all of the points raised.

Reviewer #2

(Remarks to the Author)

Textual Revisions:

The authors responded to our request to expand the model systems evaluated by incorporating an influenza infection model. Given the ethical limitations, this was a reasonable choice. However, the comparison between LCMV and influenza-induced responses remains incomplete. Notably, important aspects such as tissue pathology and the temporal dynamics of AREG⁺ Treg accumulation are missing. Moreover, the magnitude of the response in the influenza model appears significantly lower than that observed in the LCMV model (e.g., Extended Data Fig. 4D versus Fig. 2H). Since the purpose of adding a second model was to assess the generalizability of the observed phenomenon, the authors should revise the manuscript to reflect the differential magnitude and context-dependence of the AREG⁺ Treg response across distinct tissue damage settings.

Minor Experimental Revisions:

The identification of CD155-dependent signaling and the lack of a significant effect from CD226 blockade provide valuable mechanistic insights. To further strengthen this conclusion, I recommend that the authors perform a direct comparison of CD3-stimulated naïve versus LCMV-experienced Tregs in the presence or absence of plate-immobilized CD155, with or without CD28 co-stimulation. This will clarify whether prior experience alters the reliance on CD155-mediated signaling for AREG induction.

With these clarifications and additions, the manuscript would be further strengthened and fully suitable for publication.

Reviewer #3

(Remarks to the Author)

The authors responded well to the reviewers' questions.
No further comments from reviewer #3.

Decision Letter:

Our ref: NI-A39159A

31st Jul 2025

Dear Nicole

Thank you for submitting your revised manuscript "The co-inhibitory receptor TIGIT promotes tissue protective functions in T cells" (NI-A39159A). It has now been seen by the original referees and their comments are below. The reviewers find that the paper has improved in revision, and therefore we'll be happy in principle to publish it in Nature Immunology, pending minor revisions to satisfy the referees' final requests and to comply with our editorial and formatting guidelines. Please note, however, that we are willing to overrule referee #2's request to perform an additional in vitro experiment.

We will now perform detailed checks on your paper and will send you a checklist detailing our editorial and formatting requirements in about a week. Please do not upload the final materials and make any revisions until you receive this additional information from us.

If you had not uploaded a Word file for the current version of the manuscript, we will need one before beginning the editing process; please email that to immunology@us.nature.com at your earliest convenience.

Thank you again for your interest in Nature Immunology Please do not hesitate to contact me if you have any questions.

Kind regards,

Laurie

Laurie A. Dempsey, Ph.D.
Senior Editor
Nature Immunology
l.dempsey@us.nature.com
ORCID: 0000-0002-3304-796X

Reviewer #1 (Remarks to the Author):

The authors have provided insightful responses and additional data to address all of the points raised.

Reviewer #2 (Remarks to the Author):

Textual Revisions:

The authors responded to our request to expand the model systems evaluated by incorporating an influenza infection model. Given the ethical limitations, this was a reasonable choice. However, the comparison between LCMV and influenza-induced responses remains incomplete. Notably, important aspects such as tissue pathology and the temporal dynamics of AREG⁺ Treg accumulation are missing. Moreover, the magnitude of the response in the influenza model appears significantly lower than that observed in the LCMV model (e.g., Extended Data Fig. 4D versus Fig. 2H). Since the purpose of adding a second model was to assess the generalizability of the observed phenomenon, the authors should revise the manuscript to reflect the differential magnitude and context-dependence of the AREG⁺ Treg response across distinct tissue damage settings.

Minor Experimental Revisions:

The identification of CD155-dependent signaling and the lack of a significant effect from CD226 blockade provide valuable mechanistic insights. To further strengthen this conclusion, I recommend that the authors perform a direct comparison of CD3-stimulated naïve versus LCMV-experienced Tregs in the presence or absence of plate-immobilized CD155, with or without CD28 co-stimulation. This will clarify whether prior experience alters the reliance on CD155-mediated signaling for AREG induction.

With these clarifications and additions, the manuscript would be further strengthened and fully suitable for publication.

Reviewer #3 (Remarks to the Author):

The authors responded well to the reviewers' questions.
No further comments from reviewer #3.

Point-by-point reply

Manuscript NI-A39159, "The co-inhibitory receptor TIGIT promotes tissue protective functions in T cells"; Panetti et al.

We thank the reviewers for their constructive criticism. We believe that we were able to address all of their concerns and think the additional data has further strengthened our manuscript.

For a better overview, we would like to summarize the main points raised:

1. Recognition of the significance of our main discovery: All reviewers considered our data highly convincing and accepted our central finding that TIGIT drives tissue repair. Reviewers 1 and 2 highlighted that our findings represent a significant advance in the field.

2. Expression and relevance of TIGIT's co-stimulatory counterpart CD226 and their ligand CD155: Reviewers 2 and 3 requested additional investigation or discussion regarding the expression patterns and functional relevance of TIGIT's co-stimulatory counterpart CD226 (DNAM1), and their ligand CD155 (PVR). With our additional experiments outlined in detail below, we found that expression of CD226 or CD155 was largely unaffected by TIGIT modulation. Furthermore, we tested their functional impact through CD226 blockade and CD155 deletion. While CD226 does not influence Areg expression, we could identify CD155 as the functional ligand for TIGIT-mediated Areg induction *in vivo*.

3. Signals contributing to TIGIT-mediated Areg induction: Reviewers 1 and 2 suggested that we further explore additional signals and signalling pathways that might be involved in TIGIT-dependent induction of tissue repair. As outlined in detail below, our additional experiments revealed a requirement for co-stimulation in naive but not *in vivo* primed Tregs. Furthermore, SCENIC analysis revealed distinct regulon activity in WT vs TIGIT KO cells, with increased activity of transcription factors associated with Treg function and stability in the presence of TIGIT, while TCR signalling remained intact.

The detailed description of how we addressed the individual points raised is outlined in the point-by-point response below.

Reviewer #1

The biological functions of TIGIT remain incompletely understood and are of significant clinical importance. This study elucidates a role for Treg-expressed TIGIT in tissue repair in the context of viral infection. Expression of TIGIT in T cells is shown to correlate with a repair signature, including Areg and Batf, with convincing co-expression of Areg and TIGIT being demonstrated at the level of both mRNA and protein.

Experiments involving selective loss of TIGIT from CD8 T cells or Treg are used to reveal that Treg-expression of TIGIT is critical for the regulation of tissue pathology following LCMV infection. The inhibition of Areg in Treg following anti-TIGIT antibody treatment appears to be partial (Fig. 3a), so it is reassuring to see such clear data in the genetic loss experiments (Fig. 3h). Loss of

Treg TIGIT expression, and thus Areg production, is shown to be associated with tissue pathology, including vascular leakage in lung and liver. TCR analysis indicates that the TIGIT-expressing Treg responding to LCMV infection are distinct from the classic “tissue repair” Treg characterised by ST2 expression.

Lastly, Blimp-1 is identified as a downstream mediator of TIGIT-dependent Areg induction, with the authors showing that Blimp-1 directly binds upstream of the Areg promoter. Hif1a is also found to promote Areg induction, in a manner independent of TIGIT.

Overall, this is an exciting and well performed study that makes a substantial contribution to knowledge in the field. The discovery of a role for TIGIT in tissue regeneration has implications for fundamental biological research as well as clinical translation.

We thank reviewer 1 for the insightful and encouraging feedback. We truly appreciate the reviewer’s acknowledgment of our study’s significance and contribution to the field.

Minor comments:

TCR stimulation is shown to trigger Areg production in TIGIT-expressing Treg (Fig 4i): it would be interesting to explore how this is influenced by costimulation or cytokines.

Based on the reviewer’s suggestion, we examined the impact of co-stimulatory signals and cytokines on Areg production by TIGIT⁺ Tregs. In naive Tregs, co-stimulation via CD28 in addition to TCR engagement significantly increased Areg expression in a dose-dependent manner (new Fig. 5 d, e; new Extended Data Fig. 8 e). In contrast, Tregs isolated from infected mice did not show enhanced Areg production upon co-stimulation *in vitro*, as these cells have already been primed *in vivo* (new Fig. 5 d, Extended Data Fig. 8 e). Similarly, cytokines previously reported to induce Areg, such as IL-18, TGF- β , and Areg itself, didn’t further increase Areg expression. Thus, while co-stimulation is required to induce Areg in naive TIGIT⁺ Tregs, it is dispensable in TIGIT⁺ Tregs isolated from infected mice that have undergone *in vivo* priming.

Loss of TIGIT from Treg also reduced Areg production from Tconv and CD8 (Fig 3h) – did the authors check whether TIGIT expression had been altered on these populations?

We have quantified TIGIT protein expression on CD4⁺ Tconv and CD8⁺ T cells (Extended Data Fig. 4 i of the revised manuscript, originally Extended Fig. 4 d). TIGIT deletion is specific in Tregs and doesn’t affect CD4⁺ Tconv. In CD8⁺ T cells, TIGIT expression is unchanged at steady state but reduced following infection, likely due to slightly shifted response kinetics in the cKO mice (Extended Data Fig. 4i, third panel). Moreover, tamoxifen treatment does not alter *Tigit* mRNA levels in total naive Foxp3⁻ cells (Extended Data Fig. 4 j of the revised manuscript), indicating that TIGIT deletion is specific to Tregs and that the observed reduction in TIGIT expression on CD8⁺ T cells is likely a secondary effect upon infection, not genetic deletion.

Line 123: Typo, Fig. 2 g (not 2 h)

The reference is correct. Fig. 2g shows representative plots corresponding to Fig. 2f, while Fig. 2h shows the increase in Areg in TIGIT⁺ (but not TIGIT⁻) Tregs upon infection.

Reviewer #2*Strengths*

The study explores a novel function of TIGIT, transitioning its role from immune regulation to active tissue repair, thereby extending the understanding of co-inhibitory receptors in immunology.

It establishes TIGIT-induced Areg production as a critical mechanism for limiting tissue pathology during viral infections, emphasizing its translational relevance.

The study utilizes a combination of genetic models (e.g., TIGIT KO, Foxp3GFPcreERT2xTigit^{fl/fl}) and cutting-edge techniques such as single-cell RNA sequencing (scRNA-seq) and TCR repertoire analysis.

The integration of transcriptomic, proteomic, and functional assays strengthens the conclusions.

Identification of the TIGIT-Blimp1-Areg axis as a driver of tissue repair. Highlights the dependency of Tregs on both TCR and TIGIT signaling for Areg production. The use of LCMV Clone 13 infection as a model highlights the physiological relevance of the findings to infection-induced tissue pathology.

We thank the reviewer for the thoughtful and positive feedback on our manuscript and recognition of its strengths.

*Weaknesses**1. Limited Scope of Experimental Validation*

Single Viral Model: The conclusions are primarily based on the LCMV Clone 13 model. Broader validation using additional infection models (e.g., Influenza, MCMV) or sterile injury models (e.g., mechanical injury, ischemia-reperfusion) would enhance the generalizability of the findings.

We thank the reviewer for this valuable suggestion. While our current animal licenses do not cover sterile injury models, we aimed to broaden the scope of our study by including influenza as an additional infectious model. Consistent with our results from LCMV infection, we observed co-expression of TIGIT and Areg in Tregs upon influenza infection (new Extended Data Fig. 4 c). Furthermore, we found that the Blimp-1-Areg transcriptional program is similarly induced in TIGIT⁺ Tregs during influenza infection (new Extended Data Fig. 9 h). Notably, this induction is abrogated in TIGIT-deficient mice, confirming that the functional role of TIGIT in driving Areg expression is conserved across infection models (new Extended Data Fig. 4 d and 9 i). These findings support the broader relevance of our conclusions beyond the LCMV model.

2 Despite the claim that CD8⁺ T cells are dispensable for TIGIT-mediated tissue repair, the study does not explore alternative roles these cells might play. Recommendation: Investigate whether TIGIT⁺ CD8⁺ T cells contribute to other repair-related functions or indirectly modulate Tregs.

We thank the reviewer for this thoughtful suggestion. To further explore potential repair-related roles of TIGIT⁺ CD8⁺ T cells, we analyzed the expression of additional tissue repair-associated genes in CD8⁺ T cells from WT and CD8^{cre}xTigit^{fl/fl} mice. As shown in the new Extended Data Fig. 4 g of the revised manuscript, we did not observe any significant differences in the expression of repair signature genes between the two genotypes.

Additionally, we assessed whether the absence of TIGIT in CD8⁺ T cells might indirectly influence

the Treg phenotype or function. However, comprehensive characterization of Tregs in *CD8^{cre}xTigit^{fl/fl}* mice revealed no alterations in the Treg phenotype (Extended Data Fig. 4 h). These data further support our conclusion that TIGIT expression on CD8⁺ T cells doesn't contribute to tissue repair, neither directly nor indirectly through modulation of Tregs.

3. The TIGIT-DNAM1-PVR axis, which is central to TIGIT signaling, is not thoroughly explored in the context of Areg production. The roles of CD155 and DNAM1 remain unclear. The authors should characterize the dynamics of DNAM1 and CD155 expression during LCMV infection and investigate their influence on TIGIT-mediated Areg induction.

We thank the reviewer for raising this important point, which was also brought up by reviewer 3 (see below). In the revised manuscript, we have further analyzed the expression and functional roles of CD226 (DNAM1) and CD155 (PVR) in the context of TIGIT-mediated Areg production (new Fig. 3 l, m; new Extended Data Fig. 5).

We examined whether the expression dynamics of TIGIT, CD226, and CD155 were affected by blockade or deletion of any of these receptors or their ligand. TIGIT expression in Tregs was unaffected by CD226 blockade or deletion of CD155 (new Extended Data Fig. 5 a, k). CD226 expression was also unaffected in T cells from both full and conditional TIGIT KO mice (new Extended Data Fig. 5 h, i). CD155, which is highly expressed on T cells, was slightly upregulated in WT T cells following CD226 blockade but mildly reduced in TIGIT KO T cells and remained unchanged in myeloid cells (new Extended Data Fig. 5 b-g).

To assess the functional impact of CD226 on Areg induction, we performed *in vivo* blockade in WT and TIGIT KO mice. CD226 blockade did not alter Areg or Blimp-1 expression in Tregs in either genotype (Fig. 3 l; Extended Data Fig. 9 k), suggesting that Areg and Blimp-1 downregulation in TIGIT KO mice are a direct effect of TIGIT deletion. While CD226 blockade resulted in modest changes of CD155 across T cell subsets (Extended Data Fig. 5 b), this had no functional impact on Areg production.

To examine the role of CD155 in TIGIT-mediated tissue repair, we analyzed CD155 KO mice. Upon LCMV infection, CD155 deletion leads to reduced Areg induction in Tregs, suggesting that CD155 acts as the TIGIT ligand that promotes Areg expression *in vivo* (new Fig. 3 m).

4. The manuscript states that TIGIT drives Blimp-1 expression, which subsequently induces Areg. However, the upstream signaling pathways leading to Blimp-1 induction are not fully elucidated. Please explore whether TIGIT engagement activates specific signaling cascades, such as STAT or PI3K pathways, contributing to Blimp-1 regulation.

Based on the reviewer's suggestion, we further explored the signaling pathways that are active in TIGIT⁺ cells and performed a SCENIC (single-cell regulatory network inference and clustering) analysis on our single-cell RNA-Seq data comparing WT and TIGIT KO cells. This analysis identified increased activity of transcription factors important for Treg development and stability (*Crem*, *Ets2*, *Helios*, *Junb*) in the presence of TIGIT (new Fig. 5 f, g, new Extended Data Fig. 8 g). Conversely, the activity of transcription factors downstream of TCR signaling (*Nfatc1*, *Tcf7*, *Jun*) was reduced. Nevertheless, TCR signaling, which induces Blimp-1 (Martins et al., 2006), remains active upon TIGIT engagement in Tregs as signaling intermediates, such as pAKT or pS6, were present upon *in vitro* engagement of TIGIT (new Extended Data Fig. 8 j, k), allowing for Blimp-1 induction.

5. The study identifies TIGIT⁺ Tregs as distinct from classical tissue-resident Tregs but does not explore the origin or specific phenotypic differences of this subset. Please include a detailed characterization of TIGIT⁺ Tregs, comparing their transcriptomic and proteomic profiles with classical tissue Tregs.

We performed transcriptional and proteomic analysis to compare the two populations. Using our single-cell RNA-seq dataset, we aligned the transcriptional profile of TIGIT⁺ repair Tregs with published signatures of classical tissue Tregs (Delacher et al., 2020) and found that TIGIT⁺ Tregs represent a transcriptionally distinct subset (new Fig. 4 f; new Extended Data Fig. 7 c). To facilitate direct comparison throughout the study, we also highlight classical tissue Tregs and their associated markers across multiple figures (Fig. 2 c, 4 f, g, 6 a; Extended Data Fig. 7 d-f). To further define phenotypic differences, we extended our flow cytometry panel to include canonical markers of tissue-resident Tregs, including ST2, Gata3 and KLRG1. These markers were largely absent in TIGIT⁺ Tregs induced during LCMV infection, clearly distinguishing the two populations at the protein level (new Fig. 4 g; Extended Data Fig. 7 d, e).

We also observed that LCMV infection broadly shifted the Treg compartment toward a higher frequency of peripherally induced Tregs, including both TIGIT⁺ and TIGIT⁻ populations, consistent with previous findings (Schorer et al., 2020) (new Extended Data Fig. 7 f). These results confirm that TIGIT⁺ Tregs represent an infection-induced, phenotypically and transcriptionally distinct population rather than classical tissue Tregs.

6. The reported Areg expression levels in some figures (e.g., Fig. 5I) are substantially higher than those in other experiments, raising questions about consistency.

Areg detection is technically challenging, but we have optimized our procedure and routinely include naïve samples as a reference to control for variability. However, we have observed substantial variability in Areg expression across different animal facilities. Notably, the data for Figure 6 I (previously 5 I) was generated in Melbourne, whereas most other experiments were conducted in Zurich, which may account for the observed differences.

7. Baseline Areg expression in naïve mice (Fig. 2H) is inconsistent with earlier claims that TIGIT is required for infection-induced Areg production. The authors should Clarify whether baseline Areg is truly TIGIT-independent or represents a different regulatory mechanism.

At steady state, Areg is equally expressed by both TIGIT⁺ and TIGIT⁻ Tregs (as shown in Fig. 2 h) and TIGIT loss on Tregs does not have an impact on Areg levels in T cells from naive mice. In contrast, TIGIT in Tregs is required for Areg induction following infection (Fig. 3 h), indicating that its role is context dependent. These findings suggest that TIGIT is dispensable for baseline Areg expression at homeostasis but becomes essential upon challenges requiring tissue repair. To further support this distinction, we now include a summary of Areg expression in WT versus full TIGIT KO mice and conditional TIGIT KO mice at steady state (new Extended Data Fig. 4 a, k).

8. Statements such as "TIGIT acts as a mediator of tissue repair upstream of Areg" and "TIGIT directly induces the expression of Areg" are overly definitive. The indirect contribution of TIGIT (e.g., via Tregs or accessory signals) remains a possibility.

While multiple pathways, including TIGIT-independent ones, can contribute to Areg production in T cells, our data demonstrate that TIGIT is required for Areg induction in Tregs upon infection. To

address the reviewer's concern, we have carefully revised the manuscript to avoid overgeneralizations and ensure that our conclusions are appropriately supported by the data.

9. *Fig. 1 lacks information about NK cell dynamics, which are known contributors to LCMV pathology and could be influenced by TIGIT deficiency.*

We analyzed the NK cell response by flow cytometry and assessed their cytokine and cytotoxic molecule expression in WT and TIGIT KO animals. We found no differences in IFN- γ , TNF- α or Granzyme B production between genotypes, indicating that NK cells do not contribute to tissue protection in a TIGIT-dependent manner and their impact on LCMV pathology is not influenced by TIGIT (new Extended Data Fig. 1 h).

Minor Comments

Figure-Specific Feedback:

1. *Figure 1A: The ALT levels in TIGIT KO mice without infection show a trend toward higher baseline pathology. A larger sample size might clarify significance.* We increased the number of samples for the naive mice in Figure 1 a as requested by the reviewer and could confirm that TIGIT KO mice do not display pathology at steady state.

2. *Figure 3H: Adjust sample sizes in *Foxp3^{GFPcreERT2}Tigit^{fl/fl}* Tregs to improve statistical.* We increased the number of steady state datapoints in Figure 3 h, which confirmed comparable Areg production by Tregs from WT and *Foxp3^{GFPcreERT2} x Tigit^{fl/fl}* mice at steady state, but a clear dependence on TIGIT for the induction of Areg in these cells upon infection.

4. *Figure 4I: Repeat experiments with purified T cells to confirm that observed effects are T.* Our data show that TIGIT ligation is required for Areg induction (Fig. 3 a, c, h, Fig. 5 e, 6 h), but we do not exclude that other cells in the culture provide these signals to TIGIT. As the TIGIT ligand CD155 is highly expressed in both T cells and myeloid cells upon LCMV infection (new Extended Data Fig. 5 b-g), ligand availability does not seem to be a limiting factor.

5. *Ensure consistent terminology when referring to "repair Tregs," "tissue Tregs," and "TIGIT+ Tregs" to avoid confusion.* We have revised the manuscript to ensure consistent and clear terminology. "Tissue Tregs" now refers specifically to the previously described Tregs residing in tissues and associated with repair functions (e.g., Delacher et al., Burzyn et al.). The population investigated in this study is consistently referred to as "TIGIT⁺ Tregs" or "TIGIT⁺ repair Tregs".

6. *The discussion does not address potential translational implications, such as targeting TIGIT for therapeutic modulation of tissue repair. Including this perspective would enhance the manuscript's clinical relevance.* We have added some references to this in the discussion.

Reviewer #3

In this manuscript, Panetti et al showed that upon viral infection, TIGIT directly drives expression of the tissue growth factor amphiregulin (Areg) in Treg cells and contributes to tissue repair. They also showed that TIGIT engagement upon TCR stimulation induces Blimp-1, promoting Areg production and tissue repair. The studies were well performed, and the data looks convincing. However, it has already been reported that Tregs produce Areg and contribute to tissue repair. It was also known that TIGIT is expressed on Tregs and that TIGIT limits tissue pathology during viral infection. For these reasons, this paper does not represent a significant conceptual advance.

We thank the reviewer for the positive assessment of our data. We agree that previous studies have established a role for Tregs in tissue repair and that our previous work demonstrated that TIGIT can limit tissue pathology during viral infection; this literature is cited in the introduction and discussion. To ensure clarity, we have further revised the manuscript to explicitly distinguish our findings from these prior studies. However, we respectfully disagree that our work lacks conceptual advance. While Tregs and TIGIT have been independently implicated in limiting pathology, no prior studies have suggested a link between TIGIT signaling and Treg-mediated tissue repair. Our study provides the first direct mechanistic evidence that TIGIT engagement induces Blimp-1, which drives Areg expression in Tregs to promote tissue repair. This represents a meaningful advance in understanding the molecular pathways that connect immune modulation to tissue regeneration.

The specific comments are as follows:

1. The authors showed that the treatment with TIGIT-specific antibodies suppressed the Areg production from Tregs. This suggests that the binding of TIGIT to its ligands is involved in Areg production. However, it is unclear which cells express the ligand and interact with Tregs. Clarification of this issue would help our understanding of the pathogenesis.

We thank the reviewer for raising this important point (a similar comment was made by Reviewer 2, see above). We analyzed expression of CD155, the primary ligand for TIGIT *in vivo*, to clarify which cell types may interact with TIGIT⁺ Tregs. CD155 was found to be highly expressed on T cells and, to a lesser extent, on innate immune cells. In myeloid cells, expression was upregulated in CD11b⁺CD11c⁺ cells following infection but was not affected by systemic or conditional TIGIT deletion (new Extended Data Fig. 5 b-g). CD155 expression was slightly reduced on TIGIT KO Tregs upon infection (new Extended Data Fig. 5 c, d), although overall expression levels remained high. These results support the idea that CD155-expressing immune cells, particularly activated myeloid and T cells, may interact with TIGIT⁺ Tregs in the infected tissue environment. We further assessed the effect of CD155 deletion on Areg expression and found that Tregs from CD155 KO mice failed to upregulate Areg to the level observed in WT mice, without altering TIGIT expression (new Fig. 3 m; new Extended Data Fig. 5 j, k). These findings suggest that CD155 serves as the ligand for TIGIT-mediated induction of Areg *in vivo*.

2. TIGIT and DNAM-1 compete for ligand binding each other. Since TIGIT antibody treatment or TIGIT deficiency promotes the binding of DNAM-1 to CD155, the DNAM-1 downstream signaling might be involved in the Areg production. To clarify this issue, it would be helpful to use an anti-DNAM-1 antibody with anti-TIGIT antibodies or single or double knockout mice for TIGIT and

DNAM-1.

To investigate the potential role of CD226 (DNAM-1) signaling in Areg induction, we analyzed CD226 expression and performed *in vivo* CD226 blockade in both WT and TIGIT KO mice (see also response to a related comment above, as well as to comment 3 by Reviewer 2). CD226 expression on Tregs and other T cell subsets remained mostly unchanged upon TIGIT deletion (new Extended Data Fig. 5 h, i). Importantly, CD226 blockade had no functional effect on Areg or Blimp-1 expression in Tregs from either WT or TIGIT KO mice (new Fig. 3 l; new Extended Data Fig. 9 k). These findings confirm that the reduction in Areg expression observed in the absence of TIGIT is not an indirect consequence of enhanced CD226 signaling, but rather a direct result of diminished TIGIT signaling.

3. The authors showed that TCR stimulation is important for TIGIT-induced production of Areg in an in vitro assay in Fig. 4, using anti-CD3 antibodies. What stimulates the TCR of Tregs in vivo? Are the Areg+ Tregs LCMV-specific Tregs? The authors should discuss the necessity of TCR stimulation in in vivo Areg+ Tregs during LCMV infection, taking into account the immunological mechanisms.

We analyzed the frequency of LCMV-specific Tregs within the TIGIT⁺Areg⁺ Treg population induced upon LCMV infection using tetramers. As previously reported (Rost et al., 2020), LCMV-specific Tregs are exceedingly rare. While we readily detected LCMV-specific Areg⁺ conventional T cells, tetramer-positive cells were virtually absent among TIGIT⁺Areg⁺ Tregs (new Extended Data Fig. 8 b, c). These findings suggest that most TIGIT⁺Areg⁺ Tregs arising during LCMV infection are not LCMV-specific but are likely activated by self-antigens released from damaged tissue. We have updated the Results and Discussion sections to reflect this.

4. In Fig. 2, there is a big difference in the production of Areg by TIGIT between LCMV CL13-infected and LCMV WE-infected mice. The authors should explain the mechanisms of this difference. Also, in Fig. 2h, it is unclear how TIGIT promotes the production of Areg only in LCMV-infected mice but not in naive mice.

The differences in Areg expression between these models are primarily due to variations in tissue pathology, antigen load, and the resulting immune response. LCMV CI13 infection is performed with a high virus dose (2×10^6 FFU) to induce a chronic infection with significant tissue damage, resulting in higher antigen load and leading to a stronger effector response. In contrast, LCMV WE infection is elicited with a much lower viral load (200 FFU) resulting in an acute infection with reduced tissue pathology (Extended Data Fig. 1 d-f). These factors contribute to the differential induction of Areg.

Our new data (new Figure 5 d; new Extended Data Fig. 8 d, e) show that naïve Tregs require additional co-stimulatory signals beyond TCR and TIGIT engagement to induce Areg expression. In contrast, Tregs from LCMV infected mice have undergone *in vivo* priming, rendering TCR and TIGIT signaling alone sufficient to re-stimulate Areg production. This distinction underscores the differential signaling requirements for activating naïve or resting vs primed T cells.

Minor

1. Label in Fig 3f, g, l, j, and k, Ext Fig 3c and d, Ext Fig 4c and e-g. We have moved the labels in these figures to make it clearer that they refer to all panels of the respective figure.